# SUPER BAYESIAN FRAMEWORK:
# BAYESIAN ANALYSIS WITH REDUNDANT OBJECTS

## ABSTRACT

This paper proposes a fundamentally new framework for Bayesian machine learning (BML) by introducing redundant objects inspired by *extremal combinatorics*. In general, BML offers the advantage of quantifying uncertainty across all possible models (hypotheses) that explain data (observable phenomena in nature and society), making it particularly useful in applications that require transparency in data analysis, such as biomedical health and finance. However, its practical use is hindered by challenges in (1) model construction (often involving measure theory), (2) inference algorithm design (such as Markov chain Monte Carlo (MCMC) tailored to individual case studies), and (3) theoretical analysis (such as the mixing time of MCMC). This paper aims to mitigate the aforementioned obstacles, particularly for BML that infers combinatorial structures (permutations, partitions, binary sequences, binary trees, acyclic directed graphs, Cambrian trees, rectangular partitions, etc.) including consensus ranking, classification, factor analysis, hierarchical clustering, causal inference, phylogenetic analysis, and relational data analysis. Our key insight is to apply redundant universal objects, inspired by extremal combinatorics, as general-purpose generative probabilistic models independent of specific application scenarios. Our contributions include (1) a unified model construction method that expresses random target objects by randomly selecting a substructure from the universal objects, (2) a simple MCMC algorithm for Bayesian inference that is naturally derived from this model representation, and (3) a clue to the theoretical upper bound on the MCMC mixing time.

## 1 INTRODUCTION

**General framework of Bayesian machine learning (BML)** - BML is a statistical data analysis method that evaluates the uncertainty of various hypotheses (models) that explain a phenomenon from observed phenomena (data) in the natural sciences or society. We obtain some observational information $X = (\mathbf{x}_i)_{i=1}^n$ (e.g., $\mathbf{x}_i$ consists of $K$-dimensional features) from $n$ ($\in \mathbb{N}$) trials for the phenomenon of interest as a random variable. It generally contains various uncertainties, such as noise from the environment or errors in the observation system. We aim to examine a hypothesis $\theta$ that describes the mechanism of this phenomenon $X$. We can consider a set $\Theta$ of all possible hypotheses, and for each hypothesis $\theta \in \Theta$ we design a prior probability $P_{\text{prior}}(\theta)$ from our prior knowledge (if we have no prior knowledge, we use the non-informative prior). We measure the degree to which the hypothesis $\theta$ fits the observed phenomenon $X$ using the likelihood model $P_{\text{likelihood}}(X \mid \theta)$. Finally, we use Bayes' rule to evaluate the uncertainty of $\theta$ in the following posterior probability.

$$\textbf{Bayes' rule}: \quad P_{\text{posterior}}(\theta \mid X) \propto P_{\text{prior}}(\theta) \cdot P_{\text{likelihood}}(X \mid \theta). \tag{1}$$

**Essential advantages of BML** - We would have three main hopes when utilizing BML. **[A1] Model universality**: Under appropriate design, BML does not miss any hypothesis $\theta$ in the hypothesis space $\Theta$ of interest, that is, $P_{\text{posterior}}(\theta \mid X) > 0$ for any $\theta \in \Theta$. In particular, *Bayesian nonparametrics* (BNP) includes $\theta$ that requires *infinite* description length. **[A2] Transparency in data analysis**: The essence of BML lies not only in finding plausible hypotheses, but also in evaluating the certainty $P_{\text{posterior}}(\theta \mid X)$ of candidate hypotheses, where it is important for users to be explicit about the risks associated with AI decisions and suggestions. **[A3] Versatility in various applications**: As shown in Table 1, BML can deal with various application cases in a unified manner using the form of Bayes' rule (Equation 1) by incorporating various combinatorial structures within hypothesis $\theta$.

Table 1: Several applications through super Bayesian framework

| Task | Combinatorial structures | Application examples |
|------|--------------------------|----------------------|
| Traveling salesperson (Sec. 2) | Permutation | Consensus ranking, DNA sequencing |
| Classification (App. D.3) | Partition | Image classification, Bag of words |
| Hierarchical clustering (App. D.4) | Binary tree | Fine-grained classification |
| Factor analysis (App. D.5) | Binary sequences | Feature selection, Discovery of frequent patterns |
| Causality inference (App. D.6) | Directed acyclic graph | Discovery of disease causative factors |
| Relational data analysis (App. D.7) | Rectangulation | Collaborative filtering, Community detection |
| Phylogenetic analysis (App. D.8) | Cambrian tree | Inferring transmission routes during pandemic |

**Obstacles faced by current BML** - In current practical applications of BML, are its inherent advantages — *[A1] model universality, [A2] transparency in data analysis, [A3] versatility in various applications* — being realized in practice? Unfortunately, it would be no exaggeration to say that BML is far from reaching its full potential. We discuss its main causes from the three aspects:

O1 **Difficulty of model construction** - It has become quite challenging to construct state-of-the-art Bayesian models (discussed in Appendix A; specifically, stochastic processes with infinite-dimensional parameter spaces, such as those used in the BNP framework). We need to be aware of various requirements in model construction, such as *projectivity* (Bochner, 1955) and *exchangeability* (Aldous, 1981; Hoover, 1979; Kallenberg, 1992). We may also need a deep knowledge of measure-theoretic probability theory if we wish to create essentially new models.

O2 **Complication of deriving inference algorithms** - Effective inference algorithms often require highly complicated implementations, such as customized MCMC algorithms tailored to specific applications. The design of unified general-purpose algorithms is still a developing research topic (Naderiparizi et al., 2022; Lai et al., 2023; Liang et al., 2023; Barbarossa & Pistone, 2025; Stites et al., 2025; Stein & Staton, 2024; Becker et al., 2024). Even if we come up with a very innovative new model, we may have to be prepared to spend a great deal of time and effort in deriving the inference algorithm, which has undermined the inherent *[A3] versatility of BML*.

O3 **Intricacy of algorithmic theoretical properties** - Theoretical analysis of algorithms (e.g., the mixing time of MCMC) is generally not straightforward, with only a few exceptions (Peres & Sousi, 2015; Yang et al., 2016; Zhuo & Gao, 2021). In practice, it is typical to iterate the sequential inference algorithm as much as computational resources allow, or until one local mode is found. However, there are very few cases where a theoretical justification for such practical heuristics can be obtained, which has undermined the inherent *[A2] transparency of BML*.

**Our key idea: Use of redundant universal objects inspired by extremal combinatorics** - To reduce these obstacles, we need a *model that is like magic* with [O1] rich expressive power (similar to the stochastic processes utilized in the BNP framework) while providing inference algorithms that are [O2] easy to derive and [O3] theoretically transparent. Our project began with this motivation, inspired by the *magical phenomena* observed in extremal combinatorics[1]. For example, here we focus on *permutations* used in ranking, matching, or *traveling salesperson problem*. If we consider a random permutation of length $n$ (that is, a probabilistic model of all possible permutations of length $n$) as the model/hypothesis for some BML application, the number of possible cases reaches $n!$, which is difficult to handle on a computer. However, surprisingly, there is *a magical sequence of length $n(n+1)/2$ that can handle all possible permutations of length $n$* (Miller, 2009):

**Proposition 1.1** (Theorem 3.1 in Miller (2009)). *The zigzag word consists of the sequence of numbers in ascending order of odd numbers and descending order of even numbers:*

$$\underbrace{1357\ldots}_{\text{1st run}}\underbrace{\ldots 8642}_{\text{2nd run}}\underbrace{1357\ldots}_{\text{3rd run}}\underbrace{\ldots 8642}_{\text{4th run}}\underbrace{1357\ldots}_{\text{5th run}},\ldots. \tag{2}$$

*For all $n \geq 1$, the zigzag word restricted to the alphabet $[n+1] = \{1, 2, \ldots, n+1\}$ with $n$ runs contains subsequences order-isomorphic to every permutation of length $n$.*

This sequence is called *superpermutation* (Engen & Vatter, 2021) and is a representative example of the "**universal object**" in extremal combinatorics, including the *supertree* (Defant et al., 2020) and the *supergraph* (Alon, 2020). The above *zigzag word* is an example of simple superpermutation

---

[1]Extremal combinatorics, as its name suggests, typically deals with "extremal" (minimum or maximum) objects where certain special phenomena occur. In this example, a typical research interest in extremal combinatorics is what the "smallest" sequence containing all permutations of length $n$ is (Engen & Vatter, 2021).

realizations. For the case of $n = 5$, the zigzag word 135642135642135 contains as its subsequence "order-isomorphic to" every permutation of length $n = 5$. For example, **31524** can be extracted as 1**3**5642**1**3**5**64**2**135**4**2135. Also, **54312** can be obtained by extracting 135**6**4213**56**42**13**5 as a subsequence and then replacing it with the permutation **54312** order-isomorphic to **65413**.

This superpermutation notion provides us with a simple but powerful idea: We could represent a random permutation as "a random subsequence of the superpermutation." In the following, this paper will formalize this simple idea into a new formal framework for BML, which we call *Super Bayes* (SB), by analogy with the name, *super*permutation. Our claims in this paper are summarized as:[2]

- **What is the SB framework?** - It is a conceptually new BML scheme based on a strategy of representing models as "a form of randomly extracting a part of universal objects of extremal combinatorics, such as superpermutations (Engen & Vatter, 2021) and supergraphs (Alon, 2020)."

- **How does SB work?** - This modeling strategy leads us to a common model structure: a probability model on a family of substructures for the universal object. Such a model form of "the probability model on a family of sets" has developed deep theoretical insights into MCMC inference algorithms, such as the strongly Rayleigh measure (Anari et al., 2016; Li et al., 2016b; Mariet et al., 2018). Our SB benefits from such findings and derives a unified inference algorithm for various BMLs.

- **Why is SB significant?** - **(1) Model universality** ([A1], [O1]): Thanks to the universal object nature, *the SB framework can handle rich universal models* (i.e., take all models/hypotheses of interest). **(2) Unified MCMC inference** ([A3], [O2]) - The SB framework's model representation of "extraction of a random part from the universal object" *induces a unified MCMC algorithm* of sequential updating of subsets of a large set in general, independent of individual BML application cases. This eliminates the need to derive complicated inference algorithms for each specific application case, enabling the use of a unified general-purpose MCMC algorithm. **(3) Theoretical analysis of MCMC mixing time** ([A2], [O3]) - Our SB framework can provide a clue to *an upper bound on the mixing time for MCMC*. Traditionally, with a few exceptions (Peres & Sousi, 2015; Yang et al., 2016; Zhuo & Gao, 2021), the theoretical relevance of practical MCMC algorithms has been mostly unclear. In contrast, our SB framework might potentially identify bottlenecks in individual problems or provide insights to improve likelihood models for efficient inference.

- **What are the current scope and limitations?** - **(a) Focus on discrete combinatorial structure inference** ([O1]): The scope of this paper focuses on BML for discrete combinatorial structures and does not cover BML for continuous quantities such as regression. **(b) Residuals of problem-dependent terms in MCMC mixing time bounds** ([O3]): Our MCMC mixing time analysis includes a problem-dependent term. While we describe how to handle this term for several specific cases (Appendix B.4 and Remark D.3), a general approach for all cases remains a future issue.

## 2 ILLUSTRATIVE EXAMPLE OF SUPER BAYESIAN FRAMEWORK

This section describes the entire flow of our SB framework, from **(1) problem setting** (Section 2.1), **(2) SB reformulation** (Section 2.2), **(3) derivation of a MCMC inference algorithm** (Section 2.3), **(4) its mixing time analysis** (Section 2.4), to **(5) practical MCMC acceleration** (Section 2.5), through a single illustrative example. Specifically, since we aim to utilize the *superpermutation* as a representative tool of the SB framework, we will address the *traveling salesperson problem* as the most straightforward application example (though it may seem somewhat unusual in the context of machine learning). We will discuss the generalization of the SB framework in the next section.

**Notations** - Let $\mathbb{N}$, $\mathbb{R}$, and $\mathbb{R}_+$ be the sets of all natural numbers, all real numbers, and all non-negative real numbers, respectively. We also frequently use the subset $[n] := \{1, 2, \ldots, n\}$ of natural numbers from 1 to $n$ ($n \in \mathbb{N}$). Let $\Lambda_n$ be the set of all permutations of $[n]$. We write each permutation $\sigma \in \Lambda_n$ as $\sigma = \sigma(1)\sigma(2)\ldots\sigma(n)$. For example, for $\sigma = 3241$ ($\in \Lambda_4$), we have $\sigma(1) = 3$, $\sigma(2) = 2$, $\sigma(3) = 4$, and $\sigma(4) = 1$. We will use this notation for generic sequences (not necessarily permutations) as well. For example, for a sequence $\sigma' = 3234$, we have $\sigma'(1) = 3$, $\sigma'(2) = 2$, $\sigma'(3) = 3$, and $\sigma'(4) = 4$. We refer to the *support* of a distribution $\pi : \Theta \to \mathbb{R}_+$ as $\text{supp}(\pi) := \{\theta \in \Theta \mid \pi(\theta) > 0\}$. Finally, we write $2^V$ for all subsets of a certain (discrete) set $V$.

---

[2]**Related work** - We provide a comprehensive survey of previous research in **Appendix** A. **How to exhibit theoretical results** - This paper makes use of many existing and excellent findings. To distinguish them, we **color-box** the notions and theorems introduced in this paper. All proofs are given in **Appendix**.

## 2.1 PROBLEM SETTING: TRAVELING SALESPERSON PROBLEM AND ITS BAYESIAN EXTENSION

**Traveling salesperson problem (TSP)** - An undirected weighted graph $G_{\text{obs}} = (V, D)$ is given, where $V = [n] = \{1, 2, \dots, n\}$ is a set of cities indexed by the natural numbers $[n]$, and $D = (d_{\text{TSP}}(i, j))_{i,j \in V}$ represents the edge weights. Each $d_{\text{TSP}}(i, j)$ indicates the distance traveled from city $i$ to city $j$. We suppose that $d_{\text{TSP}}(i, j) > 0 \ (i \neq j)$ and $d_{\text{TSP}}(i, i) = 0$. The objective of TSP is to find the shortest possible route that visits each city exactly once and returns to the origin city. Namely, it is to find a permutation $\sigma \ (\in \Lambda_n)$ that minimize the following total length $\text{TSPdist}(\sigma; G_{\text{obs}})$:

$$\text{TSPdist}(\sigma; G_{\text{obs}}) := d_{\text{TSP}}\big(\sigma(1), \sigma(2)\big) + \cdots + d_{\text{TSP}}\big(\sigma(n-1), \sigma(n)\big) + d_{\text{TSP}}\big(\sigma(n), \sigma(1)\big). \quad (3)$$

TSP is known as one of the representative examples of NP-hard problems (Arora & Barak, 2006).

**Bayesian TSP** - Apart from the original TSP, its probabilistic formulation has also been a very interesting subject (Jaillet, 1985; Bellmore & Nemhauser, 1968; Bertsimas & Howell, 1993), since in practice the TSP may be affected by a certain noise or randomness due to several environmental factors. We here discuss a Bayesian extension of TSP. We first introduce the following likelihood model $P_{\text{likelihood}}(G_{\text{obs}} \mid \sigma)$ of the possible routes $\sigma \in \Lambda_n$: $P_{\text{likelihood}}(G_{\text{obs}} \mid \sigma) \propto \exp\big(-\beta \cdot \text{TSPdist}(\sigma; G_{\text{obs}})\big)$, where $\beta > 0$ is a (scalar) inverse temperature parameter. Then we consider a prior model $P_{\text{prior}}(\sigma)$ on possible routes. For example, we can simply employ the uniform prior: $P_{\text{prior}}(\sigma) = 1/n!$. Finally, owing to Bayes' rule (Equation 1), the objective of the Bayesian TSP is to infer the posterior distribution $P_{\text{posterior}}(\sigma \mid G_{\text{obs}}) \propto P_{\text{prior}}(\sigma) \cdot P_{\text{likelihood}}(G_{\text{obs}} \mid \sigma)$.

## 2.2 SUPER BAYESIAN REFORMULATION

The core principle of our SB framework is to represent the model in a unified way in the form of "the random extraction of a subset from a universal object" (as discussed in Section 1, **What is the SB framework?**). We will reformulate the Bayesian TSP according to this policy. We begin by introducing the *superpermutation* notion, which are useful for representing permutations.

**Superpermutation** - We generally refer to a sequence that contains a subsequence that is order-isomorphic to any permutation $\sigma \in \Lambda_n$ as a *superpermutation* (See also Section 1, **Our key idea**). Among many different constructions and examples of superpermutations (See recent excellent survey in (Engen & Vatter, 2021), especially Table 1), the zigzag word is probably the simplest. However, this is not well suited for direct application to BML for two reasons: (R1) We must pay attention to equivalence in order isomorphisms between a permutation (e.g., **54312**) and a sequence (i.e., not necessarily a permutation; e.g., **65413**) when focusing on a subsequence of the superpermutation; (R2) Any bias in the numbers appearing in the superpermutation can lead to an unintended bias against the prior model.[3] Considering these points, we wish to have a superpermutation whose alphabet is $[n]$ and whose occurrences of each number are of equal degree. Therefore, we propose the following *random permutation concatenation* (RPC) that satisfies both *universality* and *uniformity*.

> **Definition 2.1** (Random permutation concatenation). Let $\rho_n$ be a random sequence of length $n^2$ such that $\rho\big(1 + n(i-1)\big)\rho\big(2 + n(i-1)\big) \dots \rho\big(n + n(i-1)\big) \sim_{\text{i.i.d.}} \text{Uniform}(\Lambda_n)$ for $i = 1, \dots, n$. That is, $\rho_n$ is a concatenation of $n$ uniformly random permutations of length $n$.

We will denote the probability of RPC $\rho_n$ by $P_{\text{RPC}}(\rho_n)$. Let $\Xi_n$ denote the set of all possible concatenations of $n$ permutations of $[n]$. By construction, we can immediately see its universality.

> **Proposition 2.2** (Universality). *RPC $\rho_n$ contains subsequences equal to every permutation of $[n]$.*

**Random subsequence of superpermutation** (Figure 1 (a)-(c)) - Now that we have introduced the superpermutation, we will describe how to represent a random permutation as a random subsequence of the superpermutation. First, it must be noted that a subsequence of the superpermutation is not necessarily a permutation.[4] Thus we need a mechanism to restrict the random subsequence to a permutation. To do so, we introduce the following *partition matroid* strategy. Let $\mathcal{P}_i$ be a set

---

[3]For example, the sequence 1234512341523145213 is a superpermutation for $n = 5$. Many permutations of [5] appear in this as multiple subsequences. However, the permutation **54321** appears in only one subsequence 1234**5**123**4**152**3**14**5**2**1**3. This means that it induces that **54321** is less likely to appear than other permutations.

[4]For example, a subsequence **44213** of $\rho_5 = 1325\underline{4}31\underline{4}5241\underline{2}351\underline{4}235\underline{3}5412$ is a sequence of length 5, but it is not a permutation of $[5] = \{1, 2, 3, 4, 5\}$. In contrast, a subsequence **54213** is exactly a permutation.

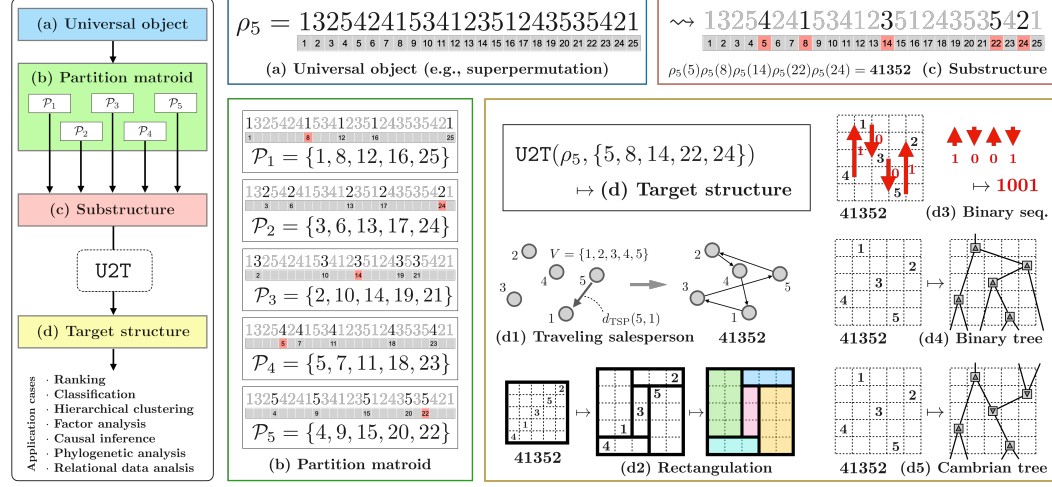

Figure 1: Overview of Super Bayesian framework (left panel) and its specific configuration (right four panels), which provides a generative probabilistic model that addresses various combinatorial structures (for typical machine learning tasks) in a unified manner. **(a) Universal object** - By the combinatorics analogy of the lottery ticket hypothesis for neural networks, we first introduce the universal object of extremal combinatorics as a "**redundant**" object including the supertree (Defant et al., 2020) and the supergraph (Alon, 2020). **(b) Partition matroid** - Next, we construct partition matroids as a mechanism for handling a random substructure derived from the universal object. **(C) Substructure - (d) Target structure** - Through the transformation U2T, the random substructure serves as a probabilistic model for the desired target structure, such as permutations (d1), partitions, binary trees (d4), binary seq. (d3), causal graphs, Cambrian trees (d5), and rectangulations (d2).

$\mathcal{P}_i := \{k \mid \rho_n(k) = i\}$ (Figure 1 (b)). Then we define the partition matroid $\mathcal{C}$ as $\mathcal{C} := \{S \subseteq [|\rho_n|] = \{1, 2, \ldots, |\rho_n|\} \mid \forall i, |S \cap \mathcal{P}_i| = 1\}$. Intuitively, this means that each $S \in \mathcal{C}$ will always pick one element from each $P_i$. Since each $P_i$ refers to the index set from which the superpermutation outputs $i$ (i.e., $\rho_n(k) = i$ for any $k \in \mathcal{P}_i$), each element of $\mathcal{C}$ can extract a subsequence from the superpermutation that will always be a permutation (Figure 1 (c)). Without loss of generality, for $S \in 2^{[|\rho_n|]}$, we will write $S = \{s_1 < s_2 < \cdots < s_n\}$ (elements in ascending order) as a convenient notation to specify a subsequence $\rho_n(s_1)\rho_n(s_2)\ldots\rho_n(s_n)$ of the superpermutation $\rho_n$ (See also the intuitive example in Figure 1 (c)). Finally, given the superpermutation $\rho_n$, we consider the following conditional probability $P_{\mathcal{C}} : 2^{[|\rho_n|]} \to \mathbb{R}_+$ on subsequences $\rho_n(s_1)\ldots\rho_n(s_n)$:

$$P_{\mathcal{C}}\Big(S \in 2^{[|\rho_n|]} \mid G_{\text{obs}}, \rho_n\Big) \propto P_{\text{likelihood}}\Big(G_{\text{obs}} \mid \rho_n(s_1)\rho_n(s_2)\ldots\rho_n(s_n)\Big)\mathbb{I}[S \in \mathcal{C}], \qquad (4)$$

where $\mathbb{I}[\cdot]$ is the indicator function, and $P_{\text{likelihood}}$ is given by Section 2.1. This SB reformulation can be attributed to the original Bayesian TSP (described in Section 2.1):

> **Proposition 2.3** (Equivalence to Bayesian TSP). *We consider the original Bayesian TSP $P_{\text{posterior}}$ given in Section 2.1 (along with a uniform prior $P_{\text{prior}}(\sigma \in \Lambda_n) = 1/n!$). For any $\sigma \in \Lambda_n$, we have $P_{\text{posterior}}\big(\sigma \mid G_{\text{obs}}\big) = \sum_{\rho_n \in \Xi_n} \sum_{\{S \mid \rho_n(s_1)\ldots\rho_n(s_n)=\sigma\}} P_{\mathcal{C}}(S \mid G_{\text{obs}}, \rho_n) \cdot P_{\text{RPC}}(\rho_n).$*

**Target distribution to infer** - As indicated in Proposition 2.3, it would be appropriate to approximate $P_{\text{posterior}}$ by using multiple sample of RPC $\rho_n$, since RPC is marginalized for all cases $\sum_{\rho_n \in \Xi_n}$. This implies that using as many RPC samples as possible ensures the uniformity of subsequences. However, using a large number of RPC samples is time-consuming, so in practice, there are often situations where we want to use only a few or even a single RPC sample. For simplicity, this paper will proceed using a single RPC. Hence, we aim to infer the target distribution $\pi_{\mathcal{C}}(S) := P_{\mathcal{C}}(S \mid G_{\text{obs}}, \rho_n)$ (from Eq. 2.3), given a *fixed* random generated RPC sample $\rho_n$. While this may potentially sacrifice uniformity of the subsequence, as demonstrated in Appendix E.1, this risk diminishes as $n$ increases.

## 2.3 UNIFIED GENERAL-PURPOSE MARKOV CHAIN MONTE CARLO

Thanks to our SB framework (described in Section 2.2), we have been able to represent the original Bayesian TSP as "*a probability model on a family of subsets,*" $S \in 2^{[|\rho_n|]}$, of the whole set

---

**Algorithm 1:** Unified general-purpose MCMC for super Bayesian models (Figure 6 in Appendix)

---

**Input:** Partition matroid $\mathcal{C} = \cup_{i=1}^n \mathcal{P}_i$ and universal object $\rho_n$ that induce target distribution $\pi_\mathcal{C}$.
**Output:** MCMC samples of $S \in \mathcal{C}$ that approximate target distribution $\pi_\mathcal{C}$.

1   Initialize $S \in \mathcal{C}$, e.g., $s_i \sim \text{Uniform}(\mathcal{P}_i)$ $(i = 1, \ldots, n)$ and $S \leftarrow \{s_1\} \cup \cdots \cup \{s_n\}$ ;
2   **while** *MCMC is not mixing into the stationary distribution* **do**
3     $b \sim \text{Bernoulli}(1/2)$ ;   /* Laziness (for avoidance of periodicity) */
4     **if** $b = 1$ **then**
5       $s \sim \text{Uniform}(S)$ ;           /* Selection of update location */
6       $s' \sim \text{Uniform}(\{k \mid \rho_n(k) = \rho_n(s) \land k \neq s\})$ ;    /* Replacement element */
7       $S \leftarrow (S \setminus \{s\}) \cup \{s'\}$ with probability $\frac{\pi_\mathcal{C}((S \setminus \{s\}) \cup \{s'\})}{\pi_\mathcal{C}(S) + \pi_\mathcal{C}((S \setminus \{s\}) \cup \{s'\})}$ ;     /* Gibbs */

---

$[\|\rho_n\|] = \{1, 2, \ldots, |\rho_n|\}$ (i.e., the index set of a superpermutation $\rho_n$). From a bird's-eye view, in general, this type of *the probability model for a family of subsets* has been thoroughly discussed in terms of unified and general-purpose MCMC inference algorithms, such as the strong Rayleigh measure (Anari et al., 2016; Li et al., 2016b; Mariet et al., 2018) and the determinantal point process (Song et al., 2024; Grosse et al., 2024; Barthelmé et al., 2023; Han et al., 2022b;a; Hemmi et al., 2022; Derezinski, 2019; Rezaei & Gharan, 2019; Derezinski et al., 2019; Li et al., 2016a). The strategic approach to inference algorithm design in our SB framework lies in leveraging MCMC for these well-studied "*probabilistic models of a family of subsets.*"

**Unified general-purpose MCMC** - Algorithm 1 represents the pseudocode of our MCMC algorithm. It is based on a simple Gibbs exchange method that sequentially updates one location from a subsequence of the superpermutation. First, we initialize the subsequence $\rho_n(s_1) \ldots \rho_n(s_n)$ for some $S = \{s_1 < \cdots < s_n\}$ (**Line 1**). In the subsequent MCMC loop, we repeatedly select one element $s$ from the subsequence uniformly at random and replace it with another element $t$ according to a certain probability (**Lines 5-7**). The procedure of becoming *lazy* with probability $1/2$ (**Lines 3-4**) is a theoretical requirement to avoid periodicity of MCMC iterations, but it can be ignored in practice.

**Non-triviality of Algorithm 1** - Some readers may feel that Algorithm 1 works only as a run-of-the-mill update rule to the TSP. That feeling is valid. In fact, it is simply an iterative update that tries to select one city sequentially and change its order. This type of update rule is already a textbook strategy and has been repeatedly discussed and widely used, so it may not seem to provide any important insights in the context of TSP. The point we wish to emphasize here is that **our SB framework allows this simple MCMC to be systematically and automatically applied to several applications**. Indeed, for other BML applications (discussed in Appendix D), it leads to non-trivial update rules.

## 2.4   Mixing time analysis of MCMC inference

Fortunately, Algorithm 1 can be theoretically evaluated in terms of its efficiency and effectiveness based on the concept of *mixing time* of MCMC. We begin with its formal definition (Sinclair, 1992).

**Definition 2.4** (Mixing time). Let $\mathbf{T}$ be the transition matrix of the Markov chain induced by Algorithm 1, with stationary distribution $\pi_\mathcal{C}$. We define its mixing time as $\tau_{S_0}(\epsilon) = \min\{t \geq 0 : \|\mathbf{T}^t(S_0, \cdot) - \pi_\mathcal{C}(\cdot)\| \leq \epsilon\}$, where $\|\cdot\|$ denotes the total variation distance and $\mathbf{T}^t(S_0, \cdot)$ represents the distribution after applying the state transition $\mathbf{T}$, $t$ times, from the initial state $S_0 \in \mathcal{C}$.

Intuitively, the mixing time refers to the number of MCMC iterations that must be performed before the sample can be considered to follow the target distribution $\pi_\mathcal{C}$ from some initial state $S_0 \in \mathcal{C}$. Owing to the previous great findings (Gotovos et al., 2015; Anari et al., 2016), our Algorithm 1 can be theoretically analyzed as follows.

**Theorem 2.5** (Upper bound). *For super Bayesian TSP (described in Section 2.2) with the cities topology $G = (V, D)$ (Section 2.1), Algorithm 1 has the following upper bound of its mixing time:*

$$\tau_{S_0}(\epsilon) \leq 4n^2 \max_i |\mathcal{P}_i| \exp\left(2\zeta_{\text{likelihood}}(G_{\text{obs}})\right) \left(\log \pi_\mathcal{C}(S_0)^{-1} + \log \epsilon^{-1}\right). \tag{5}$$

*where, using a temporary abbreviation $f(S) = -\log P_{\text{likelihood}}\left(G_{\text{obs}} \mid \rho_n(s_1)\rho_n(s_2) \ldots \rho_n(s_n)\right)$, we have $\zeta_{\text{likelihood}}(G_{\text{obs}}) = \max_{S,S' \in \mathcal{C}} |f(S) + f(S') - f(S \cap S') - f(S \cup S')|$.*

**Interpretation** - Equation (26) provides us with the following two important insights. **(1) Price of redundancy**: The SB effect of *deliberately making the model redundant* appears in the term $\max_i |\mathcal{P}_i| = n$. Our SB framework pays this price in order to achieve a unified general-purpose MCMC and its mixing time transparency. **(2) Problem-dependent term**: The effect of the likelihood on individual tasks appears in the term $\zeta_{\text{likelihood}}$. This term can be interpreted as the degree of deviation from *modularity* (Gotovos et al., 2015; Anari et al., 2016). The function $\zeta_{\text{likelihood}}$ design such that it reduces the degree of deviation from the modularity described above is an important research topic on its own in contexts such as discrete optimization and matroid theory. Our SB framework enables us to import their insights into BML and feed them back into the design of new likelihood models. We will show an example for the case of mixture models in Remark D.3.

## 2.5 (OPTIONAL) PRACTICAL ACCELERATION WITH QUANTUM SYSTEM

One significant benefit of a general-purpose inference algorithm is that improvements are not confined to a single application but can be reflected across many applications. Here, as one demonstration, we introduce one of the most powerful and non-trivial methods: acceleration through classical approximation of quantum systems, similar to the *quantum annealing* (Finnila et al., 1994; Kadowaki & Nishimori, 1998).

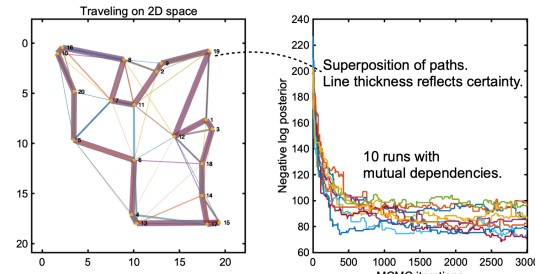

Figure 2: Quantum MCMC for Bayesian TSP

Our strategy involves the introduction of the superposition effect into the partition matroid $\mathcal{C}$. The SB framework deals with a single random element from each component $\mathcal{P}_i$ of the partition matroid $\mathcal{C}$, each of which can be regarded as a classical mixture model. Leveraging this insight, we can construct a quantum system by replacing this collection of classical statistical mixtures with a collection of "quantum superpositions." Appendix C details the derivation, and Algorithm 2 presents a special case using the Suzuki-Trotter decomposition (Suzuki, 1976; Trotter, 1959) for approximate simulation on classical computers. Roughly speaking, the quantum MCMC can be approximated by performing Algorithm 1 in infinitely parallel computation while introducing minor modifications to induce "quantum" dependencies between them. Figure 2 illustrates this parallel computation with a granularity of 10, showing the inference of Bayesian TSP for 20 cities on a two-dimensional plane.

## 3 GENERAL OVERVIEW OF SUPER BAYESIAN FRAMEWORK

The purpose of this section is to clarify a general overview of our SB framework, which can be summarized as the following five procedures. The fourth procedure is essentially the crucial step.

---

**Super Bayesian framework: Bayesian analysis with extremal combinatorics**

- **[S1] Observation data** - We are given some data $G$. (As discussed in Section 2.1, for TSP, $G = (V, D)$ consists of city indices $V$ and their pair-wise distance $D = (d_{\text{TSP}}(i, j))_{i,j \in V}$.)
- **[S2] Target structure design** - We hypothesize a latent *target structure* hidden in the observation data. (For TSP, it is a *permutation* of cities indexed by $[n]$.) Let $\mathcal{T}$ be a set of all possible target structure samples. (For TSP, $\mathcal{T}$ is a set of permutations of length $n$, that is, $\mathcal{T} = \Lambda_n$.)
- **[S3] Likelihood design** - We set a likelihood model $P_{\text{likelihood}}(G \mid t \in \mathcal{T})$. (Sec. (2.1) for TSP.)
- **[S4] Key: Introduction of universal object and partition matroid** - We introduce a convenient *universal object* $\rho_n$ and a *partition matroid* $\mathcal{C} = \cup_{i=1}^n \mathcal{P}_i$ for a family $2^{[|\rho_n|]}$ of subsets of the index set $[|\rho_n|]$ of $\rho_n$. We then construct a mapping $\text{U2T} : (\rho_n, S) \mapsto t$ from the pair of the universal object $\rho_n$ and its index subsets $S \in \mathcal{C} \subseteq 2^{[|\rho_n|]}$ to a target structure sample $t \in \mathcal{T}$.
- **[S5] Bayesian inference** - We fix the universal object $\rho_n$. We consider the target distribution $\pi_{\mathcal{C}}(S) \propto P_{\text{likelihood}}(G \mid \text{U2T}(\rho_n, S)) \, \mathbb{I}[S \in \mathcal{C}]$. Algorithm 1 provides the MCMC inference.

---

**Implications for prior models** - Maybe some readers are confused as to what part of the "*prior model*" works in the SB framework. In fact, this was the part that was handled unconsciously in

the aforementioned TSP example. Therefore, we intend to clarify it again here. We can reveal the presence of a prior model by considering a direct target distribution $\hat{\pi}_{\mathcal{C}} : \mathcal{T} \to \mathbb{R}$ for a target structure sample $t \in \mathcal{T}$. Note that in **[S5]** above, we have indirectly captured the target distribution $\pi_{\mathcal{C}} : \mathcal{C} \to \mathbb{R}$ on the index subset $S \in \mathcal{C}$ of the universal target. Therefore, we can express the target distribution $\hat{\pi}_{\mathcal{C}}(t)$ on the target structure $t \in \mathcal{T}$ as $\hat{\pi}_{\mathcal{C}}(t) := \sum_{\{S | \text{U2T}(\rho_n, S) = t\}} \pi_{\mathcal{C}}(S)$. What this means is that the prior probability for the target structure $t$ is expressed as the extent to which the universal object $\rho_n$ contains a substructure $S \in \mathcal{C}$ for which $t = \text{U2T}(\rho_n, S)$. For the aforementioned TSP, we did not need to be very aware of the presence of this prior probability, due to the *uniformity*:

**Proposition 3.1** (Uniformity for TSP case). *For any $t \neq t' \in \mathcal{T}$, we have $\hat{\pi}_{\mathcal{C}}(t) = \hat{\pi}_{\mathcal{C}}(t')$.*

**Versatility of SB framework** - Some readers may also be concerned that our SB framework can only be applied to a limited number of applications, or that one would have to come up with a very clever universal object for each individual application. There is no need to worry. **Our SB framework can be applied affordably and systematically to most BML application cases**. The key reason is that the algebraic structure of "*permutations*" has surjective or bijective mappings to a variety of combinatorial objects that frequently appear in the typical BMLs: **Mixture model** - Appendix D.3 describes the classification task based on *partitions* (Ayyer, 2010; Corteel & Nadeau, 2009; Gregor et al., 2024) (used in Chinese restaurant process (Aldous, 1985)). **Factor model** - Appendix D.5 provides the factor analtsus task based on *binary sequences* (Pilaud & Pons, 2017) (used in Indian buffet process (Griffiths & Ghahramani, 2005)). **Tree models** - Appendix D.4 shows the hierarchical clustering task based on *binary trees* (Pilaud & Pons, 2017; Gregor et al., 2024) (used in Dirichlet diffusion tree (Neal, 2003)). **Ancestral graphs** - Appendix D.8 illustrates *phylogenetic analysis* based on *Cambrian trees* (Pilaud & Pons, 2017) (fragmentation-coagulation process (Teh et al., 2011)). **Relational models** - Appendix D.7 describes *relational data analysis* based on *k-d trees* (Bassino et al., 2018; Maazoun, 2019) (Mondrian process (Roy & Teh, 2009)) and *generic rectangulations* (Reading, 2012; Merino & Mütze, 2021) (block-breaking process (Nakano et al., 2020)).

# 4 SEVERAL APPLICATIONS THROUGH SUPER BAYESIAN FRAMEWORK

This section provides empirical demonstrations of our SB framework for typical BMP applications. We show that Algorithm 1 (referred to as **SB-c**) performs comparably to existing baselines, while Algorithm 2 (**SB-q**; quantum MCMC in Section 2.5 and Appendix C) often achieves slightly superior performance. **Common settings** - For all classical MCMCs, we run 10 independent chains from different initializations. For **SB-q**, we set the Suzuki-Trotter resolution to 10. Appendix E provides more details on ablation studies, hyperparameter robustness, and MCMC diagnostics.

— TASK 1: CONSENSUS RANKING (PERMUTATION) —

This is a task of inferring a single consensus ranking (that is, permutation) $\sigma \in \Lambda_n$, given multiple rating (ranking) information $G = \{\lambda_1, \lambda_2, \dots\}$ ($\lambda_i \in \Lambda_n$), where $n$ corresponds to the number of elements to be ranked in each dataset. **Data** - We use the following three datasets, Sushi preference rankings (Sushi) (Kamishima), College football rankings (CF), and College basketball rankings (CB) (Massey). Sushi is a manually annotated ranking of 5000 ratings for $n = 10$ different types of sushi. For CF, we use 2019-2021 expert ratings for top $n = 10$ teams. For CB, we use 2020-2021 expert ratings for top $n = 10$ teams. **Likelihood** $P_{\text{likelihood}}$ - We employ the Mallows model (Mallows, 1957) with Kendall $\tau$ distance (Fligner & Verducci, 1986): $P_{\text{likelihood}}(G \mid \sigma) \propto \exp(-\text{KENdist}(G, \sigma))$, where $\text{KENdist}(G, \sigma) := \sum_{\lambda_i \in G} |\{(l, m) : l < m, [\lambda_i(l) < \lambda_i(m) \land \sigma(l) > \sigma(m)] \land [\lambda_i(l) > \lambda_i(m) \land \sigma(l) < \sigma(m)]\}|$. **Evaluation criteria** - We held out 20% ratings for testing, and each model was trained using the remaining 80%. We evaluate the inference methods using the average Kendall $\tau$ distance (i.e., negative log prediction probability ignoring the partition function) for test ratings. Each inference method collects 100 samples: every 500 iterations after 15000 burn-in until 20000 iterations. **Methods for comparison** - We employ the two popular MCMC schemes as the baselines: the Metropolis-Hastings with removing, adding, and exchanging proposals (RAE) (Jerrum et al., 2004) and the Metropolis-Hastings with leaf-and-shift proposals (LS) (Vitelli et al., 2018). **Experimental results** - Table 2 shows the mean (std) of the Kendall $\tau$ distance (Note: Smaller distances mean better prediction performance).

Table 2: Kendall $\tau$ dist (mean (std))

|  | Sushi | CF | CB |
|---|---|---|---|
| RAE | 21.983 (2.149) | 21.330 (1.514) | 19.623 (1.871) |
| LS | 20.631 (2.018) | 21.353 (1.471) | 19.591 (1.854) |
| **SB-c** | 20.947 (2.216) | 21.328 (1.486) | 19.481 (1.681) |
| **SB-q** | 19.679 (1.935) | 21.142 (1.029) | 19.402 (1.420) |

— TASK 2: HIERARCHICAL CLUSTERING (BINARY TREE) —

This is a task of inferring hierarchical clustering (that is, binary tree $t$) based on the feature similarity of observation data $G = \{\mathbf{x}_1, \mathbf{x}_2, \dots\}$ ($\mathbf{x}_i$ corresponds to a feature vector). **Data** - We use the following two datasets, Arrhythmia (Guvenir & Quinlan, 1997) and Heart Disease (Janosi & Detrano, 1989). Arrhythmia consists of subjects with features extracted from electrocardiogram and class (normal and different arrhythmia types) labels. Heart Disease consists of subjects with features and class labels (4 levels of heart disease risks). **Likelihood** - We employ the probabilistic version of Dasgupta's objective (Dasgupta, 2016) with the Gaussian radial basis (RBF) function similarity: $P_{\text{likelihood}}(G \mid t) \propto \exp(-\text{DASdist}(G, t))$, where $\text{DASdist}(G, t) = \sum_{i \neq j \in |G|} \text{RBF}(\mathbf{x}_i, \mathbf{x}_j)|C(i,j)|$ and $C(i,j)$ denote the smallest cluster of a given binary tree $t$ that contains both $\mathbf{x}_i$ and $\mathbf{x}_j$. **Evaluation criteria** - We train hierarchical clustering of observation subjects using only the corresponding features, without using the class labels. Given the trained tree, we measure the fraction of test samples in that subtree which are in the same class. The expected value of this fraction is known as the *dendrogram purity*, and can be computed exactly in a bottom up recursion on the tree (Note: The purity is if and only if all subjects in each class are contained in some pure subtree). Each inference method collects 100 samples: every 500 iterations after 15000 burn-in until 20000 iterations. **Methods for comparison** - We employ the three baselines: the standard Bayesian hierarchical clustering (BHC) (Heller & Ghahramani, 2005) (i.e., Bayesian agglomerative merging operation), and the Pitman-Yor diffusion tree hierarchical clustering with the Metropolis-Hastings sampler (PYDT-MH) (Knowles & Ghahramani, 2014) and its slice sampler (PYDT-Slice) (Adams et al., 2010). **Experimental results** - Table 3 shows the mean (std) of the dendrogram purity.

Table 3: Purity (mean (std))

|  | Arrhythmia | Heart Dis. |
|---|---|---|
| BHC | 0.482 (0.015) | 0.762 (0.016) |
| PYDT-MH | 0.491 (0.019) | 0.771 (0.024) |
| PYDT-Slice | 0.488 (0.022) | 0.766 (0.020) |
| **SB-c** | 0.475 (0.019) | 0.770 (0.022) |
| **SB-q** | 0.502 (0.019) | 0.785 (0.018) |

— TASK 3: RELATIONAL DATA ANALYSIS (RECTANGULATION) —

This is a task of inferring rectangle clusters from the observation relational matrix $G = (x_{i,j})_{i=1,2,\dots, \ j=1,2,\dots}$. **Dataset** - We employ the four datasets (Leskovec et al., 2010): Wiki, Facebook, Twitter, and Epinions. Each data consists of the network adjacency matrix with binary interactions. **Likelihood** - We employ the marginal likelihood of the beta-Bernoulli conjugate models (Appendix D.7). **Evaluation criteria** - We held out $20\%$ elements of the input data for testing, and each model was trained using the remaining $80\%$ of the elements. We evaluate the inference methods using the *perplexity* ($\exp$ (negative average $\log$ prediction)) as a criterion. **Methods for comparison** - We employ three baselines: the Mondrian process (MP) (Roy & Teh, 2009) with PMCMC (Fan et al., 2018), the block-breaking process (BBP) (Nakano et al., 2020), and the permuton-induced Chinese restaurant process (PCRP) (Nakano et al., 2021). Each inference method collects 100 samples: every 50 iterations after 1500 burn-in until 2000 iterations. **Experimental results** - Table 4 shows the mean (std) of the perplexity.

Table 4: Perplexity (mean (std))

|  | Wiki | Facebook | Twitter | Epinions |
|---|---|---|---|---|
| MP | 1.2838 (0.0094) | 1.1944 (0.0217) | 1.2316 (0.0209) | 1.4098 (0.0064) |
| BBP | 1.2712 (0.0056) | 1.1818 (0.0197) | 1.2146 (0.0058) | 1.4006 (0.0044) |
| PCRP | 1.2583 (0.0041) | 1.1545 (0.0187) | 1.2057 (0.0092) | 1.3951 (0.0054) |
| **SB-c** | 1.2592 (0.0078) | 1.1621 (0.0094) | 1.2199 (0.0102) | 1.3965 (0.0091) |
| **SB-q** | 1.2575 (0.0051) | 1.1520 (0.0067) | 1.2031 (0.0043) | 1.3955 (0.0082) |

## 5 CONCLUSION, DISCUSSION, AND OPEN PROBLEMS

**Key takeaways** - We are finally steering in a new direction along with extremal combinatorics in Bayesian machine learning (BML). *All you need is to make the model extremally redundant.* This simple policy, most interestingly, has three significant benefits in BML: (1) model universality, (2) unified MCMC inference, and (3) a clue to the theoretical analysis of MCMC mixing times.

**Limitations** - Our SB framework has the potential to evolve alongside several intriguing topics. **(1) Difficulty of delicate control of prior models**: In our SB framework, it is not easy to introduce some complicated structural knowledge into a prior model, because we represent it indirectly via universal objects. Appendix D.2 provides a more detailed discussion of three potential approaches to this issue. **(2) Residual effect on MCMC mixing time for each individual problem** - As noted in Theorem 2.5, our SB framework can provide an upper bound on the MCMC mixing time for general problems, but still leaves the problem-specific term $\zeta_{\text{likelihood}}$ (i.e., derivation from modularity). Appendix B.4 and Remark D.3 provide two specific examples illustrating how to utilize this problem-dependent term. Developing a unified, problem-independent analysis of MCMC mixing time is likely an attractive near-term challenge to tackle alongside improvements in inference algorithms.

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

## A    BACKGROUND AND RELATED WORK

This paper represents a totally new perspective from existing work, importing extremal combinatorics findings for Bayesian machine learning (BML). However, it is inspired by a variety of findings from the surrounding fields. We will provide a comprehensive and detailed survey of previous research.

### A.1    BACKGROUND OVERVIEW

**Background 1: BML have walked hand in hand with "parsimony" and "combinatorics."** -

- **Parsimony**: The Bayesian nonparametric method (BNP), which achieved a breakthrough in BML since the 2000s, is based on the fundamental principle of "parsimony." The stochastic processes used as prior models in BNP have highly rich expressive capabilities due to their infinite-dimensional parameter space. It may seem impossible to infer such infinite-dimensional parameters directly on a computer. Therefore, even though the BNP model assumes an infinite number of parameters, the model is designed so that the parameters that actually contribute to the finite observation data behave as "parsimoniously" as possible. Fortunately, the "parsimony" provides a justification for inferring effectively usable finite-dimensional parameters from the infinite-dimensional space. As a result, BNP has established a new principle in machine learning (ML): the ability to precisely infer the stochastic process models with infinite expressive power using only finite computational resources.

- **Combinatorics**: Since the 2000s, the conventional BNP has been actively utilizing tools from "combinatorics" to establish a solid foundation for structural hypothesis inference in ML. For example, the classification task in machine learning are expressed as the inference on "partitions" in combinatorics, and their stochastic process representations are modeled as the Chinese restaurant processes Aldous (1981). The factor analysis task are expressed as the inference on "binary sequences" in combinatorics, and their stochastic processes are modeled as the Indian buffet processes Griffiths & Ghahramani (2005). As shown in Table 1, various representative examples of ML tasks are expressed as combinatorial structure inference, and by studying their stochastic process representations, BNP methods have been applied to a wide range of applications, including images, audio, music, video, natural language, and biological information processing.

**Background 2: Modern machine learning (especially deep learning) actively utilizes "redundancy."** - The fundamental philosophy of deep learning is that the redundancy of models (neural networks) leads to general-purpose, unified ML models. Deep learning provides a framework for addressing various tasks in ML by formulating them as regression problems using deep neural networks with highly redundant parameters, thereby offering a general and unified approach to solving them. Since 2012 (Krizhevsky et al., 2012), advancements in hardware have caught up with the requirements necessary to realize the redundancy inherent in the fundamental philosophy of deep learning, leading to the current notable developments in ML and AI, exemplified by deep learning. Traditionally, the contribution of redundancy has been interpreted as the universal approximation theorem (Hornik et al., 1989). This interpretation suggests that incorporating redundant networks contributes to universally representing functions for various regressions. However, at the same time, practical neural network training has often exhibited behavior that induces sparse networks. This behavior has recently been termed the lottery ticket hypothesis (Frankle & Carbin, 2019), providing a new perspective on trained deep learning networks. It suggests that sufficiently redundant random networks can represent the desired regression function as its "sub"structure.

**Background 3: Combinatorics has branched off into "extremal" combinatorics, which focuses on redundancy.** - A phenomenon analogous to the lottery ticket hypothesis in deep learning has actually been studied for over 100 years in combinatorics, forming a field known as extremal combinatorics. A central focus in extremal combinatorics, as exemplified by Ramsey theory (one of the greatest theories in combinatorics), is the question of "how much redundancy must be introduced into a structure of interest to guarantee that it possesses certain properties." This question directly mirrors the one posed in deep learning: "How much redundancy must be incorporated into a network to ensure that it functions as a regression for the target task?" In fact, some insightful structures called "universal objects" have been discovered in extremal combinatorics that realize this phenomenon. For example, redundant structures that contain all permutations as their substructures are called superpermutations (Engen & Vatter, 2021), those that contain all trees as their substructures are called supertrees (Defant et al., 2020), and those that contain all graphs as their substructures are

called supergraphs (Alon, 2020), and these have been extensively studied. For example, it is known that uniformly random permutations with redundancy of $\mathcal{O}(n^2 \log \log n)$ contain all permutations of length n as their subsequences (He & Kwan, 2019). Readers who are interested in learning more about extreme value combinations can refer to the great textbook (Jukna, 2011) and the excellent recent survey paper (Engen & Vatter, 2021).

**Genesis of our Super Bayesian framework inspired by these three inspirations.** - As can be seen from **[Background 2]**, modern machine learning has contributed to its explosive spread by providing "redundancy," which brings about its versatility, unification, and universality. Considering this in light of **[Background 1]**, it can be expected that Bayesian machine learning will also bring about dramatic developments by introducing "redundancy" instead of "parsimony." Fortunately, as shown in **[Background 3]**, the combinatorics field, which has been a companion to BML, has already achieved integration with redundancy through extremal combinatorics. Therefore, we believe that importing tools from extremal combinatorics is fundamentally important to introduce "redundancy" into BML. These inspirations form the genesis of our proposed super Bayesian framework.

## A.2 Related work

**Previous breakthrough in BML history: Nonparametric Bayes** - One of the key breakthroughs in the history of BML may have been the Bayesian nonparametric framework (BNP). Until the 1990s, the standard practice in traditional BML was to utilize "finite"-dimensional parametric models. Since around 2000, the idea of using stochastic processes (i.e., parametric models with infinite-dimensional parameter space) has begun to spread widely, making Bayesian inference for models with infinite expressive power possible in principle, including the Dirichlet process (Ferguson, 1973; Antoniak, 1974) and the completely random measures (Kingman, 1967). This trend evolved rapidly, and in the 2000s, most traditional finite-dimensional parametric models achieved their infinite extensions.[5] The importance of the BNP framework can be illustrated as follows. For example, we consider a traditional statistical mixture model (SMM) with a finite number of clusters for a clustering/classification task. If we set the upper bound on the number of clusters to 100, for example, this SMM would not be able to evaluate any hypothesis that would explain the observed data with more than 100 clusters. On the other hand, the BNP extension of SMM (i.e., the infinite mixtures (Rasmussen, 1999; Walker, 2007; Crane, 2012; Hughes & Sudderth, 2013; Sato et al., 2013; Matza & Bistritz, 2021; Bourouis et al., 2021)) then considers a potentially infinite number of clusters, allowing it to consider hypotheses with any number of clusters.

**Lottery ticket hypothesis (LTH)** (Frankle & Carbin, 2019; Ramanujan et al., 2020; Malach et al., 2020; Frankle et al., 2020; Diffenderfer & Kailkhura, 2021; Raj & Mishra, 2020; Chen et al., 2020; 2021b; Brix et al., 2020; Chen et al., 2021a; Girish et al., 2021) - The key idea of our SB framework is to introduce a deliberately *redundant* representation for the model. This idea is reminiscent of the *redundancy* of neural networks in deep learning, described as LTH:

- **Model universality** - The theory of LTH indicates that, very roughly speaking, a sufficiently large random neural network is guaranteed to contain some desired NN (called a *winning ticket*) in its subnet (See Ramanujan et al. (2020) and Theorem 2.1 in Malach et al. (2020) for a more precise statement). This means that a sufficiently redundant NN, properly trained, has the universality to approximate the desired regression function (Ferbach et al., 2023; Natale et al., 2024; Sakamoto & Sato, 2022).

---

[5]E.g., the infinite factor model (Griffiths & Ghahramani, 2005; Teh et al., 2007; Thibaux & Jordan, 2007; Lee et al., 2022; Xuan et al., 2018; Tan et al., 2018; Doshi-Velez & Williamson, 2017; Dallaire et al., 2014; Ozdemir & Davis, 2014; Reed & Ghahramani, 2013; Zhai et al., 2012; Rai & III, 2011; Austerweil & Griffiths, 2010; Williamson et al., 2010; Teh & Görür, 2009; Doshi et al., 2009; Miller et al., 2008), the infinite HMM (Gael et al., 2008b;a; Wen et al., 2021; Song et al., 2019; Poorjam et al., 2019; Dawson et al., 2017; Valera et al., 2016; Zhang et al., 2014), the infinite factorial HMM (Gael et al., 2008b; Valera et al., 2016; 2015), the $\infty$-gram (Mochihashi & Sumita, 2007; Wood et al., 2009; 2011), the infinite PCFG (Liang et al., 2007), the infinite (dynamic) Bayesian network (Doshi et al., 2011; Xu et al., 2022; Chen et al., 2022; Kessler et al., 2021), the Pólya tree (Nieto-Barajas, 2024; Zhuang et al., 2023; 2022; Nieto-Barajas & Núñez-Antonio, 2021; III & Hanson, 2017), the infinite relational model (Kemp et al., 2006; Ishiguro et al., 2012; Schmidt & Albers, 2015; Jiang & Zhang, 2015; Ishiguro et al., 2016; Saad & Mansinghka, 2021), the Mondrian process (Roy & Teh, 2009; Roy, 2011), the Mondrian forest (Khannouz & Glatard, 2024; Mourtada et al., 2017; Lakshminarayanan et al., 2016; 2014), and the infinite context graph Markov model (Castellana et al., 2022), to name a few.

- **Effective inference** - LTH also provides insight into the phenomenon of neural network learning that works well empirically (Chen et al., 2021a; 2025; Brix et al., 2020). The suitable neural networks that are symmetric and redundantly parameterized may have a multiplicity of desired winning tickets, making it easier for typical algorithms such as stochastic gradient methods to find any one of the winning tickets.

This theory gives us a very neat picture of the circumstances under which a redundant model can work well in machine learning. More interestingly, beyond machine learning, such phenomena are often observed in the field of combinatorics, in particular, *extremal combinatorics*. For example, a redundant sequence of numbers, called a *superpermutation* (Engen & Vatter, 2021), contains as its subsequences all short permutations. As another example, a redundant tree, called a *supertree* (Defant et al., 2020), contains as its subtrees all small trees. These specially designed redundant objects are called *universal objects*, which, in LTH parlance, can be regarded as always having a winning ticket somewhere in themselves. From the above considerations, Bayesian modeling with universal objects is expected to yield new developments in the field of machine learning. In a high-level philosophical sense, our SB framework can be seen as a reimportation of this LTH philosophy to BML, thanks to extremal combinatorics, which is the analogy of the LTH phenomena in combinatorics.

**Probabilistic models on set families** (Anari et al., 2016; Li et al., 2016b; Mariet et al., 2018; Song et al., 2024; Grosse et al., 2024; Barthelmé et al., 2023; Han et al., 2022b;a; Hemmi et al., 2022; Derezinski, 2019; Rezaei & Gharan, 2019; Derezinski et al., 2019; Li et al., 2016a) - Probabilistic models on set families, such as *strong Rayleigh measures* and *determinantal point processes*, have been studied in depth and theoretically transparent Markov chain Monte Carlo (MCMC) inference algorithms have become manifest. One important property of these models is *negative correlation* (or its generalization, *negative association*) (Li et al., 2016b).

> **Definition A.1** (Negative correlation). A probability distribution $\mu : 2^{[n]} \to \mathbb{R}_+$ is (pair-wise) *negatively correlated* if for any pair of elements $i, j \in [n]$,
>
> $$\mathbb{P}_{S \sim \mu}[i \in S] \cdot \mathbb{P}_{S \sim \mu}[j \in S] \geq \mathbb{P}_{S \in \mu}[i, j \in S], \qquad (6)$$
>
> where $\mathbb{P}[\cdot]$ indicates the probability of events.

This property plays an important role in the mixing time analysis of typical MCMC algorithms. More specifically, we can estimate the lower bound on the probability that the overall probability of the entire system improves when one element is replaced from the target set in the context of MCMC (e.g., Gibbs sampling (Li et al., 2016b) and Metropolis-Hastings method (Anari et al., 2016; Mariet et al., 2018)), and as a result, we can estimate an upper bound on the mixing time of MCMC. For more details, we refer to Section 2.2 and Theorem 9 of (Anari et al., 2016). Our SB framework can benefit from these existing theories by setting up a redundant universal object and constructing a target model as a probabilistic model on the substructure family of the universal object. More specifically, we achieve a theoretical analysis of MCMC mixing times by translating the SB framework into the environment given in Theorem 4 of (Li et al., 2016b).

**Extremal combinatorics** - To the best of our knowledge, there seems to be little active use of extremal combinatorics (principles and phenomena) in the field of machine learning. Indeed, as we have discussed in the "**Lottery ticket hypothesis**" item, LTH in deep learning may be viewed as an extremal combinatorics-like phenomenon in the broad sense, but there seems to have been little discussion of the conceptual relationship between the two yet. This paper is the first to make substantial use of extremal combinatorics in machine learning, particularly in Bayesian machine learning. We will depict in the next subsection the basic findings on *superpermutations*, which are particularly relevant and play an important role in this paper.

## A.3 SUPERPERMUTATIONS

In this paper, we have mentioned two superpermutations (the redundant partition concatenations and the zigzag words), and we will present two other popular and interesting examples.

**Permutation matrix** - A *permutation matrix* is a square binary matrix with exactly one entry of 1 in each row and column and all other entries are 0s. For a permutation $\sigma \in \Lambda_n$, we can obtain the corresponding permutation matrix $\boldsymbol{\Psi}(\sigma) = (\Psi_{i,j}(\sigma))_{i,j \in [n]}$ as $\Psi_{i,j}(\sigma) = 1$ if $j = \sigma(i)$; $\Psi_{i,j}(\sigma) = 0$

otherwise. For example, for $\sigma = 2143$, we have

$$\mathbf{\Psi}(\sigma = 2134) = \begin{pmatrix} 0 & 1 & 0 & 0 \\ 1 & 0 & 0 & 0 \\ 0 & 0 & 1 & 0 \\ 0 & 0 & 0 & 1 \end{pmatrix}. \tag{7}$$

**Pattern containment** - We introduce the following *pattern containment* notion. We say a (larger) binary matrix $\mathbf{M}$ *contains* a (smaller)binary matrix $\mathbf{M}'$ if we can delete rows and columns of $\mathbf{M}$ and change 1s to 0s of the remaining $M$ elements to recover the matrix $\mathbf{M}'$. For example, the following matrix $M$ *contains* a matrix $\mathbf{\Psi}(2134)$:

$$M = \begin{pmatrix} 0 & 0 & 0 & 1 & 0 & 0 & 0 \\ 1 & 0 & 0 & 0 & 0 & 0 & 1 \\ 0 & 1 & 0 & 1 & 0 & 1 & 0 \\ 0 & 0 & 0 & 0 & 0 & 0 & 0 \\ 0 & 0 & 0 & 1 & 0 & 0 & 0 \\ 1 & 0 & 1 & 0 & 1 & 0 & 0 \\ 0 & 0 & 0 & 0 & 0 & 0 & 1 \end{pmatrix}, \tag{8}$$

since we can delete some rows and columns (shaded in gray in the first transformation) of $\mathbf{M}$, and change the $1$ element of the remaining elements (shaded in red in the second transformation) to 0:

$$\begin{pmatrix} 0 & 0 & 0 & 1 & 0 & 0 & 0 \\ 1 & 0 & 0 & 0 & 0 & 0 & 1 \\ 0 & 1 & 0 & 1 & 0 & 1 & 0 \\ 0 & 0 & 0 & 0 & 0 & 0 & 0 \\ 0 & 0 & 0 & 1 & 0 & 0 & 0 \\ 1 & 0 & 1 & 0 & 1 & 0 & 0 \\ 0 & 0 & 0 & 0 & 0 & 0 & 1 \end{pmatrix} \rightsquigarrow \begin{pmatrix} 0 & 1 & 0 & 0 \\ 1 & 1 & 0 & 0 \\ 0 & 0 & 1 & 0 \\ 0 & 0 & 0 & 1 \end{pmatrix} \rightsquigarrow \begin{pmatrix} 0 & 1 & 0 & 0 \\ 1 & 0 & 0 & 0 \\ 0 & 0 & 1 & 0 \\ 0 & 0 & 0 & 1 \end{pmatrix} = \mathbf{\Psi}(2134). \tag{9}$$

**Lemma A.2** (Lemma 2.2 in (He & Kwan, 2019))**.** *Let $\mathbf{B} = (B_{i,j})_{i \in [q], j \in [\lfloor r/(2q) \rfloor]}$ be a $q \times \lfloor r/(2q) \rfloor$ binary matrix whose element $B_{i,j}$ is independently drawn from $\mathrm{Bernoulli}(1/2)$, that is, $B_{i,j} \sim_{\text{i.i.d.}} \mathrm{Uniform}(\{0,1\})$. Let $\omega_r$ be a uniformly random permutation of $[r]$. Then we can couple the two events, (1) the random permutation $\omega_r$ contains a subsequence order-isomorphic to some permutation $\sigma$, and (2) the random matrix $\mathbf{B}$ contains the corresponding permutation matrix $\mathbf{\Psi}(\sigma)$. That is, for any permutation $\sigma$, whenever the random matrix $\mathbf{B}$ contains the permutation matrix $\mathbf{\Psi}(\sigma)$, then the permutation $\omega_r$ contains the permutation $\sigma$.*

Thanks to this lemma, we can transform the observation of containment in binary matrices into an assertion of containment in permutations (allowing for order isomorphisms).

**Proposition A.3** (Theorem 1.2 in (He & Kwan, 2019))**.** *We consider a uniformly random permutation $\omega_r$ of $[r]$, where $r \geq 2000n^2 \log \log n$. That is, $\omega \sim \mathrm{Uniform}(\Lambda_r)$. This random permutation $\omega_r$ contains subsequences order-isomorphic to every permutation of length $n$, with probability $1 - o(1)$.*

**Deterministic superpermutation with alphabet** $[n]$ - Another important example the following *Radomirović's superpermutation.*

**Theorem A.4** (Corollary 10 in (Radomirović, 2012-11-22) and Theorem 8 in (Engen & Vatter, 2021))**.** *For any natural number $n \geq 7$, there is a sequence of length $\lceil n^2 - 7n/3 + 19/3 \rceil$ whose alphabet is restricted to $[n]$ that contains subsequences equal to every permutation of $[n]$, where $\lceil \cdot \rceil$ is the ceiling function.*

# B  PROOFS OMITTED IN MAIN TEXT

We will provide proofs for several theorems and propositions discussed in Section 2. In the main text, to distinguish between observed data and models, we denote the TSP graph corresponding to observed data as $G_{\text{obs}}$. Here, however, we will simply denote it as $G$ to avoid cumbersome notation.

## B.1 Proof of Proposition B.1: Universality of random permutation concatenation

> **Proposition B.1** (Universality). *RPC $\rho_n$ contains subsequences equal to every permutation of $[n]$.*

*Proof of Proposition B.1.* We will verify that, for any RPC sample $\rho_n$, any permutation $\sigma$ of length $n$ can be explicitly extracted as a subsequence of $\rho_n$.

**Sketch** - We recall that each RPC we are now interested in is a concatenation of $n$ (random) permutations of $[n]$. From this, for any permutation $\sigma \in \Lambda_n$, the element corresponding to each $\sigma(j)$ can be chosen one by one from the $j$-th (random) *sub*-permutation as follows:

$$\underbrace{\rho_n(1)\dots\rho_n(n)}_{\sigma(1)} \underbrace{\rho_n(n+1)\dots\rho_n(2n)}_{\sigma(2)} \dots \underbrace{\rho_n((n-1)n+1)\dots\rho_n(n^2)}_{\sigma(n)}. \tag{10}$$

**Details** - Without loss of generality, we can suppose

$$\rho_n = \rho_n(1)\rho_n(2)\rho_n(3)\dots\rho_n(n^2-1)\rho_n(n^2). \tag{11}$$

By construction, for any $i \in [n]$, we have

$$\rho_n\left((i-1)n+1\right)\rho_n\left((i-1)n+2\right)\dots\rho_n\left((i-1)n+n\right) \in \Lambda_n, \tag{12}$$

where we recall that $\Lambda_n$ indicates the set of all permutations of $[n]$. This means that, for any $k \in [n]$,

$$k \in \left\{ \rho_n\left((i-1)n+1\right), \rho_n\left((i-1)n+2\right), \dots, \rho_n\left((i-1)n+n\right) \right\}. \tag{13}$$

Therefore, for any permutation $\sigma = \sigma(1)\sigma(2)\dots\sigma(n) \in \Lambda_n$, we can immediately check that

$$\sigma(j) \in \left\{ \rho_n\left((j-1)n+1\right), \rho_n\left((j-1)n+2\right), \dots, \rho_n\left((j-1)n+n\right) \right\}, \tag{14}$$

for every $j = 1, 2, \dots, n$. This implies that, for every $j = 1, 2, \dots, n$, there exists $s_j \in [n^2]$ such that (1) $(j-1)n+1 \le s_j \le (j-1)n+n$, which immediately leads to $s_j < s_{j'}$ for $j < j'$. We can obtain $S = \{s_1, s_2, \dots, s_n\}$ such that (1) $s_1 < s_2 < \dots < s_n$ and (2) $\rho(s_j) = \sigma(j)$. Finally, we can extract a subsequence $\rho_n(s_1)\rho_n(s_2)\dots\rho_n(s_n) = \sigma$ from $\rho_n$. This completes the proof. □

## B.2 Proof of Proposition B.2: Uniformity of random permutation concatenation

> **Proposition B.2** (Uniformity for TSP case). *For any $t \ne t' \in \mathcal{T}$, we have $\hat{\pi}_{\mathcal{C}}(t) = \hat{\pi}_{\mathcal{C}}(t')$.*

*Proof of Proposition B.2.* We will figure out how to enumerate how many times a given permutation $\sigma \in \Lambda_n$ appears in the all possible RPC samples, and we will make sure that, for any permutation $\sigma \in \Lambda_n$, they are equal in number.

**Sketch** - We first divide some target permutation $\sigma$ into $\sigma(1)\dots\sigma(u_1)$, $\sigma(u_1 + 1)\dots\sigma(u_2)$, $\sigma(u_{n-1}+1)\dots\sigma(u_n)$, where $1 \le u_1 < u_2 < \dots < u_n = n$. Then, Here we focus on how many times the $i$-th segment $\sigma(u_{i-1}+1)\dots\sigma(u_i)$ appears in the $i$-th concatenated segment $\rho_n((i-1)n+1)\dots\rho_n(in)$ of $\rho_n$:

$$\underbrace{\rho_n(1)\dots\rho_n(n)}_{\sigma(1)\dots\sigma(u_1)} \underbrace{\rho_n(n+1)\dots\rho_n(2n)}_{\sigma(u_1+1)\dots\sigma(u_2)} \dots \underbrace{\rho_n((n-1)n+1)\dots\rho_n(n^2)}_{\sigma(u_{n-1}+1)\dots\sigma(u_n)}. \tag{15}$$

Since each segment $\rho_n((i-1)n+1)\dots\rho_n(in)$ is uniformly at random, that is, $\rho_n((i-1)n+1)\dots\rho_n(in) \sim \text{Uniform}(\Lambda_n)$, the number of times $\sigma(u_{i-1}+1)\dots\sigma(u_i)$ appears does not depend on the permutation $\sigma$ we are focusing on. The only remaining operation required for enumeration is to cover every pattern of how the target permutation $\sigma$ is divided into $\sigma(1)\dots\sigma(u_1)$, $\sigma(u_1 + 1)\dots\sigma(u_2)$, $\sigma(u_{n-1}+1)\dots\sigma(u_n)$. Needless to say, this operation is independent of the permutation of interest, since it is determined simply by the overall length of the sequence. From the above, we can see that any permutation $\sigma \in \Lambda_n$ has an equal number of appearances in all possible RPC samples.

**Details** - As in the main text, we use the notation $S = \{s_1 < \cdots < s_n\} \in \mathcal{C}$ (elements in ascending order) to indicate a set in the partition matroid $\mathcal{C}$. Similarly, we also use $S' = \{s'_1 < \cdots < s'_n\} \in \mathcal{C}$. It suffices to show that we have

$$\mathbb{E}_{\rho_n}|\{S \mid \rho_n(s_1)\ldots\rho_n(s_n) = \sigma\}| = \mathbb{E}_{\rho_n}|\{S' \mid \rho_n(s'_1)\ldots\rho_n(s'_n) = \sigma'\}| \tag{16}$$

for any $\sigma \neq \sigma' \in \Lambda_n$. We denote $g(y, z)$ as a function that outputs 1 if a number sequence $y$ contains a number sequence $z$ as a subsequence (ignoring the order isomorphism) and 0 otherwise. For example, $g(y = \mathbf{45221}, z = \mathbf{51}) = 1$ and $g(y = \mathbf{342115}, z = \mathbf{12}) = 0$. Let $\Omega_n$ be the set of partitions of the sequence $\mathbf{1234}\ldots\mathbf{(n-2)(n-1)n}$ with the order of its elements preserved. For example, $\{\mathbf{1}, \mathbf{2}\}, \{\mathbf{3}, \mathbf{4}, \mathbf{5}\}, \{\mathbf{6}\} \in \Omega_{\mathbf{6}}$ and $\{\mathbf{5}, \mathbf{1}\}, \{\mathbf{6}, \mathbf{2}\}, \{\mathbf{3}, \mathbf{4}\} \notin \Omega_{\mathbf{6}}$. For any $\sigma, \sigma' \in \Lambda_n$ and any $\{1, \ldots, u_1\}, \{u_1 + 1, \ldots, u_2\}, \ldots, \{u_{n-1} + 1, \ldots, u_n\} \in \Omega_n$ (Note some block may be a empty set), we have

$$\prod_{i=1}^{n} \sum_{\rho_n((i-1)n+1)\ldots\rho_n(in)\in\Lambda_n} g\Big(\rho_n((i-1)n+1)\ldots\rho_n(in), \sigma(u_{i-1}+1)\ldots\sigma(u_i)\Big)$$

$$= \prod_{i=1}^{n} \sum_{\rho_n((i-1)n+1)\ldots\rho_n(in)\in\Lambda_n} g\Big(\rho_n((i-1)n+1)\ldots\rho_n(in), \sigma'(u_{i-1}+1)\ldots\sigma'(u_i)\Big) \tag{17}$$

We obtain Equation (16) by summing over all possible $\{1, \ldots, u_1\}, \{u_1 + 1, \ldots, u_2\}, \ldots, \{u_{n-1} + 1, \ldots, u_n\} \in \Omega_n$ for both sides of the above equation. This completes the proof. $\square$

### B.3 Proof of Proposition B.3: Equivalence between standard Bayesian TSP (Section 2.1) and super Bayesian reformulation (Section 2.2)

**Proposition B.3** (Equivalence to Bayesian TSP). *We consider the original Bayesian TSP $P_{\text{posterior}}$ given in Section 2.1 (along with a uniform prior $P_{\text{prior}}(\sigma \in \Lambda_n) = 1/n!$). For any $\sigma \in \Lambda_n$, we have $P_{\text{posterior}}(\sigma \mid G) = \sum_{\rho_n \in \Xi_n} \sum_{\{S|\rho_n(s_1)\ldots\rho_n(s_n)=\sigma\}} P_{\mathcal{C}}(S \mid \rho_n) \cdot P_{\text{RPC}}(\rho_n).$*

*Proof of Proposition B.3.* We use the symbol $P_{\text{posterior}}^{(\text{sb})}$ to distinguish the posterior probabilities induced by SB reformulation. It follows from Equation (4) that, for any $S \in [|\rho_n|]$, we have

$$P_{\mathcal{C}}(S \mid \rho_n) \propto P_{\text{likelihood}}\Big(G \mid \rho_n(s_1)\ldots\rho_n(s_n)\Big) \cdot \mathbb{I}[S \in \mathcal{C}]$$

$$\Rightarrow P_{\mathcal{C}}(S \mid \rho_n) \propto P_{\text{likelihood}}\Big(G \mid \rho_n, S\Big) \cdot \mathbb{I}[S \in \mathcal{C}]$$

$$\Rightarrow P_{\mathcal{C}}(S \mid \rho_n) P_{\text{RPC}}(\rho_n) \propto P_{\text{likelihood}}\Big(G \mid \rho_n, S\Big) P_{\text{RPC}}(\rho_n) \cdot \mathbb{I}[S \in \mathcal{C}]. \tag{18}$$

Then, owing to Bayes' rule, we have

$$P_{\text{posterior}}^{(\text{sb})}(\sigma \mid G) = \sum_{\rho_n \in \Xi_n} \sum_{\{S|\rho_n(s_1)\ldots\rho_n(s_n)=\sigma\}} P_{\text{posterior}}^{(\text{sb})}\Big(\rho_n(s_1)\ldots\rho_n(s_n) \mid G\Big)$$

$$= \sum_{\rho_n \in \Xi_n} \sum_{\{S|\rho_n(s_1)\ldots\rho_n(s_n)=\sigma\}} P_{\text{posterior}}^{(\text{sb})}\Big(\rho_n, S \mid G\Big)$$

$$= \sum_{\rho_n \in \Xi_n} \sum_{\{S|\rho_n(s_1)\ldots\rho_n(s_n)=\sigma\}} P_{\mathcal{C}}\Big(S \mid \rho_n\Big) P_{\text{RPC}}(\rho_n). \tag{19}$$

Next, we recall that, if $S \in 2^{[|\rho_n|]}$ holds $\rho_n(s_1)\ldots\rho_n(s_n) = \sigma$ (i.e., the substructure $S$ of the superpermutation $\rho_n$ leads to a permutation of $[n]$), then $S \in \mathcal{C}$, since $S$ satisfies the partition matroid constraints of $\mathcal{C}$. Therefore, it follows from Equation (18) that we have

$$P_{\text{posterior}}^{(\text{sb})}(\sigma \mid G) = \sum_{\rho_n \in \Xi_n} \sum_{\{S|\rho_n(s_1)\ldots\rho_n(s_n)=\sigma\}} P_{\mathcal{C}}\Big(S \mid \rho_n\Big) P_{\text{RPC}}(\rho_n)$$

$$\propto \sum_{\rho_n \in \Xi_n} \sum_{\{S|\rho_n(s_1)\ldots\rho_n(s_n)=\sigma\}} P_{\text{likelihood}}\Big(G \mid \rho_n, S\Big) P_{\text{RPC}}(\rho_n)$$

$$\propto P_{\text{likelihood}}\Big(G \mid \sigma\Big) \sum_{\rho_n \Xi_n} \sum_{S \in \mathcal{C}} \mathbb{I}\Big[\rho_n(s_1)\ldots\rho_n(s_n) = \sigma\Big]. \tag{20}$$

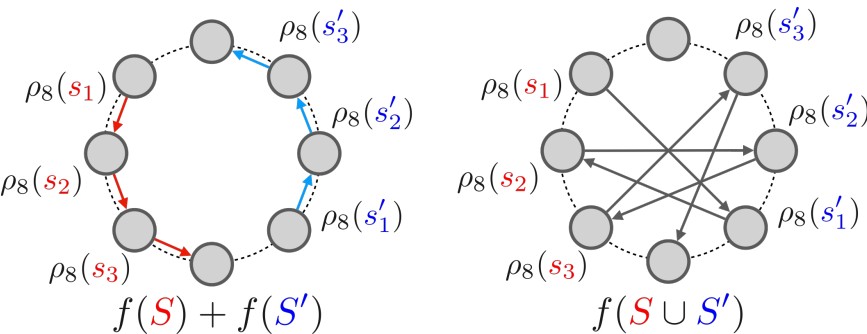

Figure 3: Illustrative example of $\zeta_{\text{likelihood}}(G) = \mathcal{O}(n)$ case. We consider a situation where cities are equally aligned on a two-dimensional circumference. We consider a route as $S$, which is a counterclockwise circle from one city, and $S'$, which is a counterclockwise circle from the city on the other side of the circumference. Then we can immediately check that $f(S) + f(S')$ is upper bounded by twice the length of the circumference. On the other hand, $f(S \cup S') = \mathcal{O}(n)$.

Owing to the uniformity (Proposition 3.1), for $\sigma, \sigma' \in \Lambda_n$, we have

$$\sum_{\rho_n \Xi_n} \sum_{S \in \mathcal{C}} \mathbb{I}\Big[\rho_n(s_1) \dots \rho_n(s_n) = \sigma\Big] = \sum_{\rho_n \Xi_n} \sum_{S \in \mathcal{C}} \mathbb{I}\Big[\rho_n(s_1) \dots \rho_n(s_n) = \sigma'\Big]. \tag{21}$$

Therefore we have

$$P_{\text{posterior}}^{(\text{sb})}(\sigma \mid G) \propto P_{\text{likelihood}}(G \mid \sigma). \tag{22}$$

We also recall that, as discussed in Section 2.1, we have

$$P_{\text{posterior}}(\sigma \mid G) \propto P_{\text{prior}}(\sigma) \cdot P_{\text{likelihood}}(G \mid \sigma). \tag{23}$$

Therefore, if we choose the uniform prior $P_{\text{prior}}(\sigma) = 1/n!$, then we have

$$P_{\text{posterior}}(\sigma \mid G) \propto P_{\text{likelihood}}(G \mid \sigma). \tag{24}$$

Finally, using the usual requirement of probability notions, we have

$$P_{\text{posterior}}^{(\text{sb})}(\sigma \mid G) = P_{\text{posterior}}(\sigma \mid G). \tag{25}$$

This completes the proof. $\qquad\square$

### B.4 Proof of Theorem B.4: Upper bound of MCMC mixing time of Algorithm 1

**Theorem B.4** (Upper bound). *For super Bayesian TSP (described in Section 2.2) with the cities topology $G = (V, D)$ (Section 2.1), Algorithm 1 has the following upper bound of its mixing time:*

$$\tau_{S_0}(\epsilon) \le 4n^2 \max_i |\mathcal{P}_i| \exp\left(2\zeta_{\text{likelihood}}(G)\right) \left(\log \pi_{\mathcal{C}}(S_0)^{-1} + \log \epsilon^{-1}\right). \tag{26}$$

*where, using a temporary abbreviation $f(S) = -\log P_{\text{likelihood}}\left(G \mid \rho_n(s_1)\rho_n(s_2) \dots \rho_n(s_n)\right)$, we have $\zeta_{\text{likelihood}}(G) = \max_{S, S' \in \mathcal{C}} |f(S) + f(S') - f(S \cap S') - f(S \cup S')|$.*

*Proof of Theorem B.4.* This is derived as a corollary to the special case of Theorem 4 (its second statement) in (Li et al., 2016b), which provides the upper bound of a normal Gibbs exchange MCMC for probabilistic models on the partition matroid. Specifically, we can rewrite the function $\zeta_{\text{likelihood}}$ in $\beta \zeta_F$ in (Li et al., 2016b), which can be attributed to the existing Theorem 4 in (Li et al., 2016b). This completes the proof. $\qquad\square$

**Additional note 1: Clarification of our contribution for this statement** - Our contribution to this theorem lies in *our nontrivial exploitation* itself. In fact, the technical basis of this claim leverages the excellent results of existing MCMC mixing time analysis. However, it is important to emphasize

that the way in which this is utilized is highly non-trivial. We have succeeded in introducing a form in Theorem 4 in (Li et al., 2016b) can be exploited by Bayesian modeling as a sub-structure of the universal objects from extremal combinatorics perspective.

**Additional note 2: Special case of** $\zeta_{\text{likelihood}}$ - we see that the application-specific influence appears in $\zeta_{\text{likelihood}}(G)$. This can be interpreted as the degree of deviation from *modularity*. For example, in the case of the TSP above, if we were miraculously able to evaluate $\zeta_{\text{likelihood}}(G) = \mathcal{O}(\log(\text{poly}(n)))$, our method might have been a polynomial-time solution to the TSP, since the entire right-hand side of Equation (26) would be upper bounded by a polynomial in $n$. However, of course, such a ground-breaking result is not available. Unfortunately, as we show in Figure 3, we can constructively make an example where $\zeta_{\text{likelihood}}(G) = \mathcal{O}(n)$. What we want to emphasize here is that through the set function $\zeta_{\text{likelihood}}$, Equation (26) can feed back the efficiency of inference to the design of the likelihood model $P_{\text{likelihood}}$.

## C  QUANTUM EXTENSION OF SUPER BAYESIAN FRAMEWORK

Our super Bayes (SB) promises high affinity with quantum computing. The reason is that it can be easily introduced *quantum superposition effects* on random substructures from universal objects. This affinity is of crucial importance in BML, because current Bayesian nonparametric (BNP) methods have technical difficulties with quantum extensions. It has been pointed out that quantum effects generally break down the *projectivity* of stochastic processes (see Section 5 of (Milz & Modi, 2021)), which are the heart of the BNP methods. In contrast, our SB naturally avoids this issue.

### C.1  PRELIMINARIES: QUANTUM BAYES WITH DENSITY MATRIX

The purpose of this subsection is to clarify the fundamental guideline for extending the Bayesian framework to a quantum version using the *density matrix* notion (Warmuth, 2005; Warmuth & Kuzmin, 2006). For a more general tutorial on quantum Bayesian and quantum stochastic processes, see, for example, (Milz & Modi, 2021). Here, we will explain a self-contained quantum Bayesian framework that uses classical probability theory and simple linear algebra, without using quantum physics or wave functions.

**Classical Bayes** - We will begin with an example that expresses classical statistics using a density matrix. As a useful example that is also utilized in the quantum extension of our super Bayesian framework , we will consider a system that assigns $K$ class labels with indices $1, ..., K$ to the target variables $x_1, \ldots, x_N$. Let $z_k$ be the latent class label to which $x_k$ belong. For example, if $N = 2$ and $K = 2$, the state of this system can be distinguished into the following four cases: $(z_1, z_2) = (1, 1)$, $(z_1, z_2) = (1, 2)$, $(z_1, z_2) = (2, 1)$, and $(z_1, z_2) = (2, 2)$. We will formally express these states using *binary indicator vectors*. We introduce a $K$-dimensional binary indicator vector $|\tilde{\sigma}_i\rangle$ for the $i$-th data $x_i$ $(i = 1, \ldots, N)$, and suppose that the $z_i$-th element of $|\tilde{\sigma}_i\rangle$ is 1 and all other elements are 0. For example, if $K = 2$ and $z_i = 2$, then we have

$$|\tilde{\sigma}_i\rangle = \begin{bmatrix} 0 \\ 1 \end{bmatrix}. \tag{27}$$

The class assignment of all target variables is also represented by an $K^N$-dimensional binary indicator vector $|\sigma\rangle := \bigotimes_{i=1}^{N} |\tilde{\sigma}_i\rangle$, where $\otimes$ is the Kronecker product[6]. Intuitively, $|\sigma\rangle$ serves to identify all states with a very long one-hot vector. Let $|\sigma^{(j)}\rangle$ be the $j$-th state of $|\sigma\rangle$, that is, the $j$-th element of $|\sigma^{(j)}\rangle$ is 1 and all other elements are 0. For the case of $K = 2$ and $N = 2$, the possible four states can be distinguished by

$$\left|\sigma^{(1)}\right\rangle = \begin{bmatrix} 1 \\ 0 \\ 0 \\ 0 \end{bmatrix}, \quad \left|\sigma^{(2)}\right\rangle = \begin{bmatrix} 0 \\ 1 \\ 0 \\ 0 \end{bmatrix}, \quad \left|\sigma^{(3)}\right\rangle = \begin{bmatrix} 0 \\ 0 \\ 1 \\ 0 \end{bmatrix}, \quad \left|\sigma^{(4)}\right\rangle = \begin{bmatrix} 0 \\ 0 \\ 0 \\ 1 \end{bmatrix}. \tag{28}$$

Then, we can freely design the prior model (e.g. Dirichlet distribution) and the likelihood model (e.g. categorical distribution) to set the posterior probability $p_j$ for the $j$-the state $(j = 1, \ldots, K^N)$.

---

[6] $A \otimes B = \begin{pmatrix} a_{11}B & a_{12}B \\ a_{21}B & a_{11}B \end{pmatrix}$, where $A = \begin{pmatrix} a_{11} & a_{12} \\ a_{21} & a_{11} \end{pmatrix}$.

Let $\mathcal{H}_c := \text{diag}\{-\log p_1, \ldots, -\log p_{K^N}\}$ be a $K^N$-by-$K^N$ diagonal matrix. Intuitively, $\mathcal{H}_c$ can be regarded as the *Hamiltonian* $\mathcal{H} = \mathcal{H}_c$ that represents the system's state energy. Finally, the classical posterior probability of the system can be expressed as $\langle \sigma | \frac{e^{-\mathcal{H}}}{\text{Tr}(e^{-\mathcal{H}})} | \sigma \rangle$, where $\langle A | = |A\rangle^\top$, and $e^A$ is the matrix exponential, i.e., $e^A = \sum_{i=0}^{\infty} \frac{1}{i!} A^i$. Note here that $\text{Tr}\left(e^{-\mathcal{H}_c}\right) = 1$. For example case of $K = 2$ and $N = 2$, we can calculate the posterior probability of the state $|\sigma^{(3)}\rangle$ as

$$\left\langle \sigma^{(3)} \left| e^{-\mathcal{H}_c} \right| \sigma^{(3)} \right\rangle = \begin{bmatrix} 0 & 0 & 1 & 0 \end{bmatrix} \begin{bmatrix} p_1 & 0 & 0 & 0 \\ 0 & p_2 & 0 & 0 \\ 0 & 0 & p_3 & 0 \\ 0 & 0 & 0 & p_4 \end{bmatrix} \begin{bmatrix} 0 \\ 0 \\ 1 \\ 0 \end{bmatrix} = p_3. \tag{29}$$

**Quantum Bayes** - The quantum extension of the classical Bayes described above can be achieved by (1) relaxing the one-hot vector $|\sigma\rangle$ to a *superposition* state and (2) introducing non-diagonal terms into the Hamiltonian. First, we remove the restriction that the state $|\sigma\rangle$ used in classical Bayes is a one-hot vector, and simply make it a unit real[7] vector that satisfies $\text{Tr}(\langle \sigma | \sigma \rangle) = 1$. As a result, for example, we can handle a superposition state

$$|\sigma\rangle = \begin{bmatrix} \sqrt{3}/2 \\ 0 \\ 1/2 \\ 0 \end{bmatrix} \tag{30}$$

where the first and third clusterings, $(z_1, z_2) = (1, 1)$ and $(z_1, z_2) = (2, 1)$, coexist (needless to say, note that this is not a statistical mixture, but coexistence). Next, we will introduce the non-diagonal terms $\mathcal{H}_q$ into the Hamiltonian $\mathcal{H}$ that represents the system's state energy, and set $\mathcal{H} = \mathcal{H}_c + \mathcal{H}_q$. We can freely choose the design of this non-diagonal term $\mathcal{H}_q$ according to the target system we hope to realize or the ease of handling the implementation. Some readers may be concerned that the calculation of the matrix exponential $e^{-\mathcal{H}} = \sum_{i=0}^{\infty} \frac{(-1)^i}{i!} (\mathcal{H}_c + \mathcal{H}_q)^i$ becomes intractable when the Hamiltonian $\mathcal{H}$ is not diagonal. This problem is a typical issue in quantum informatics, and it is standard to solve it approximately using the *Suzuki-Trotter decomposition* (Suzuki, 1976; Trotter, 1959):

$$e^{A+B} = \lim_{M \to 0} \left( e^{\frac{A}{M}} e^{\frac{B}{M}} \right)^M, \tag{31}$$

where $A$ and $B$ are represent some noncommutative operators. Namely, it is typical to set the non-diagonal terms $\mathcal{H}_q$ of the Hamiltonian $\mathcal{H} = \mathcal{H}_c + \mathcal{H}_q$ in a way that makes this form $(e^{\frac{\mathcal{H}_c}{m}} e^{\frac{\mathcal{H}_q}{m}})^m$ tractable.

### C.2 QUANTUM EXTENSION OF SUPER BAYES

Our idea for quantum expansion is very simple. Specifically, in the classical SB, we selected one element from each component $\mathcal{P}_i$ of the partition matroid $\mathcal{C}$, but in the quantum SB, we replace this with a state in which all elements have the *superposition*. Here, for simplicity, we use the TSP described in Section 2.1 as a concrete example, but the same framework can be applied to other applications with only minor modifications.

**Density matrix representation of substructure** $S \in \mathcal{C}$ - For convenience, we write $\mathcal{P}_i = \{\mathcal{P}_i(1) < \cdots < \mathcal{P}_i(n)\}$ to explicitly access each element $\mathcal{P}_i(\cdot)$ of $\mathcal{P}_i$. We consider a partition matroid sample $S \in \mathcal{C}$. Without loss of generality, we can suppose that $S = \{s'_1, \ldots, s'_n\}$ and $s'_i = \mathcal{P}_i(v_i)$. That is, we can suppose that, for each $\mathcal{P}_i$, the substructure index set $S$ chooses the $v_i$-th element of $\mathcal{P}_i$. For each $i$ ($i \in [n]$), we introduce $n$-dimensional binary indicator vector $|\tilde{\sigma}_i\rangle$, and suppose that the $v_i$-th element of $|\tilde{\sigma}_i\rangle$ is 1 and all other elements are 0. Then, each classical state of the super Bayesian TSP is represented by an $n^2$-dimensional binary indicator vector $|\sigma\rangle := \bigotimes_{i=1}^{n^2} |\tilde{\sigma}_i\rangle$. Therefore, we can equivalently treat $S$ as $|\sigma\rangle$, and define the classical Hamiltonian as follows:

$$\langle \sigma | e^{-\mathcal{H}_c(G)} | \sigma \rangle := P_\mathcal{C}(|\sigma\rangle \mid \rho_n) = P_\mathcal{C}(S \mid \rho_n) \tag{32}$$

---

[7]In principle, complex numbers are also allowed, but for the sake of simplicity, this paper only deals with real numbers.

**Design of quantum effects** - The most promising design for $\mathcal{H}_q$ is the following form, which is easy to handle computationally (a simple form can be derived using the Suzuki-Trotter decomposition described in Section C.1), inspired by its use in quantum annealing (Sato et al., 2009; Kurihara et al., 2009; Sato et al., 2013): $\mathcal{H}_q := \sum_{i=1}^{n^n} \varphi_i$, where

$$
\varphi_i = \left( \bigotimes_{h=1}^{i-1} \mathbf{E}_n \right) \otimes \varphi_0 \otimes \left( \bigotimes_{h=i+1}^{n} \mathbf{E}_n \right), \quad \varphi_0 = \Gamma(\mathbf{E}_n - \mathbf{I}_n),
$$

where $\mathbf{E}_n$ is the $n$-by-$n$ identity matrix, $\mathbf{I}_n$ is the $n$-by-$n$ matrix of ones (whose all elements are 1), and $\Gamma > 0$ is the tunable parameter that controls the strength of the quantum effect. Accordingly, by combining it with the diagonal term $\mathcal{H}_c(G)$, we can obtain the quantum SB as a quantum system with the Hamiltonian $\mathcal{H} = \mathcal{H}_c + \mathcal{H}_q$. We can evaluate the posterior distribution of a quantum state $|\sigma\rangle$ as

$$
P_{\text{posterior}} \left( |\sigma\rangle \mid G, \rho_n \right) := \langle\sigma| \frac{e^{-\mathcal{H}(G)}}{\text{Tr}\left(e^{-\mathcal{H}(G)}\right)} |\sigma\rangle. \tag{33}
$$

**Tractable representation of quantum SB** - As we discussed in Section C.1, it is difficult to explicitly calculate $P_{\text{Post.}}$ for $\mathcal{H}(G)$ with non-diagonal terms, so we wish to obtain some expression for $P_{\text{Post.}}$ that is easy to handle on a (classical) computer. Owing to the Suzuki-Trotter decomposition (Section C.1), We can obtain the following representation as a chain of multiple classical MPs in which the adjacent ones have a dependency relationship:

> **Theorem C.1** (Classical approximation of quantum SB). *Let $\Pi$ be the set of $n^2$-dimensional one-hot vectors. We have $P_{\text{Posterior}} \left( |\sigma\rangle \mid G, \rho_n \right) = \lim_{M \to \infty} \Psi / \text{Tr}\left(e^{-\mathcal{H}(G)}\right)$, where*
>
> $$
> \Psi = \sum_{\pi_2 \in \Pi} \cdots \sum_{\pi_M \in \Pi} \left[ \left( \langle\sigma| e^{-\frac{\mathcal{H}_c(G)}{M}} |\sigma\rangle \cdot e^{f(\Gamma)g(|\sigma\rangle, |\pi_2\rangle)} \right) \right.
> $$
> $$
> \cdot \left( \langle\pi_M| e^{-\frac{\mathcal{H}_c(G)}{M}} |\pi_M\rangle \cdot e^{f(\Gamma)g(|\pi_M\rangle, |\sigma\rangle)} \right)
> $$
> $$
> \left. \prod_{m=2}^{M-1} \langle\pi_m| e^{-\frac{\mathcal{H}_c(G)}{M}} |\pi_m\rangle \cdot e^{f(\Gamma)g(|\pi_m\rangle, |\pi_{m+1}\rangle)} \right],
> $$
> $$
> f(\Gamma) = \log\left(1 + \frac{n^2}{e^{\frac{\Gamma(n^2)}{M}} - 1}\right), \text{ and } g(|\pi\rangle, |\pi'\rangle) = g(\bigotimes_i |\tilde{\pi}_i\rangle, \bigotimes_i |\tilde{\pi}'_i\rangle) = \sum_i \langle\tilde{\pi}_i| |\tilde{\pi}'_i\rangle.
> $$

**Intuitive interpretation** - The above representation means that the posterior distribution $P_{\text{Post.}}(|\sigma \mid G\rangle)$ of the quantum superposition state $|\sigma\rangle$ can be inferred via the collection of classical state *replicas* $\pi_2, \ldots, \pi_M$ (i.e., one-hot vectors). Each replica $|\pi_m\rangle$ has the term of $\langle\pi| e^{-\frac{\mathcal{H}_c(G)}{M}} |\pi\rangle$ as a classical system. It is noteworthy that replicas with adjacent indexes, $|\pi_m\rangle$ and $|\pi_{m+1}\rangle$, have a chain dependency on each other in the term of $e^{\langle\pi_m||\pi_{m+1}\rangle f(\Gamma)}$. This term is exactly what reflects quantum effects. To put it roughly, the more similar $|\pi_m\rangle$ and $|\pi_{m+1}\rangle$ are, the higher the probability of the quantum system as a whole. In summary, we can think of the quantum superposition state $|\sigma\rangle$ as being inferred via the marginalization $\sum_{\pi_2 \in \Pi} \cdots \sum_{\pi_M \in \Pi}$ of all possible patterns of the classical state replicas $\pi_2, \ldots, \pi_M$ (ideally, $M \to \infty$) with the dependencies of the chained quantum effects.

*Proof.* As discussed in Section C.1, we have

$$
P_{\text{posterior}} \left( |\sigma\rangle \mid G \right) = \langle\sigma| \frac{e^{-\mathcal{H}(G)}}{\text{Tr}\left(e^{-\mathcal{H}(G)}\right)} |\sigma\rangle \tag{34}
$$

$$
\propto \langle\sigma| e^{-\mathcal{H}(G)} |\sigma\rangle. \tag{35}
$$

Owing to the Suzuki-Trotter decomposition (Suzuki, 1976; Trotter, 1959), we obtain the following formulation:

$$
\langle\sigma| e^{-\mathcal{H}(G)} |\sigma\rangle = \lim_{M \to \infty} \langle\sigma| \left( e^{-\frac{\mathcal{H}_c(G)}{M}} e^{-\frac{\mathcal{H}_q}{M}} \right)^M |\sigma\rangle. \tag{36}
$$

The above formulation can be simplified by using the following technique: For some $A$, we have

$$\langle\sigma|\left(e^A e^A\right)|\sigma\rangle = \langle\sigma|e^A \mathbf{E}_n e^A|\sigma\rangle \tag{37}$$

$$= \langle\sigma|e^A\left(\sum_{\pi\in\Pi}|\pi\rangle\langle\pi|\right)e^A|\sigma\rangle \tag{38}$$

$$= \sum_{\pi\in\Pi}\langle\sigma|e^A|\pi\rangle\langle\pi|e^A|\sigma\rangle. \tag{39}$$

Using the above relationship $(2M-1)$ times repeatedly, we can obtain

$$\langle\sigma|\left(e^{-\frac{\mathcal{H}_{\mathrm{c}}(G)}{M}}e^{-\frac{\mathcal{H}_{\mathrm{q}}}{M}}\right)^M|\sigma\rangle \tag{40}$$

$$= \sum_{\pi_2\in\Pi}\sum_{\pi_2'\in\Pi}\cdots\sum_{\pi_M\in\Pi}\sum_{\pi_{M+1}'\in\Pi}\langle\sigma|e^{-\frac{\mathcal{H}_{\mathrm{c}}(G)}{M}}|\pi_2'\rangle\langle\pi_2'|e^{-\frac{\mathcal{H}_{\mathrm{q}}}{M}}|\pi_2\rangle$$

$$\cdots\langle\pi_M|e^{-\frac{\mathcal{H}_{\mathrm{c}}(G)}{M}}|\pi_{M+1}'\rangle\langle\pi_{M+1}'|e^{-\frac{\mathcal{H}_{\mathrm{q}}}{M}}|\sigma\rangle. \tag{41}$$

Henceforth we will consider the classical term $\langle\pi_m|e^{-\frac{\mathcal{H}_{\mathrm{c}}(G)}{M}}|\pi_{m+1}'\rangle$ and the quantum term $\langle\pi_m'|e^{-\frac{1}{M}\mathcal{H}_{\mathrm{q}}}|\pi_m\rangle$ separately. First, we can obtain the simpler expression for the classical term:

$$\langle\pi_m|e^{-\frac{\mathcal{H}_{\mathrm{c}}(G)}{M}}|\pi_{m+1}'\rangle = \langle\pi_m|e^{-\frac{\mathcal{H}_{\mathrm{c}}(G)}{M}}|\pi_m\rangle\,\mathbb{I}\left[|\pi_m\rangle=|\pi_{m+1}'\rangle\right], \tag{42}$$

where $\mathbb{I}[x=y]$ is the indicator function, that is, $\mathbb{I}[x=y]=1$ if $x=y$ and $\mathbb{I}[x=y]=0$ otherwise. Next, we consider the simpler expression for the quantum term. For notational convenience, we use $|\pi_m\rangle := \bigotimes_i|\tilde{\pi}_{m,i}\rangle$. Here we use the following two techniques: (1) For some $A, B, C, D$, we have $(A\otimes B)(C\otimes D)=(AC)\otimes(BD)$. (2) For some $A, B$ such that $AB=BA$, we have $e^{A+B}=e^A e^B$. Using these equations, we can obtain

$$\langle\pi_m'|e^{-\frac{1}{M}\mathcal{H}_{\mathrm{q}}}|\pi_m\rangle$$

$$= \langle\pi_m'|e^{-\frac{1}{M}\sum_{i=1}^N\rho_i}|\pi_m\rangle$$

$$= \langle\pi_j'|e^{-\frac{1}{M}\rho_1}e^{-\frac{1}{M}\varphi_2}\cdots e^{-\frac{1}{M}\rho_N}|\pi_m\rangle$$

$$= \left(\bigotimes_{i=1}^{n^2}\langle\tilde{\pi}_{m,i}'|\right)\left(\prod_{k=1}^K e^{-\frac{1}{M}\left(\otimes_{h=1}^{i-1}\mathbf{E}_n\right)\otimes\varphi_0\otimes\left(\otimes_{h=i+1}^K\mathbf{E}_n\right)}\right)\left(\bigotimes_{i=1}^{n^2}|\tilde{\pi}_{m,i}\rangle\right)$$

$$= \left(\bigotimes_{i=1}^{n^2}\langle\tilde{\pi}_{m,i}'|\right)\left(\prod_{k=1}^K\sum_{h=0}^\infty\frac{1}{h!}\left\{-\frac{1}{M}\left(\bigotimes_{h=1}^{i-1}\mathbf{E}_n\right)\otimes\varphi_o\otimes\left(\bigotimes_{h=i+1}^K\mathbf{E}_n\right)\right\}^h\right)\left(\bigotimes_{i=1}^{n^2}|\tilde{\pi}_{m,i}\rangle\right)$$

$$= \left(\bigotimes_{i=1}^{n^2}\langle\tilde{\pi}_{m,i}'|\right)\left(\prod_{k=1}^K\sum_{h=0}^\infty\frac{1}{h!}\left\{-\frac{1}{M}\left(\bigotimes_{h=1}^{i-1}\mathbf{E}_n\right)\otimes\varphi_0^h\otimes\left(\bigotimes_{h=i+1}^K\mathbf{E}_n\right)\right\}\right)\left(\bigotimes_{i=1}^{n^2}|\tilde{\pi}_{m,i}\rangle\right)$$

$$= \prod_{i=1}^{n^2}\left(\langle\tilde{\pi}_{m,i}'|e^{-\frac{\Gamma}{M}}|\tilde{\pi}_{m,i}\rangle + \frac{e^{-\frac{\Gamma(1-n^2)}{M}-\frac{\Gamma}{M}}}{n^2}\right). \tag{43}$$

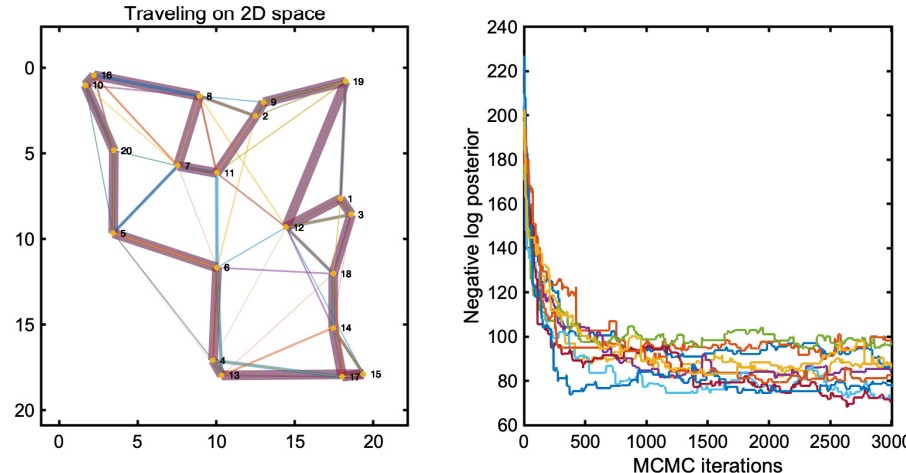

Figure 4: Demonstrative example of quantum MCMC (Algorithm 2) for super Bayesian TSP. We independently draw a location (indicated by $\mathbf{l}_i$) of each $i$-th city from $\mathrm{Uniform}([0, 20] \times [0, 20])$ and set $d_{\mathrm{TSP}}(i, j) = |\mathbf{l}_i - \mathbf{l}_j|_2$ (i.e., the Euclidean norm). We set the Suzuki-Trotter decomposition level $M = 10$. **Left**: Current superpositional state of $|\sigma\rangle$ (induced from $S_1, \ldots, S_M$). **Right**: MCMC evolution of negative log posterior probabilities for each component for Suzuki-Trotter decomposition. **Supplementary materials** also contain the reproducible code and an illustrative animation (.mp4) for MCMC evolution.

Owing to Equations (42) and (43), we can obtain

$$
\langle\sigma| \left( e^{-\frac{\mathcal{H}_c(G)}{M}} e^{-\frac{\mathcal{H}_q}{M}} \right)^M |\sigma\rangle = \sum_{\pi_2 \in \Pi} \cdots \sum_{\pi_M \in \Pi} \left[ \left( \langle\sigma| e^{-\frac{\mathcal{H}_c(G)}{M}} |\pi_2'\rangle \cdot e^{f(\Gamma)g(|\sigma\rangle, |\pi_2\rangle)} \right) \mathbb{I} \left[ |\sigma\rangle = |\pi_2'\rangle \right] \right.
$$

$$
\cdot \left( \langle\pi_M| e^{-\frac{\mathcal{H}_c(G)}{M}} |\pi_M'\rangle \cdot e^{f(\Gamma)g(|\pi_M\rangle, |\sigma\rangle)} \right) \mathbb{I} \left[ |\pi_M\rangle = |\pi_{M+1}'\rangle \right]
$$

$$
\left. \prod_{m=2}^{M-1} \langle\pi_m| e^{-\frac{\mathcal{H}_c(G)}{M}} |\pi_{m+1}'\rangle \cdot e^{f(\Gamma)g(|\pi_m\rangle, |\pi_{m+1}\rangle)} \mathbb{I} \left[ |\pi_m\rangle = |\pi_{m+1}'\rangle \right] \right]
$$

$$
= \sum_{\pi_2 \in \Pi} \cdots \sum_{\pi_M \in \Pi} \left[ \left( \langle\sigma| e^{-\frac{\mathcal{H}_c(G)}{M}} |\sigma\rangle \cdot e^{f(\Gamma)g(|\sigma\rangle, |\pi_2\rangle)} \right) \right.
$$

$$
\left. \cdot \left( \langle\pi_M| e^{-\frac{\mathcal{H}_c(G)}{M}} |\pi_M\rangle \cdot e^{f(\Gamma)g(|\pi_M\rangle, |\sigma\rangle)} \right) \prod_{m=2}^{M-1} \langle\pi_m| e^{-\frac{\mathcal{H}_c(G)}{M}} |\pi_m\rangle \cdot e^{f(\Gamma)g(|\pi_m\rangle, |\pi_{m+1}\rangle)} \right].
$$

$$\square$$

Owing to this representation, we can finally obtain the quantum extension of classical Markov chain Monte Carlo inference. For the $m$-th classical state $|\pi_m\rangle$ with the corresponding index set $S_m$, we can obtain the target distribution $\pi_{\mathcal{C}}(S_m)$ as follows

$$
\pi_{\mathcal{C}}^{(\mathrm{Q})}(S_m) = \langle\pi_m| e^{-\frac{\mathcal{H}_c(G)}{M}} |\pi_m\rangle \cdot e^{f(\Gamma)g(|\pi_{m-1}\rangle, |\pi_m\rangle)} \cdot e^{f(\Gamma)g(|\pi_m\rangle, |\pi_{m+1}\rangle)}, \tag{44}
$$

where (for notational convenience) $\pi_1 = \sigma$, $\pi_0 = \pi_M$, and $\pi_{M+1} = \sigma$. Algorithm 2 provides the whole procedure. Figure 4 shows an illustration of Algorithm 2 for the quantum extension of the super Bayesian TSP.

# D    DETAILS: SEVERAL APPLICATIONS OF SUPER BAYESIAN FRAMEWORK

## D.1    ADDITIONAL NOTES ON ALGORITHM 1

The computational complexity of Algorithm 1 has three aspects: (1) the time complexity of MCMC updates, (2) the space complexity of MCMC updates, and (3) the mixing time of MCMC (the number

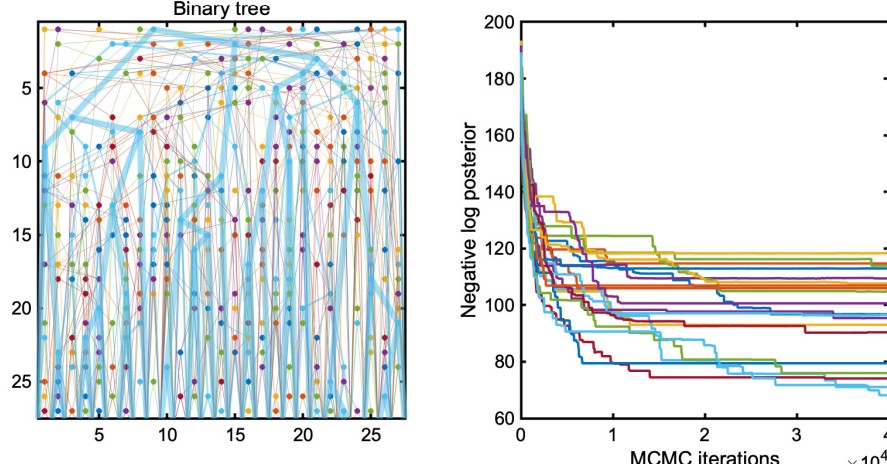

Figure 5: Demonstrative example of quantum MCMC (Algorithm 2) for super Bayesian binary tree model. We use **Arrhythmia** (Guvenir & Quinlan, 1997) data (Number of samples: 452, feature dimension: **279**, number of classes 16). The original task is to classify the presence or absence of cardiac arrhythmia using 279 features (e.g., *R wave width*, *QRS angle*, *Q-T interval*) extracted from electronic cardiogram. We simply extract 28 samples uniformly at random from the whole data. We use the probabilistic version of the Dasgupta objective (Dasgupta, 2016) for the likelihood model, and set the Suzuki-Trotter decomposition level $M = 20$. **Left**: Current superpositional state of $|\sigma\rangle$ (induced from $S_1, \ldots, S_M$). **Right**: MCMC evolution of negative log posterior probabilities for each component for Suzuki-Trotter decomposition. **Supplementary materials** also contain the reproducible code and an illustrative animation (`.mp4`) for MCMC evolution.

---

**Algorithm 2:** Quantum MCMC for super Bayesian models

**Input:** Partition matroid $\mathcal{C} = \cup_{i=1}^{n} \mathcal{P}_i$, universal object $\rho_n$ that induce target distribution $\pi_{\mathcal{C}}$, and Suzuki-Trotter decomposition level $M \in \mathbb{N}$

**Output:** MCMC samples of $S_m \in \mathcal{C}$ $(m = 1, \ldots, M)$ that approximate target distribution $\pi_{\mathcal{C}}$.

1 Initialize each $S_m \in \mathcal{C}$, e.g., $s_i \sim \mathrm{Uniform}(\mathcal{P}_i)$ $(i = 1, \ldots, n)$ and $S_m \leftarrow \{s_1\} \cup \cdots \cup \{s_n\}$ ;

2 **while** *MCMC is not mixing into the stationary distribution* **do**

3    **for** $m = 1$ *to* $M$ **do**

4       $b \sim \mathrm{Bernoulli}(1/2)$ ;    /* `Laziness (avoidance of periodicity)` */

5       **if** $b = 1$ **then**

6          $s \sim \mathrm{Uniform}(S_m)$ ;        /* `Selection of update location` */

7          $s' \sim \mathrm{Uniform}(\{k \mid \rho_n(k) = \rho_n(s) \wedge k \neq s\})$ ;   /* `Replacement element` */

8          $S_m \leftarrow (S_m \setminus \{s\}) \cup \{s'\}$ with probability $\dfrac{\pi_{\mathcal{C}}^{(Q)}\left((S_m \setminus \{s\}) \cup \{s'\}\right)}{\pi_{\mathcal{C}}^{(Q)}(S_m) + \pi_{\mathcal{C}}^{(Q)}((S_m \setminus \{s\}) \cup \{s'\})}$ ;

         /* `Gibbs exchange` */

---

of iterations required to asymptotically converge to the target distribution). Theorem 2.5, the main theoretical contribution of this paper, deals with (3) mixing time, which is the most non-trivial and difficult to handle of these three aspects. On the other hand, in relation to the above empirical reports, it is considered useful to provide simple supplementary information regarding (1) time complexity and (2) space complexity.

- **(1) Time complexity of MCMC updates** - We would like to emphasize that the redundancy used in our super Bayesian framework has no negative impact on time complexity. This may be a non-obvious claim, but it can certainly be achieved by using the hashing technique. For example, when targeting a permutation of length $n$, standard MCMC requires $\mathcal{O}(n)$ time complexity. On the other hand, our super Bayesian framework may be thought to require $\mathcal{O}(n^2)$ time complexity in principle because it uses a superpermutation of length $n^2$. However, by pre-assigning a real value in the range $[0, 1]$ to each element of the superpermutation of length $n^2$ to preserve the total order of the superpermutation elements, we can convert each subsequence of length $n$ from the

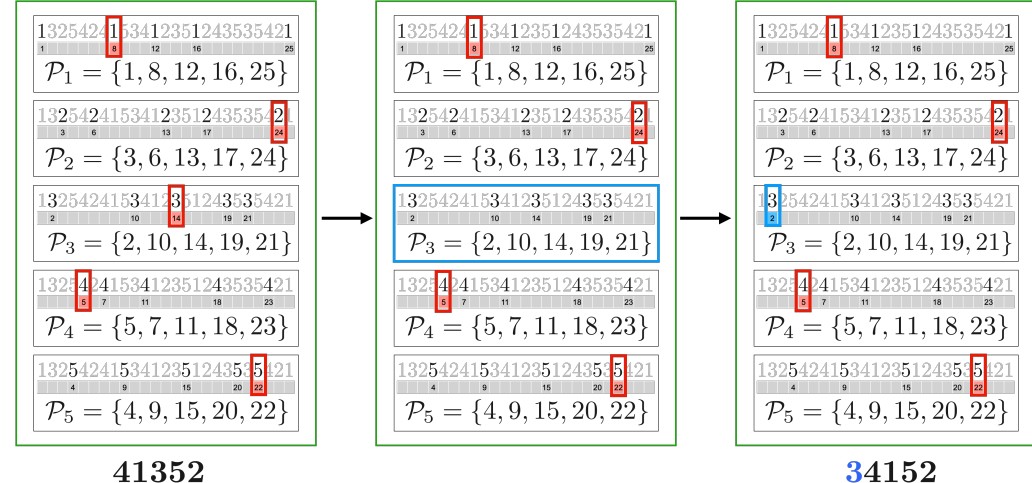

Figure 6: Illustration of Algorithm 1. **Left** - We suppose that the current MCMC state corresponds to $S = \{5, 8, 14, 22, 24\}$. This leads to a current permutation state $\rho_5(5)\rho_5(8)\rho_5(14)\rho_5(22)\rho_5(24) =$ **41352**. **Middle: Line** 5 **of Algorithm 1** - We select the element $8 \in \mathcal{P}_3$ uniformly at random from $S$ as a candidate for MCMC update. **Right: Lines** $6 - 7$ **of Algorithm 1** - We then reselect a new element from $\mathcal{P}_3$. Finally, we accept it with the Gibbs exchange scheme.

superpermutation of length $n^2$ into a "permutation" using only the corresponding hash values of size n. Therefore, although our Super Bayes method involves the superpermutation of length $n^2$, the computational complexity of its MCMC updates remains $\mathcal{O}(n)$.

- **(2) Space complexity of MCMC updates** - On the other hand, the redundancy used in the super Bayesian imposes an additional $\mathcal{O}(n)$ load on the space complexity. Specifically, for example, when the target structure is a permutation of length n, while standard MCMC requires $\mathcal{O}(n)$ space complexity, our super Bayesian framework requires $\mathcal{O}(n^2)$ space complexity. These facts mean that, as long as sufficient computer memory is available as hardware, our super Bayesian can be executed without sacrificing calculation speed compared to the standard Bayesian methods.

**Hyperparameters and auxiliary variables** - Our Algorithm 1 demonstrates the inference mechanism for the most important target structure (combinatorial structure). However, for more practical applications, we can also introduce updates for hyperparameters and auxiliary variables. More specifically, we only need to add update rules for hyperparameters and auxiliary variables within the main loop in **Lines 4-7**. However, we must note that with such additions, the upper bound on mixing time from Theorem 2.5 may cease to hold.

### D.2 TOWARD DESIGN OF INFORMATIVE PRIORS

This section discusses the prospects for constructing an informational prior model within the SB framework. Since the SB framework indirectly constructs generative probabilistic models of target combinatorial structures (e.g., permutations, partitions, binary sequences, etc.) via extremal combinatorial universal objects (e.g., superpermutations), introducing any structural prior information into the prior model is not straightforward, as discussed in Section 5. Overcoming this challenge is undoubtedly a crucial research topic for the next step in the SB framework. Interestingly, the BNP method — a major breakthrough in BML during the 2000s — faced a similar challenge in its early days. This stemmed from the fact that the stochastic processes used in the BNP method also required strong constraints like projectivity and exchangeability for their construction, making free model design difficult. While we intend to advance specific detailed methods in a subsequent series of studies, here we wish to discuss three fundamental future directions.

- **(1) Weighting toward the superpopulation** - When modeling natural science phenomena, it is often desirable to incorporate prior knowledge such as balance laws or power laws into the prior model. Examples include situations where one wishes to induce balanced partitions in hierarchical

$$\texttt{U2T} : \rho_5(3)\rho_5(15)\rho_5(22)\rho_5(29)\rho_5(48) \mapsto \{\mathbf{3}, \mathbf{1}\}, \{\mathbf{4}, \mathbf{5}\}, \{\mathbf{2}\}$$

$$\rho_5 = 11\underline{\mathbf{3}}32255442244\underline{\mathbf{1}}155334\underline{\mathbf{4}}11$$
$$2233\underline{\mathbf{5}}5112244335533554442\underline{\mathbf{2}}11$$

Figure 7: Illustration of super Bayesian mixture model.

clustering using binary trees, or to induce the number of elements in each cluster to follow the power law. Looking back at the history of the BNP method, a well-known successful example of designing such an informative prior model is the extension to the Pitman-Yor process (Pitman & Yor, 1997) to introduce the power law for an infinite mixture model using the Dirichlet process. Such an informative prior model could likely be achieved by weighting population frequencies in the universal objects of extremal combinatorics.

- **(2) Avoiding patterns in universal objects of extremal combinatorics** - Conversely, when modeling a phenomenon, there are often situations where we intentionally want to impose specific constraints on the prior model. For example, for relational data analysis based on rectangulation, a case in point is when we wish to restrict the rectangulation pattern to hierarchical rectangulations (represented by a $k$-d tree via the Mondrian process (Roy & Teh, 2009)). Such an informative prior model can be achieved by introducing pattern avoidance when designing extremal combinatorial universal objects. For instance, in the aforementioned scenario where we wish to restrict rectangular partitioning to hierarchical partitioning, one strategy is to constrain the superpermutation to a separable permutation (Maazoun, 2019; Bassino et al., 2018). This approach would be unique to the SB method and was not present in the BNP method.

- **(3) Incorporating data-dependent covariates** - Informative prior design has long been a challenge in BNP methods. Fortunately, a general approach using kernel methods gained widespread adoption in the 2010s (Williamson et al., 2010; Zhou et al., 2011; Wang et al., 2015; Blei & Frazier, 2011). This strategy avoids direct modifications to the difficult-to-handle infinite-dimensional stochastic process. Instead, it applies reweighting using similarity between observational data vectors and latent covariates to the target stochastic process at a later stage. This strategy provides informative prior design for various BNP models, as shown in the table below. For our SB method, applying this post-hoc, data-driven reweighting to the partition matroid also allows us to construct a data-dependent model.

### D.3 MIXTURE MODEL FOR CLUSTERING AND CLASSIFICATION

A mixture model is one that expresses that the entire data has a hidden cluster structure and that each piece of data belongs to one cluster. We will show its SB reformulation following **[S1]-[S5]** . Figure 7 shows an intuitive illustration.

**[S1]** - We are given the observation data $G = (V, X)$, where $V = [n]$ is an index set of the data, and $X = (\mathbf{x}_i)_{i \in V}$ with $\mathbf{x}_i = (x_{i,1}, x_{i,2}, \ldots, x_{i,K}) \in \mathbb{R}^K$. That is, $x_{i,k}$ indicates the $k$-th feature of the $i$-th observation data sample. **[S2]** - For clustering/classification, we can hypothesize the *partition* of $V = [n]$ as the latent *target structure* $t \in \mathcal{T}$. That is, $\mathcal{T}$ indicates the set of all possible partition of $[n]$. For example, for $n = 5$, we have a partition sample $t = \{3, 1\}, \{2, 4\}, \{5\} \in \mathcal{T}$. **[S3]** - Given the observation data $G$ and the target structure $t$, we can employ any likelihood model $P_{\text{likelihood}}(G \mid t)$. It would be standard to use a "marginal" likelihood that has been integrated over the auxiliary parameters (e.g., the mean and covariance of each cluster) using conjugate priors. **[S4: Key for SB]** - To achieve our SB reformulation, we need a universal object that can contain any partition of $V = [n]$. The key insight is that a partition can be represented by sequentially assigning elements of a permutation to blocks. From this, we come up with the idea of additionally granting a block assignment mechanism to the superpermutation. We first introduce the following tool:

**Definition D.1** (Double-length superpermutation). *Let $\rho'_n$ be a RPC sample defined as Definition 2.1. We set up another number sequence $\rho_n$ of length $2n^2$ such that $\rho_n(2k-1) \leftarrow \rho'_n(k)$ and $\rho_n(2k) \leftarrow \rho'_n(k)$. We call this sequence $\rho_n$ a* double-length superpermutation.

Next we set $\mathcal{P}_i = \{k \mid \rho_n(k) = i\}$ and let $\mathcal{C} = \cup_{i=1}^n \mathcal{P}_i$. Then we define $\text{U2T} : (\rho_n, S \in \mathcal{C}) \mapsto t \in \mathcal{T}$ as follows. For $S = \{s_1 < \cdots < s_n\}$, we consider sequentially partitioning the elements of the sequence $\rho_n(s_1) \ldots \rho_n(s_n)$ into blocks, starting from the beginning, and if $s_k$ is odd, we will leave it as it is, and if $s_k$ is even, we will assign $\rho_n(s_k)$ to a new block.[8] These satisfy the *universality*:

**Theorem D.2** (Universality). *Let $\rho_n$ be a double-length superpermutation (Definition D.1). Then, for any $t \in \mathcal{T}$, there exists $S \in \mathcal{C}$ such that $\text{U2T}(\rho_n, S) = t$. Therefore, we have $\text{supp}(\hat{\pi}_\mathcal{C}) = \mathcal{T}$.*

**[S5]** - With everything ready, we can use Algorithm 1 for Bayesian inference. This gives us a somewhat novel type of MCMC update rules, unlike the TSP case, where the block partitioning and cluster assignment are updated at the same time. As a final addition, we will reconsider the design of the likelihood model with respect to the MCMC mixing time, i.e., the term $\zeta_{\text{likelihood}}(G)$ in Theorem 2.5. For example, inspired by *topological data analysis*, we can consider the following setup:

**Remark D.3** (Polynomial mixing time). *Let $M_1, M_2 = \mathcal{O}(\log(\text{poly}(n)))$ be a natural number and a real number, respectively. We consider a collection of any desired* anchor sets *$\mathcal{Q}_1, \mathcal{Q}_2, \ldots, \mathcal{Q}_{M_1} \in \mathcal{C}$. For notational convenience, for $S \in \mathcal{C}$, without loss of generality, we suppose that a partition sample $\text{U2T}(\rho_n, S)$ consists of a collection of sets $R_1, \ldots, R_H \subseteq [n]$ such that $\cup_{h=1}^H R_h = [n]$ and $R_h \cap R_{h'} = \phi$ ($h \neq h' \subseteq [n]$). Next, we introduce a* score function *$f_\mathcal{Q}(S) := \sum_{h=1}^H \sum_{m=1}^{M_1} \min\{|R_h \cap \mathcal{Q}_m|, M_2\}$, and reconsider and set the likelihood $P_{\text{likelihood}}(G \mid \text{U2T}(\rho, S)) \propto \exp(-f_\mathcal{Q}(S))$. Then, Algorithm 1 has polynomial mixing w.r.t. $n$.*

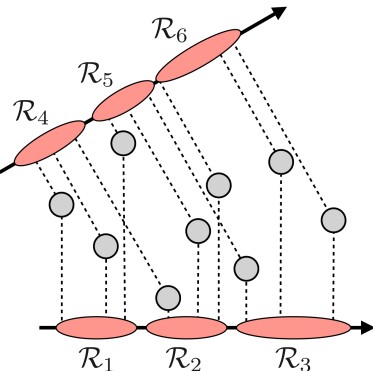

Figure 8: Illustration of topological mixture model.

*Proof of Remark D.3.* By construction, we have

$$f_\mathcal{Q}(S) = \sum_{h=1}^H \sum_{m=1}^{M_1} \min\{|R_h \cap \mathcal{Q}_m|, M_2\}$$

$$\leq \sum_{h=1}^H \sum_{m=1}^{M_1} M_2 \leq H M_1 M_2 = \mathcal{O}(\log(\text{poly}(n))). \tag{45}$$

Owing to Theorem 2.5, we can obtain the upperbound of the mixing time of Algorithm 1 as follows:

$$\tau_{S_0}(\epsilon) \leq 4n^2 \max_i |\mathcal{P}_i| \exp(2\zeta_{\text{likelihood}}(G)) \left(\log \pi_\mathcal{C}(S_0)^{-1} + \log \epsilon^{-1}\right)$$

$$\leq 4n^3 \text{poly}(n) \left(\log \pi_\mathcal{C}(S_0)^{-1} + \log \epsilon^{-1}\right). \tag{46}$$

This completes the proof. □

---

[8]For example, for $\rho_5 = 11\mathbf{3}3225544224411553344112233\mathbf{5}5112244335533554442\mathbf{2}11$ and $S = \{3, 15, 22, 29, 48\}$, we have $\text{U2T}(\rho_5, S) = \{\mathbf{3}, \mathbf{1}\}, \{\mathbf{4}, \mathbf{5}\}, \{\mathbf{2}\}$, since $s_3 = 29$ (even) and $s_5 = 48\}$ (even) generate two blocks for the sequence $\rho_5(s_1)\rho_5(s_2)\rho_5(s_3)\rho_5(s_4)\rho_5(s_3) \rightsquigarrow \mathbf{3}1;\mathbf{4}5;\mathbf{2}$.

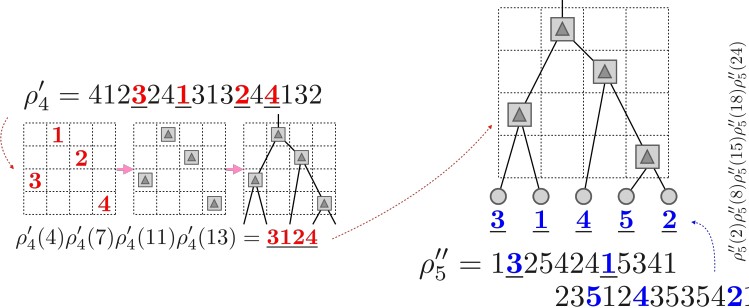

Figure 9: Illustration of super Bayesian binary tree model.

**Additional note: Topological mixture model** - The above likelihood form provides an important connection to "topological" mixture models (Huntsman, 2018) that are different from standard parametric mixture models. As shown in Figure 8, for example, we use multiple partitions as anchor sets $\mathcal{R}_\infty, \ldots, \mathcal{R}_H$, such that there are an equal number of data on the projection of the observed data in various directions. This likelihood can be evaluated nonparametrically by the degree of overlap with the anchor sets to evaluate the likelihood.

### D.4 BINARY TREE MODEL FOR HIERARCHICAL CLUSTERING

A binary tree model is a model that represents all data as a hidden bifurcated tree structure, with each piece of data belonging to a single terminal node. Figure 9 shows an intuitive illustration.

**[S1]** - The observation data $G = (V, X)$ has the same settings as in the above mixture model case. **[S2]** - For hierarchical clustering, we can hypothesize the *binary tree* whose terminal nodes consist of $V = [n]$ as the latent *target structure* $t \in \mathcal{T}$. **[S3]** - Given the observation data $G$ and the target structure $t$, we can employ any likelihood model $P_{\text{likelihood}}(G \mid t)$ (e.g., a probabilistic version of Dasgupta's objective (Dasgupta, 2016) or the Dirichlet diffusion tree (Neal, 2003)). **[S4: Key for SB]** - We need a universal object that can contain any binary tree $t \in \mathcal{T}$. The key insight here begins with the basic fact in combinatorics that *binary trees can be represented by two permutations* (Figure 1 **(c) top**) : (1) for encoding the tree structure and (2) for representing the order of the terminal nodes. That is,

> **Lemma D.4** (Surjective map from two permutations to binary tree). *There exists a surjective map* $\text{P2B} : \Lambda_{n-1} \times \Lambda_n \to \mathcal{T}$. *(Figure 1 **(c)** shows an illustration of* $\text{P2B}$. **Appendix** *provides a detail.)*

From this, we come up with the idea of having a pair of the two superpermutations (one for the tree structure and one for the order of the terminal nodes) as a universal object to represent binary trees.

> **Definition D.5** (Duo-superpermutations). Let $\rho'_{n-1}$ and $\rho''_n$ be a pair of two RPCs (Definition 2.1). We set up another object $\rho_n = (\rho'_{n-1}, \rho''_n)$. We call this object $\rho_n$ a *Duo-superpermutations*.

Next we set $\mathcal{P}_{i,j} := \{k \mid \rho_{n-1,1}(k) = i\} \times \{k' \mid \rho_{n,2}(k') = j\}$ and let $\mathcal{C} = \cup_{i=1}^{n-1} \cup_{j=1}^{n} \mathcal{P}_{i,j}$. We can uniquely obtain a binary tree $t \in \mathcal{T}$ from the universal object $\rho_n$ and its index subset $S = \{s_{1,1} < \cdots < s_{n-1,1}\} \times \{s_{1,2} < \ldots s_{n,2}\} \in \mathcal{C}$ by using $\rho_{n-1,1}(s_{1,1}) \ldots \rho_{n-1,1}(s_{n-1,1})$ as the tree structure and $\rho_{n,2}(s_{1,2}) \ldots \rho_{n,2}(s_{n,2})$ as the order of the terminal nodes. Then we have

> **Theorem D.6** (Universality). *Let* $\rho_n = (\rho'_{n-1}, \rho''_n)$ *be a duo-superpermutations (Definition D.1). Then, for any* $t \in \mathcal{T}$, *there exists* $S \in \mathcal{C}$ *such that* $\text{U2T}\left((\rho'_{n-1}, \rho''_n), S\right) :=$ $\text{P2B}\left(\rho'_{n-1}(s_{1,1}) \ldots \rho'_{n-1}(s_{n-1,1}), \rho''_n(s_{1,2}) \ldots \rho''_n(s_{n,2})\right) = t$. *Hence, we have* $\text{supp}(\hat{\pi}_C) = \mathcal{T}$.

**[S5]** - With everything ready, we can use Algorithm 1 for Bayesian inference. This leads to a nontrivial update rule that simultaneously updates the tree structure and the assignment of the terminal nodes.

### D.5 SUPER BAYESIAN FACTOR MODEL

We will show the SB reformulation following **[S1]-[S5]** described at the beginning of Section 3.

**[S1] Observation data** - We are given the observation data $G = (V, X)$, where $V = [n]$ is an index set of the data, and $X = (\mathbf{x}_i)_{i \in V}$ with $\mathbf{x}_i = (x_{i,1}, x_{i,2}, \ldots, x_{i,K}) \in \mathbb{R}^K$. That is, $x_{i,k}$ indicates the $k$-th feature of the $i$-th observation data sample.

**[S2] Target structure design** - For factor analysis, we can hypothesize the *collection of binary sequences* as the latent *target structure* $t \in \mathcal{T}$. Each data $\mathbf{x}_i$ has a binary sequence $\mathbf{z}_i = (z_{i,1}, z_{i,2}, \ldots, z_{i,K}) \in \{0, 1\}^K$, which indicates 1 if the $i$-th data is affected by the $k$-th factor and 0 otherwise. We can regard the collection of binary sequences (i.e., a binary matrix), $\mathbf{Z} = (z_{i,k})_{i \in [n], k \in [K]}$, as a target structure $t \in \mathcal{T}$.

**[S3] Likelihood design** - Given the observation data $G$ and the target structure $t$, we can employ any likelihood model $P_{\text{likelihood}}(G \mid t)$ for the usual Bayesian factor models.

**[S4] Introduction of universal object and partition matroid** - To achieve our SB reformulation of Bayesian factor models, we need a universal object that can contain any collection of binary sequences (in other words, any $n \times K$ binary matrix). The key insight is that permutations of length $K + 1$ have a surjective map to binary sequences of length $K$. More precisely, the following fact has been founded (See **Example** 4 (iv) in (Pilaud & Pons, 2017)):

> **Proposition D.7** (Surjection from permutations to binary sequences). *There exists a surjective map from permutations of length $K + 1$ to binary sequences of length $K$.*

We can explicitly construct a map P2BS to transform from a permutation $\sigma$ of length $(K + 1)$ to a binary sequence $\mathbf{z} = (z_1, z_2, \ldots, z_K)$ of length $K$: If $\sigma(i) < \sigma(i + 1)$ then $z_i = 0$; otherwise $z_i = 1$. For example, P2BS($\mathbf{265143}$) = $(0, 1, 1, 0, 1)$ and P2BS($\mathbf{68413275}$) = $(0, 1, 1, 0, 1, 0, 1)$. We then introduce the following tool:

> **Definition D.8** ($N$-superpermutations). Let $\rho_{K+1}^{(1)}, \rho_{K+1}^{(2)}, \ldots, \rho_{K+1}^{(n)}$ be a collection of $n$ RPCs (Definition 2.1). We set up another object $\rho_n = (\rho_{K+1}^{(1)}, \rho_{K+1}^{(2)}, \ldots, \rho_{K+1}^{(n)})$. We call this object $\rho_n$ a $N$-superpermutations.

Next we set $\mathcal{P}_{i_1, \ldots, i_n} := \{k^{(1)} \mid \rho_{K+1}^{(1)}(k^{(1)}) = i_1\} \times \{k^{(2)} \mid \rho_{K+1}^{(2)}(k^{(2)}) = i_2\} \times \cdots \times \{k^{(n)} \mid \rho_{K+1}^{(n)}(k^{(n)}) = i_n\}$ and let $\mathcal{C} = \cup_{i_1=1}^{n} \cup_{i_2=1}^{n} \cdots \cup_{i_n=1}^{n} \mathcal{P}_{i_1, i_2, \ldots, i_n}$. We can uniquely obtain a binary matrix $t = \mathbf{Z} \in \mathcal{T}$ from the universal object $\rho_n$ and its index subset $S = \{s_{1,1} < \cdots < s_{n,1}\} \times \{s_{1,2} < \ldots s_{n,2}\} \times \cdots \times \{s_{1,n} < \cdots < s_{n,n}\} \in \mathcal{C}$ by using each $\rho_{K+1,i}(s_{1,i}) \ldots \rho_{K+1,i}(s_{n,i})$ for the $i$-th binary sequence $\mathbf{z}_i = (z_{i,1}, \ldots, z_{i,K})$, that is, U2BS $(\rho_{K+1,i}(s_{1,i}) \ldots \rho_{K+1,i}(s_{n,i})) = \mathbf{z}_i$. Then we can immediately check the *universality*

> **Corollary D.9** (Universality). *Let $\rho_n$ be a $N$-superpermutations (Definition D.8). Then, for any $t \in \mathcal{T}$, there exists $S \in \mathcal{C}$ such that U2T $(\rho_n, S) :=$ P2B $(\rho_{K+1,i}(s_{1,i}) \ldots \rho_{K+1,i}(s_{n,i})) = t (= \mathbf{Z})$. Hence, we have $\text{supp}(\hat{\pi}_C) = \mathcal{T}$.*

*Proof of Corollary D.9.* For any sample $t \in \mathcal{T}$, since (1) the $N$-superpermutation $\rho_n$ contains all possible combinations of any permutations, and (2) the mapping U2BS is surjective, we can confirm that there exists $S \in \mathcal{C}$ such that U2T $(\rho_n, S) :=$ P2B $(\rho_{K+1,i}(s_{1,i}) \ldots \rho_{K+1,i}(s_{n,i})) = t$. This completes the proof. $\square$

**[S5] Bayesian inference** - With everything ready, we can use Algorithm 1 for Bayesian inference.

### D.6 SUPER BAYESIAN NETWORK

**[S1] Observation data** - We are given the observation data $G = (V, X)$, where $V = [n]$ is an index set of the data, and $X = (\mathbf{x}_i)_{i \in V}$ with $\mathbf{x}_i = (x_{i,1}, x_{i,2}, \ldots, x_{i,K}) \in \mathbb{R}^K$. That is, $x_{i,k}$ indicates the $k$-th feature of the $i$-th observation data sample.

**[S2] Target structure design** - For causality inference tasks, we can hypothesize the *directed acyclic graph* (DAG) as the latent *target structure* $t \in \mathcal{T}$.

**[S3] Likelihood design** - Given the observation data $G$ and the target structure $t$, the Bayesian network represent a factorization of the joint probability $P_{\text{likelihood}}(G \mid t)$ into the product of conditional

distributions:

$$P_{\text{likelihood}}(\mathbf{x}_1, \ldots, \mathbf{x}_n \mid t) = \prod_{i=1}^{n} P_{\text{localLikelihood}}(\mathbf{x}_i \mid \text{Parent}(\mathbf{x}_i)), \tag{47}$$

where $\text{Parent}(\mathbf{x}_i)$ represents all the parents of the $i$-th node $\mathbf{x}_i$ in DAG $t$, and we can freely use any desired local likelihood model $P_{\text{localLikelihood}}$. For example, we can use the linear functional model for the likelihood model $P_{\text{likelihood}}$ as

$$G = W_\gamma^\top \boldsymbol{X} + \boldsymbol{\epsilon}, \tag{48}$$

where $\boldsymbol{X} := [X_{1,1:N}^\top, \ldots, X_{n,1:N}^\top]$, $\boldsymbol{\epsilon}$ is a random noise such that $\mathbb{E}(\boldsymbol{\epsilon}) = \mathbf{0}$ and $\mathbb{V}(\boldsymbol{\epsilon}) = \text{diag}(\sigma_1^2, \ldots, \sigma_n^2)$ with hyper-parameters $\sigma_1, \ldots, \sigma_n > 0$, and $W_\gamma$ is a weights matrix such that

$$\begin{aligned}
(W_\gamma)_{i,j} \mid \gamma_{i,j} = 1 &\sim \text{Normal}(0, \sigma_j^2), \\
(W_\gamma)_{i,j} \mid \gamma_{i,j} = 0 &\sim \delta_0,
\end{aligned} \tag{49}$$

where $\delta_0$ is the Dirac function concentrated on 0.

**[S4] Introduction of universal object and partition matroid** - To achieve our SB reformulation of Bayesian network models, we need a universal object that can contain any DAG. The key insight is that permutations (also called *topological orders* in this context) have a surjective map to DAGs. We then introduce the following tool:

> **Definition D.10** (Superpermutation)**.** Let $\rho_n$ be RPC, as defined in Definition 2.1.

Next we set $\mathcal{P}_i = \{k \mid \rho_n(k) = i\}$ and let $\mathcal{C} = \cup_{i=1}^n \mathcal{P}_i$. Then we define $\texttt{P2BN} : (\rho_n, S \in \mathcal{C}) \mapsto t \in \mathcal{T}$ as follows. We write $S = \{s_1 < \ldots, s_n\}$ to indicate a subsequence $\rho_n(s_1) \ldots \rho_n(s_n)$. If $\rho_n(s_i) < \rho_n(s_j)$, then $\mathbf{x}_{s_j} \in \text{Parent}(\mathbf{x}_{s_i})$; otherwise $\mathbf{x}_{s_j} \notin \text{Parent}(\mathbf{x}_{s_i})$. We note that $\text{Parent}(\cdot)$ uniquely characterizes a DAG $t$. Then we can immediately check the *universality*

> **Corollary D.11** (Universality)**.** *Let $\rho_n$ be a superpermutation (Definition D.10). Then, for any $t \in \mathcal{T}$, there exists $S \in \mathcal{C}$ such that $\texttt{U2T}(\rho_n, S) = t$. Hence, we have $\text{supp}(\hat{\pi}_C) = \mathcal{T}$.*

*Proof of Corollary D.11.* For any sample $t \in \mathcal{T}$, since (1) RPC $\rho_n$ contains all possible permutations, and (2) the mapping $\texttt{U2BN}$ is surjective, we can confirm that there exists $S \in \mathcal{C}$ such that $\texttt{U2T}(\rho_n, S) := \texttt{P2BN}(\rho_n(s_1) \ldots \rho_n(s_n)) = t$. This completes the proof. $\qquad \square$

**[S5] Bayesian inference** - With everything ready, we can use Algorithm 1 for Bayesian inference. For the auxiliary variables $W_\gamma$, we can infer it concurrently within the main loop using Gibbs sampling.

### D.7 SUPER BAYESIAN RELATIONAL MODEL

We will show the SB reformulation following **[S1]-[S5]** described at the beginning of Section 3.

**[S1] Observation data** - We are given the observation data $G = (V, X)$, where $V = [n]$ is an index set of the data, and $X = (x_{i,j})_{i,j \in V}$. For simplicity, we suppose that $x_{i,j} \in \mathbb{R}$.

**[S2] Target structure design** - For relational data analysis, we can hypothesize the *rectangulation* as the latent *target structure* $t \in \mathcal{T}$. We can suppose that each element $\mathbf{x}_{i,j}$ belongs to one cluster of the rectangulation sample $t$. For simplicity, we suppose that $t$ is a rectangulation on $[0, 1] \times [0, 1]$.

**[S3] Likelihood design** - Given the observation data $G$ and the target structure $t$, we can employ any likelihood model $P_{\text{likelihood}}(G \mid t)$ for the usual Bayesian relational models (e.g., the Aldous-Hoover-Kallenberg representation (Aldous, 1981; Hoover, 1979; Kallenberg, 1992). Specifically, we introduce $\boldsymbol{U}^{(\text{row})} := \{U_1^{(\text{row})}, \ldots, U_{|V|}^{(\text{row})}\}$ and $\boldsymbol{U}^{(\text{column})} := \{U_1^{(\text{column})}, \ldots, U_{|V|}^{(\text{column})}\}$, and use the beta-Bernoulli hierarchical model:

$$\begin{aligned}
U_i^{(\text{row})} &\sim \text{Uniform}([0, 1]) \quad (i = 1, \ldots, N), \\
U_j^{(\text{column})} &\sim \text{Uniform}([0, 1]) \quad (j = 1, \ldots, M), \\
\vartheta_k &\underset{\text{i.i.d.}}{\sim} \text{Beta}(\beta_0) \quad (k = 1, \ldots), \\
x_{i,j} &\underset{\text{i.i.d.}}{\sim} \text{Bernoulli}\left(\vartheta_{k(U_j^{(\text{column})}, U_i^{(\text{row})})}\right),
\end{aligned}$$

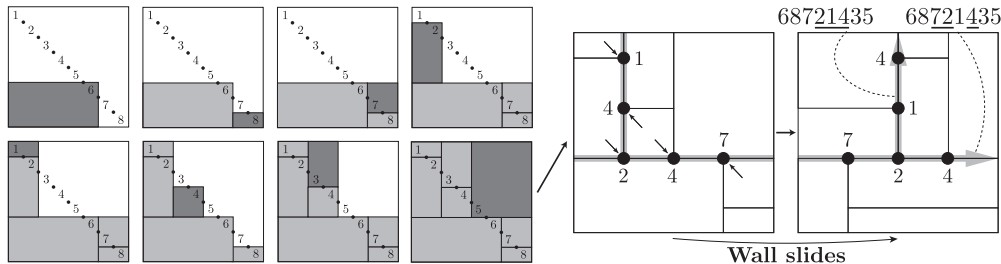

Figure 10: Mapping P2R from permutation (e.g., 68721435) of length $n \in \mathcal{S}$ to rectangulation with $n$ blocks $\in \mathcal{R}$. **Left**: We first draw $n + 1 = 9$ distinct *diagonal points* on the diagonal, with one of the points being the top-left corner and another being the bottom-right corner. Let $T$ be the union of the rectangles drawn in the first $i - 1$ steps. To draw the $i$th rectangle, we consider the label $i$ on the diagonal. If the diagonal point $p$ on the diagonal immediately above or left of the label $i$ is not in $T$, then the upper left corner of the new rectangle is the rightmost point of $T$ immediately to the left of $p$. If the diagonal point $p$ immediately below or right of the label $i$ is not in $T$, then the lower right corner of the new rectangle is the highest point of $T$ immediately below $p$. If $p$ is in $T$, then the lower-right corner of the new rectangle is the rightmost point of $T$ immediately to the right of $p$. **Right**: Then we assign to the vertex on the *walls* (colored gray) the label of the block that contains that vertex as its own upper left or lower right corner (left panel). Finally, the order of the vertices on the wall will be rearranged according to the permutation. Specifically, vertices on the horizontal wall are aligned in permutation order from left to right, and vertices on the vertical wall are aligned in permutation order from bottom to top (right panel).

where $k(U_j^{(\text{column})}, U_i^{(\text{row})})$ represents the block index to which $(U_j^{(\text{column})}, U_i^{(\text{row})})$ belongs in the rectangulation $t$ on $[0,1] \times [0,1]$, and $\beta_0$ is a non-negative hyper parameter. This beta-Bernoulli hierarchical model can lead to the following marginal likelihood:

$$p_{\text{likelihood}}\left( X \mid t, \boldsymbol{U}^{(\text{row})}, \boldsymbol{U}^{(\text{column})} \right) \propto \prod_k \left( \frac{\Gamma(\mathcal{C}_{k,0} + \beta_0)\Gamma(\mathcal{C}_{k,1} + \beta_0)\Gamma(2\beta_0)}{\Gamma(\mathcal{C}_{k,0} + \mathcal{C}_{k,1} + 2\beta_0)\Gamma(\beta_0)^2} \right), \qquad (50)$$

where $\mathcal{C}_{k,\cdot}$ denotes the number of elements in both the $k$-th block and the binary ($\cdot \in \{0,1\}$) variables. The posterior distribution of the parameters $\boldsymbol{U}^{(\text{row})}, \boldsymbol{U}^{(\text{column})}$ to be estimated is proportional to this joint probability density.

**[S4] Introduction of universal object and partition matroid** - To achieve our SB reformulation of Bayesian relational models, we need a universal object that can contain any rectangulation. The key insight is that permutations of length $n$ have a surjective map to rectangulation of $n$ blocks. More precisely, the following fact has been founded (See Proposition 4.2 in (Reading, 2012)):

**Proposition D.12** (Surjection from permutations to rectangulations). *There exists a surjective map from permutations of length $n$ to rectangulations of $n$ blocks.*

We can explicitly construct a map P2R to transform from a permutation $\sigma$ of length $n$ to a rectangulation P2R$(\sigma)$ as shown in Figure 10.

**Corollary D.13** (Universality). *Let $\rho_n$ be a superpermutation (Definition D.10). Then, for any $t \in \mathcal{T}$, there exists $S \in \mathcal{C}$ such that* U2T $(\rho_n, S) = t$. *Hence, we have* supp$(\hat{\pi}_C) = \mathcal{T}$.

*Proof of Corollary D.13.* For any sample $t \in \mathcal{T}$, since (1) RPC $\rho_n$ contains all possible permutations, and (2) the mapping P2R is surjective, we can confirm that there exists $S \in \mathcal{C}$ such that U2T $(\rho_n, S) :=$ P2R $(\rho_n(s_1) \ldots \rho_n(s_n)) = t$. This completes the proof. $\qquad \square$

**[S5] Bayesian inference** - With everything ready, we can use Algorithm 1 for Bayesian inference. For the auxiliary variables $\boldsymbol{U}^{(\text{row})}$ and $\boldsymbol{U}^{(\text{column})}$, we can infer them concurrently within the main loop using the Metropolis-Hastings scheme with Uniform$([0,1])$ proposals.

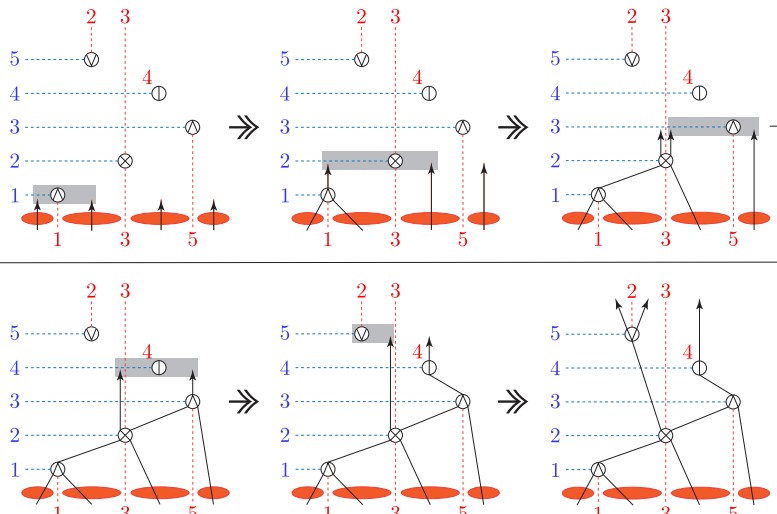

Figure 11: Illustration from permutation matrix to a leveled permutree. First, auxiliary lines (red dashed lines) are drawn below decorations $\otimes$, $\oslash$ and above decorations $\otimes$, $\oslash$. From this point on, we will extend the permutree edges, but it is important to emphasize that the permutree edges do not cross these auxiliary lines. Next, if we look at the auxiliary lines extending all the way to the bottom, we can see that this divides the lower region into smaller sub-regions (indicated by the red oval). Then, one edge is extended from each sub-region. The edges are extended from bottom to top, and when the height of each vertex is reached, the edges adjacent to that vertex are connected (indicated by the gray boxes). This is done until all vertices are covered, resulting in a *leveled permutree*. Finally, if we forget about the vertical position of each vertex in the leveled permutree and focus only on its structure as a directed tree, we obtain the corresponding *permutree*.

### D.8  PERMUTREE: UNIFIED REPRESENTATION OF BINARY TREE AND CAMBRIAN TREE

The purpose of this subsection is to clarify the mapping from permutations to binary and Cambrian trees. Very interestingly, it has recently been known that these combinatorial objects can be transformed in a unified way by a concept called *permutree* (Pilaud & Pons, 2017). Therefore, we will here clarify the mapping to more general permutrees.

**Permutree** (Pilaud & Pons, 2017) - A *permutree* is a new mathematical tool invented recently in the field of combinatorics, which not only represent permutations, trees, partitions, and binary sequences as special cases, but can also interpolate between them (Pilaud & Pons, 2017). Let us begin with the definition of a permutree. We consider a directed tree $\mathbf{T}$ with a vertex set $\mathbf{V}$ of $n$ ($n \in \mathbb{N}$) vertices of degree at least 2, and a set of terminal nodes of degree 1. For technical reasons (discussed immediately below), we dare to pay particular and explicit attention here to the set $\mathbf{V}$ of the "interior vertices" (i.e., vertices of degree at least 2) other than the terminal nodes. Each vertex $\mathbf{v} \in \mathbf{V}$ is assigned a natural number $p(\mathbf{v})$ as a label, using the bijective vertex labeling (one-to-one correspondence) $p : \mathbf{V} \to [n] := \{1, 2, \ldots, n\}$ based on the following *permutree requirements* (Definition 1 in (Pilaud & Pons, 2017)):

(C1) Each vertex $\mathbf{v} \in \mathbf{V}$ has one or two parents, and one or two children.

(C2) If a vertex $\mathbf{v}$ has a left parent (or child), then all labels in the subtree of the left ancestor (or descendant) of $\mathbf{v}$ are smaller than $p(\mathbf{v})$. If $\mathbf{v}$ has a right parent (or child), then all labels in the subtree of the right ancestor (or descendant) of $\mathbf{v}$ are greater than $p(\mathbf{v})$.

A directed tree $\mathbf{T}$ that satisfies the above requirements can be expressed more intuitively and clearly by introducing the notion of *decorations* to the vertices $\mathbf{V}$. We introduce the $n$-tuple decorations $\delta(\mathbf{T}) := (\delta(\mathbf{T})_1, \ldots, \delta(\mathbf{T})_n) \in \{\oplus, \otimes, \oslash, \oslash\}^n$, defined as follows: (i) $\delta(\mathbf{T})_{p(\mathbf{v})} = \oplus$ if $\mathbf{v}$ has one parent and one child, (ii) $\delta(\mathbf{T})_{p(\mathbf{v})} = \otimes$ if $\mathbf{v}$ has two parents and two children, (iii) $\delta(\mathbf{T})_{p(\mathbf{v})} = \oslash$ if $\mathbf{v}$ has one parent and two children, and (iv) $\delta(\mathbf{T})_{p(\mathbf{v})} = \oslash$ if $\mathbf{v}$ has two parents and one child. The representative feature of permutrees can represent various combinatorial objects in a unified manner as follows:

**Remark D.14.** (See Example 4 in (Pilaud & Pons, 2017).) **Binary tree** - Permutrees with decoration $\oslash^n$ have a one-to-one correspondence with rooted planar binary trees on $n$ vertices. **Cambrian tree** - Permutrees with decoration $\{\oslash, \oslash\}^n$ are exactly the Cambrian trees proposed in (Reading, 2006; Chatel & Pilaud, 2014).

**Leveled permutree** - To define the leveled permutree, we start by introducing an additional notion of an *increasing tree*. We consider a directed tree $\mathbf{T}$ with vertex set $\mathbf{V}$. Each vertex $\mathbf{v} \in \mathbf{V}$ is assigned a natural number label $q(\mathbf{v})$, using the bijective vertex labeling (one-to-one correspondence) $q : \mathbf{V} \to [n]$ such that, if $\mathbf{v} \in \mathbf{V}$ is the parent of $\mathbf{w} \in \mathbf{V}$, then $q(\mathbf{v}) < q(\mathbf{w})$ is satisfied. Intuitively, the function $q$ serves to label the vertices $\mathbf{V}$ from 1 to $n$ vertically from bottom to top. Then, a *leveled permutree* is a directed tree $\mathbf{T}$ with a vertex set $\mathbf{V}$ endowed with two bijective vertex labelings $p, q : \mathbf{V} \to [n]$ which respectively define a permutree and an increasing tree. By using two types of labels $p$ and $q$, the horizontal and vertical arrangement of the vertices $\mathbf{V}$ can be explicitly specified.

**Decorated permutation** - For the description of decorated permutations, the notion of a *permutation table* should be prepared first. A permutation table is a geometrical representation of a permutation $\sigma$ with $n$ length by the $(n \times n)$-table, with rows labeled by positions from bottom to top and columns labeled by values from left to right, and with a dot at column $i$ and row $\sigma(i)$ for all $i \in [n]$ (Björner & Wachs, 1991). Now that we are ready, we move on to the description of a decorated permutation. A decorated permutation is a permutation table where each dot is decorated by $\oslash, \otimes, \oslash$, or $\oslash$. One of the important properties of decorated permutations is shown below.

**Proposition D.15.** *(See Proposition 8 in (Pilaud & Pons, 2017).) There exists one-to-one correspondence between decorated permutations with decorations $\hat{\delta} \in \{\oslash, \otimes, \oslash, \oslash\}^n$ and leveled permutrees with $\delta(\mathbf{T}) = \hat{\delta}$. (Figure 11 explicitly shows this bijective map. )*

# E ADDITIONAL EXPERIMENTS AND ABLATION STUDIES

## E.1 RISK OF FIXED UNIVERSAL OBJECTS

This section examines the risks of utilizing a single RPC rather than ensemble multiple RPC samples for universal objects in the SB framework. Our main argument is that **while a single RPC sample carries a significant risk of substantially compromising the uniformity of random subsequences from the RPC sample when the problem size $n$ is small, this risk appears to diminish as the problem scale increases to a certain extent** (e.g., $n \geq 300$).

**Motivation** - As discussed in Section 2.2, to guarantee the uniformity of random subsequences from the universal object in our SB framework, we theoretically need to perform an ensemble over all RPC samples. However, since all candidates of RPC have $n^n$ cases, it is fundamentally impractical to ensemble all cases. Therefore, we wish to understand the risk when using ensembles of very few RPC samples or, in extreme cases, just a single RPC sample.

**Investigation** - We numerically investigate how much a random subsequence extracted from a single RPC sample detracts from uniformity. The space of permutations (i.e., $n!$ possible permutations of length $n$) is quite large, making it difficult to directly examine the degree of deviation from uniformity. Thus, we will indirectly examine the degree of deviation from uniformity by observing the histogram of the position "1" (which, due to symmetry, does not lose generality for any value from 1 to $n$) in the random subsequence extracted from the RPC sample. Figure 12 shows, for $n = 25, 50, 100, 200, 300$, and 400, the histogram of the positions of "1" in $100,000$ subsequences generated from a single RPC sample (left) and the histogram of the positions of "1" in uniformly random permutations (right). The corresponding $p$-value reflects the probability that two histograms are generated identically from the same distribution, as measured by the $\chi^2$ test. Indeed, when $n$ is small (e.g., $n \leq 200$), deviations from uniformity are immediately apparent. However, as $n$ increases (e.g., $n \geq 300$), a behavior gradually approaching uniformity can also be observed.

## E.2 ABLATION AND HYPERPARAMETER ROBUSTNESS FOR QMCMC

This section investigates ablation studies and parameter robustness for QMCMC (Algorithm 2) described in Section 2.5 and Appendix C. Our claims can be summarized in two points:

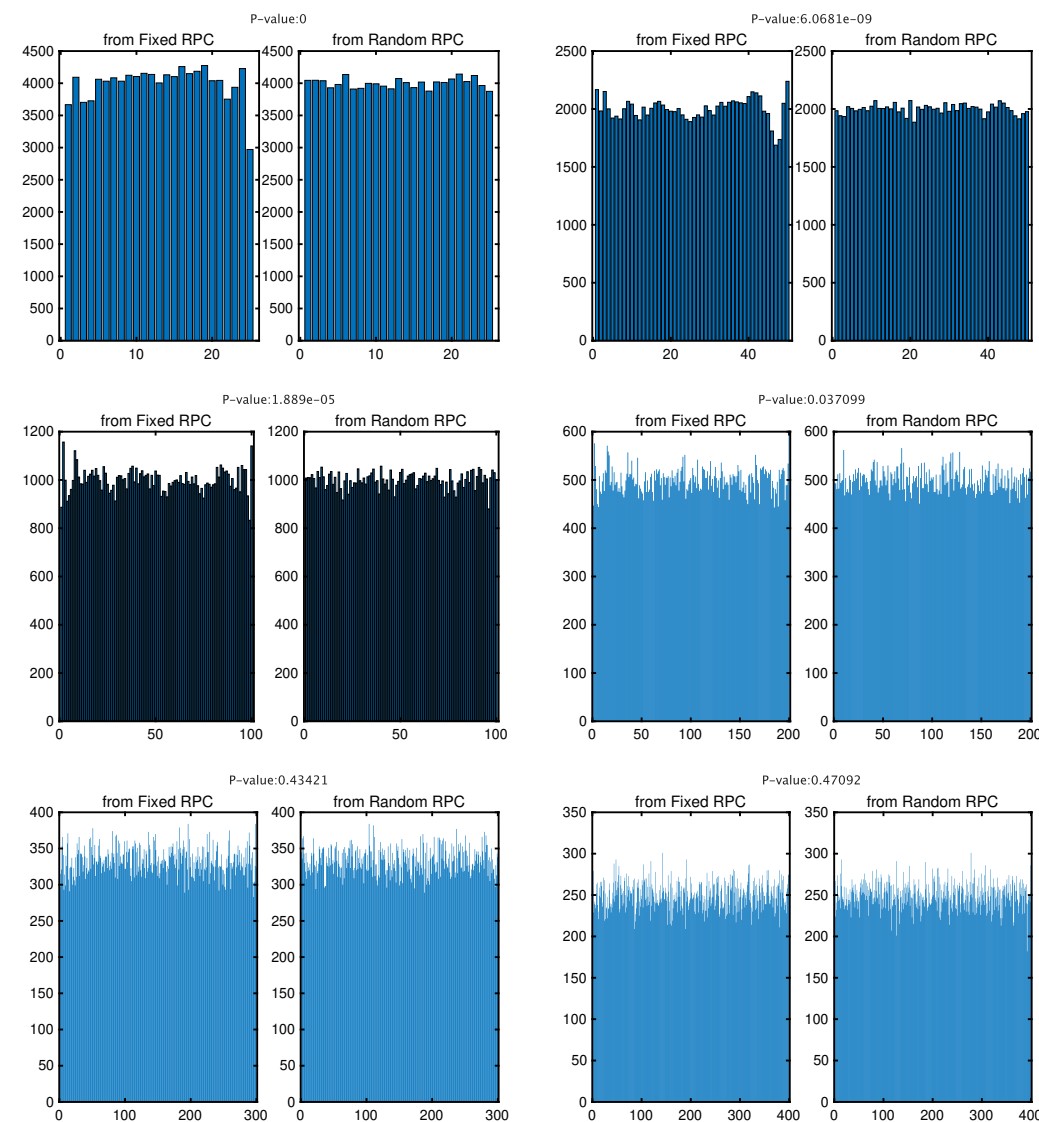

Figure 12: Histogram of the positions of "1" in $100,000$ subsequences generated from a single RPC sample (left) and histogram of the positions of "1" in uniformly random permutations (right) for $n = 25$ (top left), $50$ (top right), $100$ (middle left), $200$ (middle right), $300$ (bottom left), and $400$ (bottom right). Each $p$-value reflects the probability that two histograms are generated identically from the same distribution, as measured by the $\chi^2$ test.

- The inverse temperature parameter $\beta$, which reflects the intensity of the likelihood (that is, $P_{\text{posterior}}(\cdot) \propto \exp(-\beta \cdot P_{\text{likelihood}}(\cdot))$), is typically stable around values like $\beta = 1$. Choosing an extremely small beta could blur the shape of the posterior distribution, introducing significant uncertainty into inference (though scenarios requiring consideration of large uncertainty exist, in which case selecting a small beta may be a viable strategy).

- The coupling parameter $\Gamma$, reflecting the strength of quantum effects, is quite robust regardless of its setting; around $\Gamma = 1$ is reasonable. The Suzuki-Trotter resolution $M$ is effective even at $M \approx 10$. Increasing $M$ shifts the behavior from concentrating posterior probabilities on local modes of the distribution to capturing the overall shape. Empirically, a large $M$ is preferable when focusing on the uncertainty of the distribution, while a small $M$ is better when interested in local modes or maximizing posterior probability.

**Data** - We will newly introduce the three data sets provided by Boehmer & Schaar (2022):

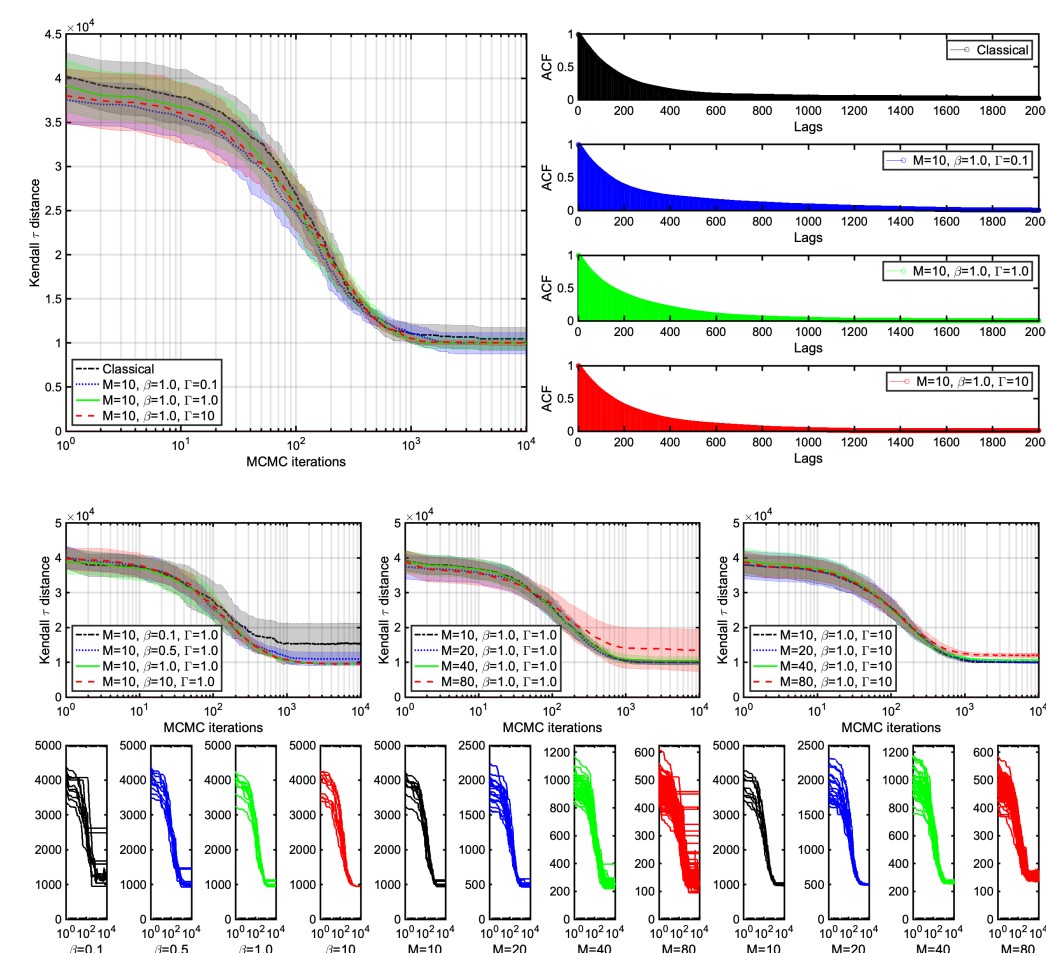

Figure 13: QMCMC diagnosis for **Tennis world ranking** (62 ranked players with 41 voting).

- **Tennis world ranking**: The tennis data includes the weekly rankings of the top 100 male tennis players announced by the Association of Tennis Professionals (ATP) from January 1990 to September 2019. Each year, a selection process for the tennis top 100 was conducted. Each player was a candidate, and each vote corresponded to a player's ranking position during that week. We use 41 voting results for 62 ranked players from the 2018 data.

- **Tour De France**: For each Tour de France from 1903 to 2021, the data includes the finishing times of all riders on every stage. For each edition, a Tour de France election is created where riders are candidates, each vote corresponds to a stage, and riders are ranked by their finishing time. Here, we use 114 ranked riders from 21 votes based on the 2020 data.

- **Spotify**: Spotify data (collected by Oliveira) contains daily rankings of the top 200 most-played songs in any of 53 countries for each day from January 1, 2017, to January 9, 2018. We use data from these rankings, referred to as the "Spotify Monthly Poll," where each vote corresponds to the domestic ranking on a specific day of that month. Specifically, we use the 72 ranked items of 31 voters.

**Investigations on QMCMC diagnosis** - We used these data to address the application issue of TASK 1: CONSENSUS RANKING described in Section 4.Figures 13, 14, and 15 shows QMCMC diagnosis results for **Tennis world ranking**, **Tour De France**, and **Spotify**, respectively. The top panel observes how the behavior of quantum MCMC changes with the strength of the quantum effect $\Gamma$. The left side shows the evolution of the score function during MCMC iterations for $\Gamma = 0$ (i.e., classical MCMC of Algorithm 1), $\Gamma = 0.1$, $\Gamma = 1.0$, and $\Gamma = 10$. The right side depicts the autocorrelation function (ACF). As these illustrate, empirically, increasing the quantum effect strength does not

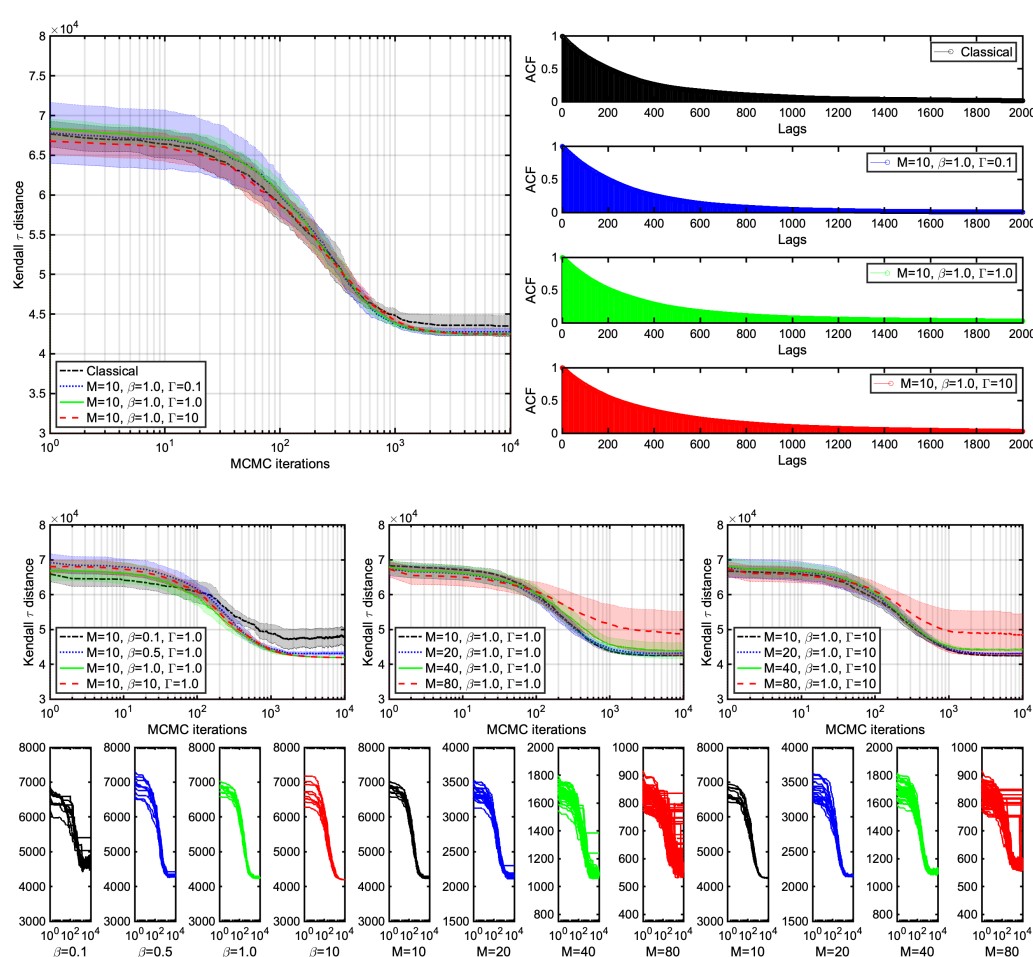

Figure 14: QMCMC diagnosis for **Tour De France** (114 ranked riders with 21 voting).

appear to directly contribute to improving the convergence speed of the MCMC. The middle panel shows the change in MCMC behavior when varying the inverse temperature parameter $\beta$ and the Suzuki-Trotter resolution $M$. The bottom panel shows the raw behavior of each of the $M$ MCMC components decomposed by Suzuki-Trotter decomposition, as shown in the middle panel. As shown in the left figure, setting the inverse temperature parameter to a small value like $\beta = 0.1$ results in significant uncertainty in the inference. As shown in the middle and right figures, selecting a large value like Suzuki-Trotter resolution $M = 80$ causes the $M$ MCMC components to shift their behavior. Rather than concentrating on specific local modes of the posterior probability, they begin to focus on the overall shape of the distribution. From the above, a straightforward strategy would be to first try default settings around $M = 10$, $\beta = 1$, $\Gamma = 1$. If the estimated posterior probability appears too peaky compared to the user's expectations, it would be appropriate to reduce $\beta$ or increase $M$. This allows the inference to focus on the overall shape of the distribution that explicitly captures the uncertainty in the posterior probability.

**Investigations on computational complexity** - We examine QMCMC from the perspective of empirical computational complexity. For comparison, we employ the two popular MCMC schemes as the baselines: the Metropolis-Hastings with removing, adding, and exchanging proposals (RAE) (Jerrum et al., 2004) and the Metropolis-Hastings with leaf-and-shift proposals (LS) (Vitelli et al., 2018). Figure 16 shows QMCMC behavior with the default setting $M = 10$, $\beta = 1$, and $\Gamma = 1$ compared to RAE and LS. The left panel shows the improvement in the score function, while the right panel displays the autocorrelation function. Figure 17 illustrates the empirical time and space complexities of these MCMC methods. All these MCMC update rules exhibit $\mathcal{O}(n)$ time complexity, though their constant factors appear slightly different. Our SB-c and SB-q methods are somewhat

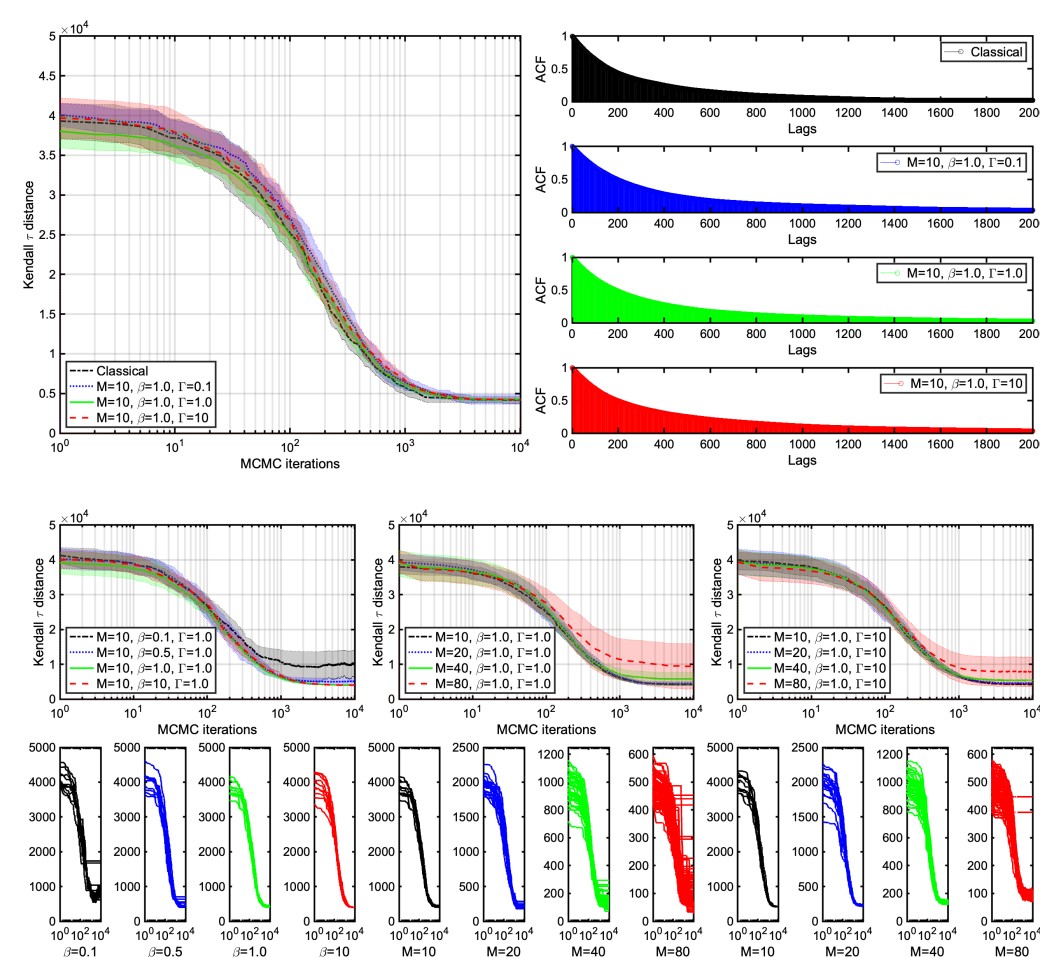

Figure 15: QMCMC diagnosis for **Spotify** (72 ranked songs with 31 voting).

slower due to handling slightly more redundant models. Regarding spatial complexity, while RAE and KS are both $\mathcal{O}(n)$, SB-c and SB-q are $\mathcal{O}(n^2)$. This redundancy stems from the SB framework handling superpermutations of length $n^2$ as models. Empirically, however, the physical memory allocated to observational data appears dominant, and the model-side redundancy does not seem to cause significant overhead. These experiments are conducted in the following environment: 3.8 GHz 8 cores Intel Core i7 with 16 GB 2667 MHz DDR4.

### E.3 ADDITIONAL EXPERIMENTS

The purpose of this section is to provide supplementary information regarding the experiments described in Section 4, TASK 3: RELATIONAL DATA ANALYSIS. We employ the four datasets (Leskovec et al., 2010):

- **Wiki**: 7115 nodes and 103689 edges with diameter 7.

- **Facebook**: 4039 nodes and 88234 edges with diameter 8.

- **Twitter**: 81306 nodes and 1768149 edges with diameter 7.

- **Epinions**: 75879 nodes and 508837 edges with diameter 14.

For readability (since SB-c's behavior deviates only slightly from SB-q), we omit SB-c and report the results for SB-q's default settings ($M = 10$, $\beta = 1.0$, $\Gamma = 1.0$). For comparison, we use the infinite relational model (IRM) (Kemp et al., 2006), the Mondrian process (MP) (Roy & Teh, 2009), the rectangular tiling process (RTP) (Nakano et al., 2014), the block-breaking process (BPP) (Nakano et al., 2020), and the permuton-induced Chinese restaurant process (PCRP) (Nakano et al., 2021).

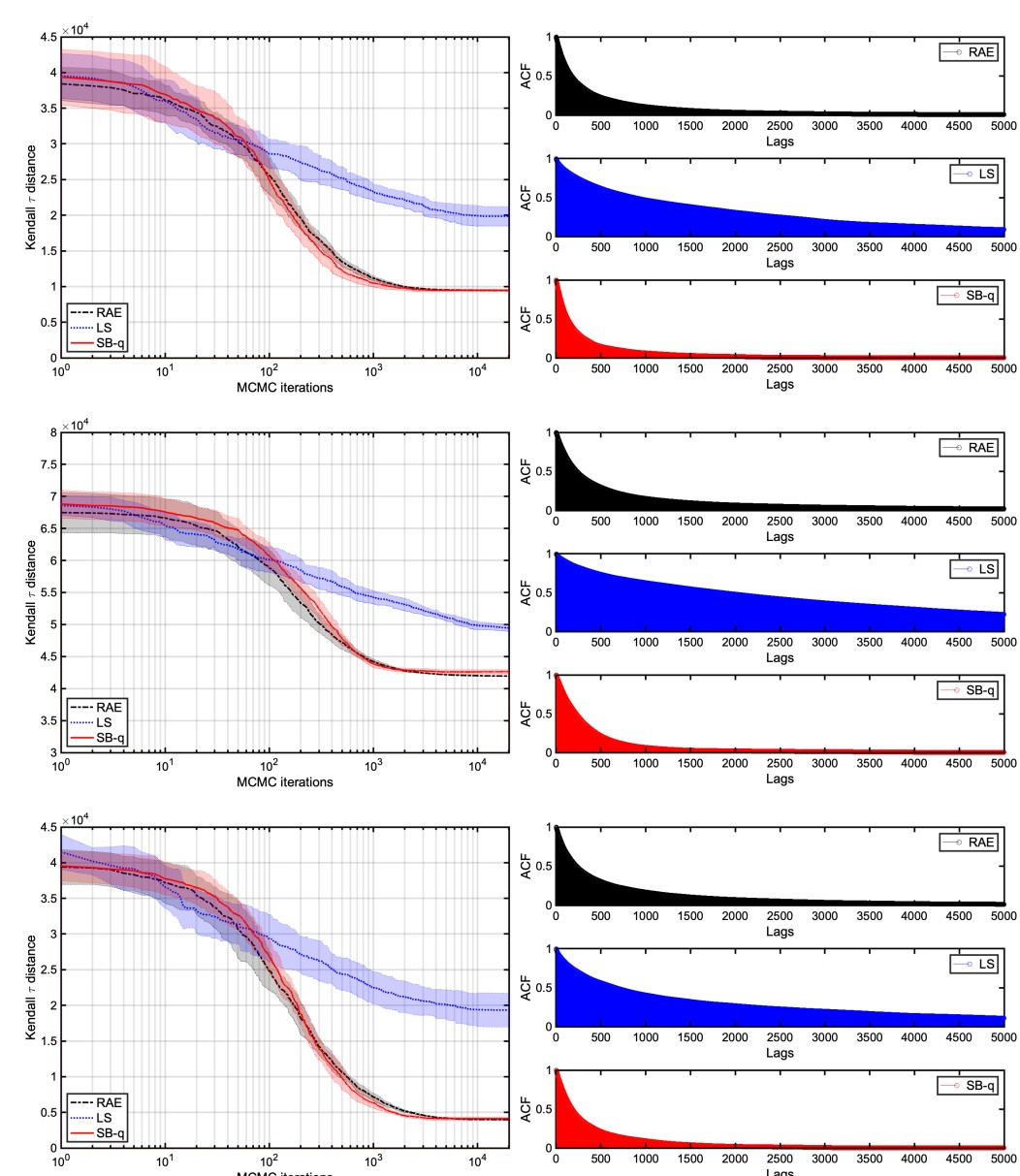

Figure 16: QMCMC diagnosis compared to the Metropolis-Hastings with removing, adding, and exchanging proposals (RAE) (Jerrum et al., 2004) and the Metropolis-Hastings with leaf-and-shift proposals (LS) (Vitelli et al., 2018) for **Tennis world ranking** (top), **Tour De France** (middle), and **Spotify** (bottom). It should be emphasized that LS exhibits a somewhat slower convergence rate here, which is not necessarily optimized for the Kendall tau distance criterion model assumed in this paper. This merely suggests that LS does not fit the Kendall $\tau$ distance criterion model particularly well; it functions very efficiently, for example, with the Mallows model based on the Ulam distance criterion.

Figure 18 reports MCMC evolutions with wall-clock time. These experiments are conducted in the following environment: 3.8 GHz 8 cores Intel Core i7 with 16 GB 2667 MHz DDR4.

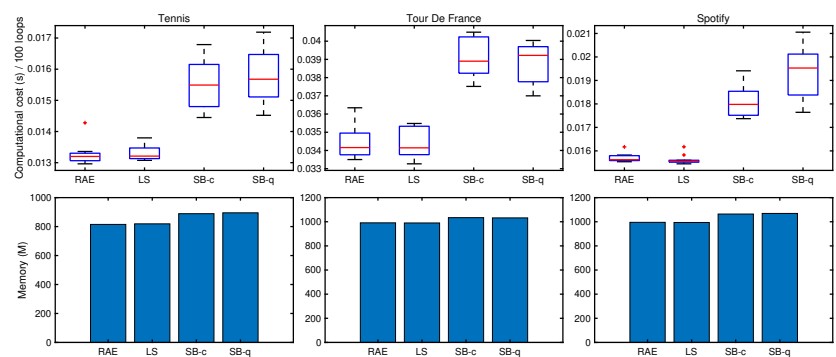

Figure 17: Empirical time and space complexity of SB-c and SB-q compared to RAE and LS.

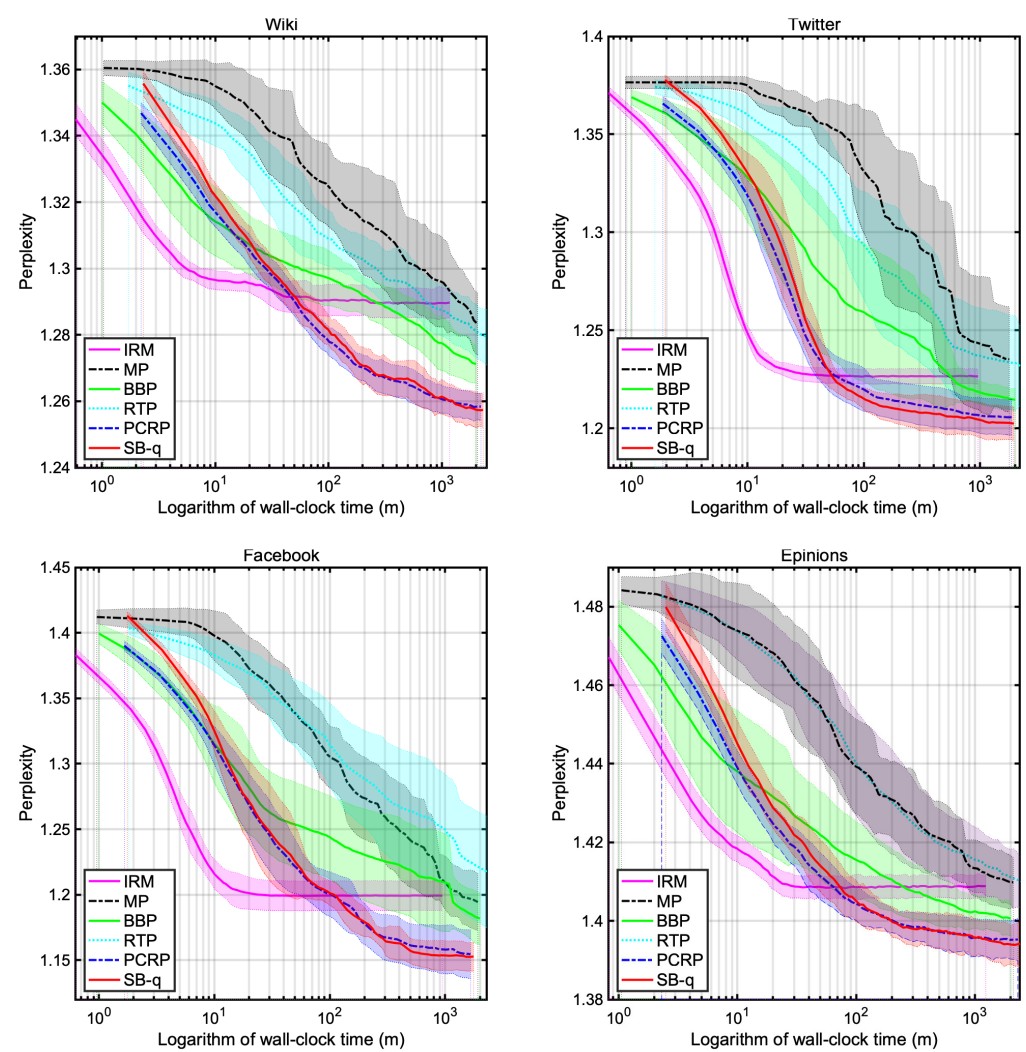

Figure 18: MCMC evolutions for relational data analysis with the infinite relational model (IRM) (Kemp et al., 2006), the Mondrian process (MP) (Roy & Teh, 2009), the rectangular tiling process (RTP) (Nakano et al., 2014), the block-breaking process (BPP) (Nakano et al., 2020), the permuton-induced Chinese restaurant process (PCRP) (Nakano et al., 2021), and the proposed SB-q ($M = 10$, $\beta = 1.0$, and $\Gamma = 1.0$).

