# OpenReview forum: "Bayesian Combinatorial Lottery Ticket Machine: Bayes Meets Extremal Combinatorics"
_ICLR.cc/2026/Conference — Submitted to ICLR 2026_

### Official Review · Reviewer_oaAQ · 2025-10-21

**Soundness:** 4
**Presentation:** 4
**Contribution:** 3
**Rating:** 8
**Confidence:** 2

**Summary:**

The paper introduces a new approach for Bayesian inference with discrete structures. The authors propose to utilize ideas from extremal combinatorics for model construction, propose a generic MCMC inference algorithm and analyze convergence of this algorithm. Furthermore, the paper discusses the application of the framework to, among other problems, the Bayesian Traveling salesperson problem, consensus ranking, hierarchical clustering and relational data analysis.

**Strengths:**

The idea of the paper seems extremely interesting, yet well founded and very natural. I agree with the authors that their idea could be highly relevant for future applications as well as future research.

The paper is methodologically **very** thorough and detailed and the conceptual and theoretical advancements are well supported by the discussed applications. Some experiments on real-world data confirm that the proposed method is sound and actually works in practice.

The paper is also well written, very detailed and has nice supplementary material. I also found that the paper provides a nice introduction to the necessary concepts from extremal combinatorics

**Weaknesses:**

The paper (blissfully) claims to revolutionize all of Bayesian ML, while not even once mentioning that it only considers discrete structures. (This is not a big deal, but making this restriction very clear would certainly help readers from the generic Prob. ML direction)

While the contribution of the paper is indeed nice, I do have the feeling that selling it as a remedy to "all obstacles faced by current BML" is not supported by the evidence provided in the paper. To support such a claim, even more analyses and experiments would be needed.

I see one big issues with the presented method: Prior specification seems to be a big issue because it needs to be done wrt. the super-structure which might not be a "natural" or easy-to-work with object for the relevant problem. However, priors are of course very important for Bayesian inference

Finally, given that the proposed method is quite different from a lot of other methods, I would very much like to see a more detailed discussion regarding (a) weaknesses and (b) potential future directions regarding applications and extensions.

**Questions:**

Could the authors please provide more reasons for this statement: "Our current standpoint is that we do not need much delicate control over a prior model, and that it is sufficient if the universality of the model is preserved."?

Please also feel free to respond to the weaknesses mentioned above.

---

> ### Author Response · Authors · 2025-11-21
> **Response to Reviewer oaAQ (Part 1)**
>
> We are deeply grateful for your thorough reading of our paper and your constructive comments. we are very encouraged by your positive feedback. We respect your generous spirit in appreciating this research, even though our approach represents a new endeavor somewhat different from current machine learning trends.
>
> ***
>
> # Design of Informative Prior
>
> > Could the authors please provide more reasons for this statement: "Our current standpoint is that we do not need much delicate control over a prior model, and that it is sufficient if the universality of the model is preserved."?
>
> > I see one big issues with the presented method: Prior specification seems to be a big issue because it needs to be done wrt. the super-structure which might not be a "natural" or easy-to-work with object for the relevant problem. However, priors are of course very important for Bayesian inference
>
> **Direct answer.** Thank you for taking the time to discuss this crucial point. Designing more flexible prior models is a key challenge that our super Bayesian (SB) framework should address in the next phase. The most promising method we are considering involves using kernel methods via covariates. More specifically, by focusing the SB model itself on universality (i.e., the support of the probability model covering the entire hypothesis space) and applying data-dependent reweighting via covariates to its probability masses, we can achieve a flexible prior model.
>
> **Details.** As you pointed out, flexible prior model design should inherently be the most crucial feature of Bayesian machine learning. Yet, interestingly, looking back at the history of BML, we repeatedly encounter this difficulty in prior model design whenever a new modeling strategy emerges. The most striking example is Bayesian nonparametrics (BNP), which emerged around the year 2000. BNP methods, as a new modeling strategy, are based on novel insights that leverage infinite-dimensional stochastic processes. However, these infinite-dimensional stochastic process models impose very strong theoretical constraints (such as projectivity and exchangeability), resulting in a loss of modeling flexibility. Nevertheless, in the 2010s, the following general-purpose solutions emerged and rapidly advanced:
>
> | Combinatorial object | Standard BNP | with informative prior |
> | ---- | ---- | ---- |
> | Partition | Chinese restaurant process (CRP) | Distance dependent CRP [Blei&Frazier, JMLR2011] |
> | Binary sequences | Indian buffet process (IBP) | Dependent IBP [Williamson+, AISTATS2010] |
> | Hierarchical rectangulation | Mondrian process (MP) | Data-dependent MP [Wang+, ICML2015] |
>
> - [Blei&Frazier, JMLR2011] Blei, D. M., & Frazier, P. I. (2011). Distance dependent Chinese restaurant processes. Journal of Machine Learning Research, 12(8).
>
> - [Williamson+, AISTATS2010] Williamson, S., Orbanz, P., & Ghahramani, Z. (2010, March). Dependent Indian buffet processes. In Proceedings of the thirteenth international conference on artificial intelligence and statistics (pp. 924-931).
>
> - [Wang+, ICML2015] Wang, Y., Li, B., Wang, Y., & Chen, F. (2015, June). Metadata dependent Mondrian processes. In International Conference on Machine Learning (pp. 1339-1347).
>
> These strategies apply reweighting using similarity between observational data vectors and latent covariates to the target stochastic process at a later stage. This strategy provides flexible prior design for various BNP models. For our SB method, applying this post-hoc, data-driven reweighting to the partition matroid also allows us to construct a data-dependent model.

---

> ### Author Response · Authors · 2025-11-21
> **Response to Reviewer oaAQ (Part 2)**
>
> ***
>
> # Emphasis on Our Focus Being on Discrete Combinatorial Structures
>
> > The paper (blissfully) claims to revolutionize all of Bayesian ML, while not even once mentioning that it only considers discrete structures. (This is not a big deal, but making this restriction very clear would certainly help readers from the generic Prob. ML direction)
>
> **Answer.** Thank you for your very important suggestion. As you pointed out, our focus lies in discrete combinatorial structures, and we wish to strongly emphasize this point. In particular, we aim to clarify its positioning within the context of probabilistic programming as follows.
>
> - The concept of realizing general-purpose BML has been systematized as probabilistic programming (including STAN and PyMC) since the 2000s and has undergone rapid development. In fact, general-purpose BML equipped with general-purpose inference mechanisms such as black-box variational inference, Hamilton Monte Carlo, and sequential Monte Carlo has been realized for applications dealing with continuous quantities, such as the posterior distribution of parameters in parametric models and regression. However, automatic modeling and inference mechanisms for combinatorial structures and structural hypotheses remain an unsolved and challenging problem. Our SB framework provides general-purpose modeling and inference for these combinatorial structures and structural hypotheses. Through hybrid utilization with existing probabilistic programming, this will likely become the foundation for BML that can unify the treatment of continuous quantities and discrete structures in the near future.
>
> ***
>
> # Further discussion on weaknesses and future direction
>
> > Finally, given that the proposed method is quite different from a lot of other methods, I would very much like to see a more detailed discussion regarding (a) weaknesses and (b) potential future directions regarding applications and extensions.
>
> **Answer.** Indeed, our SB framework is still in its nascent stages, so we are very pleased to have this opportunity for discussion. We will add further in-depth discussion on the following two topics in supplementary materials.
>
> - *Does MCMC inference exist that guarantees polynomial-time mixing for all combinatorial structure inference problems?* We are currently focused on designing a new MCMC algorithm that eliminates problem-dependent terms (from the current MCMC mixing time bound of Theorem 2.5) and guarantees polynomial-time mixing regardless of the problem. Our strategy is inspired by the following two insights: (1) Our current SB method is a probabilistic model on a *partition matroid*. (2) The strong Rayleigh distribution generally possesses polynomial-time mixing MCMC inference (independently of the problem setting). These two insights suggest that if we can devise a general procedure to transform a probabilistic model on a partition matroid into a strong Rayleigh distribution, we can immediately obtain MCMC inference with guaranteed polynomial-time mixing.
>
> - *NP vs. BPP issue in computational complexity theory.* We expect that our SB framework may provide significant insights into the unsolved problem of NP vs. BPP in computational complexity theory (P vs. NP is very well-known, and this is an equally famous unsolved problem). NP is the class that the problem is **easy to check**. BPP is the class that is **easy to probabilistically solve**. Interestingly, it is currently unknown whether NP < BPP or BPP < NP holds in general. Our SB framework may offer a new perspective on this question. Indeed, as exemplified by the Traveling Salesperson Problem, our SB framework includes several NP-hard problems as probability maximization problems. If it were possible to use the SB framework to control the maximum probability by a constant, this would suggest NP\subseteq BPP. This is a highly intriguing topic, so we wish to discuss what is currently known in depth in the supplementary materials.

---

> > ### Comment · Reviewer_oaAQ · 2025-11-21
> >
> > Thank you for your detailed answer, and for addressing my remaining concerns about the focus on discrete structures, prior elicitation, as well as future directions.
> >
> > I firmly believe that the newly proposed framework is interesting enough, especially in combination with the thorough evaluation of the paper, to offset the current weaknesses of the proposed framework in terms of prior elicitation and similar performance to existing specialized algorithms that were also pointed out by other reviewers.

---

> > > ### Author Response · Authors · 2025-11-21
> > >
> > > We are deeply grateful to Reviewer oaAQ for the foresight and generous assessment. We are greatly encouraged by your kind comments.
> > >
> > > Sincerely,
> > >
> > > The Authors

---

> > > > ### Author Response · Authors · 2025-12-03
> > > > **Paper revision (to Reviewer oaAQ)**
> > > >
> > > > We have highlighted the major improvements in the revised manuscript in dark blue.
> > > >
> > > > **[Outlining prospects for Informative prior design]**
> > > >
> > > > - We have explicitly included this discussion for this topic in **Appendix D.2**.
> > > >
> > > > **[Explicit focus on discrete combinatorial structures]**
> > > >
> > > > - For the reader's ease of reading, we have specified in **Section 1: Introduction**, **lines 136-138**, that our focus lies on discrete combinatorial structures.
> > > >
> > > > ***
> > > > We are greatly encouraged by Reviewer oaAQ’s visionary feedback, so let us express our gratitude once again. Thanks to the constructive discussion, we have been able to improve this paper and gain a clearer perspective on future research directions. We sincerely appreciate your dedicated feedback.
> > > >
> > > > Sincerely,
> > > >
> > > > The Authors

---

### Official Review · Reviewer_iyaa · 2025-10-31

**Soundness:** 3
**Presentation:** 3
**Contribution:** 2
**Rating:** 6
**Confidence:** 3

**Summary:**

The paper proposes a unifying Bayesian modeling and inference scheme—“Super Bayes (SB)”—that imports redundancy-centric ideas from the Lottery Ticket Hypothesis and extremal combinatorics into Bayesian machine learning. The core recipe is proposed as representing latent combinatorial objects (e.g., permutations, partitions, trees) by extracting a constrained subsequence/substructure from a “universal object” (e.g., super-permutation), enforced via a partition matroid. This yields a common target distribution over index sets, enabling a single, simple Gibbs-style MCMC (Algorithm 1) that swaps one index at a time. The also authors provide an explicit upper bound on the mixing time (Theorem 2.5), separating a “price of redundancy” term from a problem-dependent deviation-from-modularity term, and discuss a quantum-inspired acceleration (Algorithm 2). Empirically, the paper instantiates SB for several tasks, consensus ranking, hierarchical clustering, and relational data, showing results comparable or slightly better than bespoke baselines. The quantum variant (SB-q) often edges out the classical one.

**Strengths:**

Originality. The “universal object + partition matroid” formulation is a creative bridge between extremal combinatorics and Bayesian modeling. Framing many Bayesian combinatorial models as probability measures over matroid-constrained subsets yields a common inference mechanism (Algorithm 1). The mixing-time analysis provides transparency often missing in bespoke Bayesian samplers, with an interpretable decomposition into redundancy and likelihood-dependent pieces. The quantum-inspired parallelization, while exploratory, is a novel angle within Bayesian inference for combinatorial structures.

Quality. The SB reformulation is worked out concretely for permutations via a random permutation concatenation (RPC) super-permutation and a partition-matroid constraint ensuring subsequences are permutations. Equivalence to Bayesian TSP under a uniform prior is made explicit (Proposition 2.3). The mixing-time upper bound (Theorem 2.5) is nontrivial and helps pinpoint where model/likelihood design hurts or helps convergence. Multi-domain experiments (ranking, trees, rectangulations) show the unified sampler is competitive with tailored baselines, and SB-q often improves further.

Clarity. The pipeline (S1–S5) is clearly enumerated, and Figure 1 usefully situates universal objects, matroid constraints, substructures, and target structures. Algorithm 1 is simple and easy to implement; the narrative repeatedly connects back to the “family of subsets” view to orient the reader.

Significance. If robust, the approach could reduce engineering churn in Bayesian inference for new combinatorial models by reusing a single sampler and a common analysis template. The analysis highlights a path to principled mixing-time statements in practical Bayesian pipelines, which is a rare contribution in this space.

**Weaknesses:**

Prior control is indirect and limited. The framework “bakes in” prior mass via how often a target structure appears as a valid substructure of the universal object. Fine-grained priors are hard to encode, as acknowledged by the authors in Limitations. It may be helpful to provide recipes to shape priors beyond uniformity: e.g., weighted partition matroids, non-uniform RPC (with provable lack of bias), temperature-controlled counts, or auxiliary potentials over $S$ that map to interpretable priors over $T$.

Fixing a single RPC sample may introduce variance/bias. The paper argues that one RPC is “already highly redundant,” but this is just an assumption. For finite $n$, the realized coverage and multiplicities of target structures under a single draw could distort posterior estimates. The authors may consider adding an ablation comparing: (i) one RPC (fixed), (ii) multiple RPCs with Rao-Blackwellization over $\rho$, (iii) resampling RPC during burn-in. It is valuable to quantify effects on calibration, variance, and convergence.

Scalability and memory footprint is questionable. Time complexity $\mathcal{O}(n)$ per update is good, but space requirement $\mathcal{O}(n^2)$ (storing RPC and indices) may be a bottleneck for large $n$ or high-dimensional generalizations.

Theoretical dependence on $\zeta$-likelihood lacks concrete guidance. The bound depends exponentially on a deviation-from-modularity term. Readers need intuition for realistic magnitudes and how to design likelihoods with a small $\zeta$.

Quantum MCMC section is promising but underspecified. The SB-q description is brief; the performance lift is intriguing but the mechanism and costs (e.g., Trotter steps, coupling parameters) lack ablations. The authors may consider adding a controlled study varying the Trotter number, coupling strengths, and wall-clock vs. ESS/sec comparisons against SB-c and baselines. Clarify computational overhead and when SB-q is worth it.

Experimental scope is limited. Datasets are relatively modest. It’s unclear if gains persist at scale or under distributional shift. The stopping criterion “match empirical convergence of baselines” is subjective. The authors are suggested to report effective sample size (ESS), R-hat, and wall-clock/ESS to standardize comparisons, and to include larger datasets (e.g., $>50$ items in ranking; $>10k$ nodes in networks) and stress tests (noisy rankings and imbalanced matrices).

Generality beyond permutations requires more detail. While the paper outlines mappings $U2T$ from permutations to partitions/trees/rectangulations, the construction details (including ensuring lack of induced prior bias and tractable $\Pi$) deserve more explicit, worked examples. For at least one non-permutation target (e.g., binary trees), the authors are suggested to include a full construction of $\rho$, $C$, $U2T$, demonstrate uniformity/non-uniformity of the induced prior, and discuss computational/storage complexity.

**Questions:**

1.	Can you propose a principled way to incorporate informative priors (e.g., smooth permutations; sparse/balanced partitions) within the SB framework without losing the unified MCMC? A short addendum with a weighted-matroid or auxiliary-potential construction would be valuable.
2.	What is the empirical sensitivity to the particular RPC draw? Could you report variance across multiple $\rho$ samples and whether marginalizing over $\rho$ materially changes predictions or credible intervals?
3.	For each presented $\zeta$-likelihood, can you compute or bound $\zeta$ on the used datasets, and show how it correlates with observed mixing (R-hat/ESS)? This would strengthen the theoretical-empirical bridge.
4.	Please include ablations on the Suzuki–Trotter resolution, coupling parameters, and their effect on ESS/sec and solution quality. What is the additional compute cost vs. SB-c?
5.	Can you add experiments for (a) $n\ge 50$ in ranking with Mallows, (b) larger graphs/matrices in relational tasks, and (c) trees with >1k leaves, reporting memory and runtime?
6.	In Proposition 3.1, uniformity holds for TSP/permutations. For trees/rectangulations/partitions, what priors are induced by your concrete universal objects? Are there tasks where this induced prior is clearly undesirable?
7.	Could you standardize comparisons using ESS, R-hat, autocorrelation time, and wall-clock? “Match empirical convergence” is of qualitative nature and hard to reproduce.
8.	How does SB’s unified sampler compare to state-of-the-art black-box variational inference or generic MCMC within modern probabilistic programming languages on the same tasks (both accuracy and compute)?

---

> ### Author Response · Authors · 2025-11-21
> **Response to Reviewer iyaa (Part 1)**
>
> We are deeply grateful to the reviewer for the thoughtful and constructive suggestions. We are very pleased to receive such comments that delve into the technical essence during the review stage. We are very thankful for the effort and time you have devoted to our paper.
>
> ***
>
> # Informative prior design
>
> > Can you propose a principled way to incorporate informative priors (e.g., smooth permutations; sparse/balanced partitions) within the SB framework without losing the unified MCMC?
>
> **Direct answer.** We address the difficulty of designing informative priors in our SB framework by adding in-depth discussion on the following three approaches, guided by the historical context where Bayesian nonparametrics faced similar challenges in its early stages: (1) weighting toward the superpopulation, (2) avoiding patterns in universal objects of extremal combinatorics, and (3) incorporating data-dependent covariates.
>
> **Details.** Thank you for your highly enlightening suggestion. We too recognized that designing this informative prior is exactly the next intriguing research topic for the SB framework. And we would like to emphasize that, interestingly enough, Bayesian nonparametrics (BNP; which emerged from the historical BML breakthrough of the 2000s) also struggled with designing information prior models during its early stages. This stems from the fact that the design of the stochastic processes used in BNP methods (Dirichlet process, completely random measure, Chinese restaurant process, Indian buffet process) was subject to strict constraints such as projectivity and exchangeability, severely limiting modeling freedom. The three principles outlined below are derived from BNP methods for the first and third, while the second represents a unique principle specific to our SB method.
>
> - *(1) Weighting toward the superpopulation.* When modeling natural science phenomena, it is often desirable to incorporate prior knowledge such as balance laws or power laws into the prior model. Examples include situations where one wishes to induce balanced partitions in hierarchical clustering using binary trees, or to induce the number of elements in each cluster to follow the power law. Looking back at the history of the BNP method, a well-known successful example of designing such an informative prior model is the extension to the Pitman-Yor process [Pitman&Yor, AP1997] to introduce the power law for an infinite mixture model using the Dirichlet process. Such an informative prior model can likely be achieved by weighting population frequencies in the universal objects of extremal combinatorics.
>
> [Pitman&Yor, AP1997] Pitman, J., & Yor, M. (1997). The two-parameter Poisson-Dirichlet distribution derived from a stable subordinator. The Annals of Probability, 855-900.
>
> - *(2) Avoiding patterns in universal objects of extremal combinatorics.* Conversely, when modeling a phenomenon, there are often situations where we intentionally want to impose specific constraints on the prior model. For example, for relational data analysis based on rectangulation, a case in point is when we wish to restrict the rectangulation pattern to hierarchical rectangulations (represented by a $k$-d tree via the Mondrian process [Roy&Teh, NeurIPS2009]). Such an informative prior model can be achieved by introducing pattern avoidance when designing extremal combinatorial universal objects. For instance, in the aforementioned scenario where we wish to restrict rectangular partitioning to hierarchical partitioning, one strategy is to constrain the superpermutation to a separable permutation. This approach is unique to the SB method and was not present in the BNP method.
>
> [Roy&Teh, NeurIPS2009] Roy, D. M., & Teh, Y. W. (2008). The Mondrian process. In Advances in neural information processing systems (pp. 1377-1384).
>
>
> *(3) incorporating data-dependent covariates.* Informative prior design has long been a challenge in BNP methods. Fortunately, a general approach using kernel methods gained widespread adoption in the 2010s. This strategy avoids direct modifications to the difficult-to-handle infinite-dimensional stochastic process. Instead, it applies reweighting using similarity between observational data vectors and latent covariates to the target stochastic process at a later stage. This strategy provides informative prior design for various BNP models, as shown in the table below. For our SB method, applying this post-hoc, data-driven reweighting to the partition matroid also allows us to construct a data-dependent model.
>
> | Combinatorial object | Standard BNP | with informative prior |
> | ---- | ---- | ---- |
> | Partition | Chinese restaurant process (CRP) | Distance dependent CRP [Blei&Frazier, JMLR2011] |
> | Binary sequences | Indian buffet process (IBP) | Dependent IBP [Williamson+, AISTATS2010] |
> | Hierarchical rectangulation | Mondrian process (MP) | Data-dependent MP [Wang+, ICML2015] |

---

> ### Author Response · Authors · 2025-11-21
> **Response to Reviewer iyaa (Part 2)**
>
> [Blei&Frazier, JMLR2011] Blei, D. M., & Frazier, P. I. (2011). Distance dependent Chinese restaurant processes. Journal of Machine Learning Research, 12(8).
>
> [Williamson+, AISTATS2010] Williamson, S., Orbanz, P., & Ghahramani, Z. (2010, March). Dependent Indian buffet processes. In Proceedings of the thirteenth international conference on artificial intelligence and statistics (pp. 924-931).
>
> [Wang+, ICML2015] Wang, Y., Li, B., Wang, Y., & Chen, F. (2015, June). Metadata dependent Mondrian processes. In International Conference on Machine Learning (pp. 1339-1347).
>
> ***
>
> # Prior Property Induced from Random Permutation Concatenation
>
> > In Proposition 3.1, uniformity holds for TSP/permutations. For trees/rectangulations/partitions, what priors are induced by your concrete universal objects? Are there tasks where this induced prior is clearly undesirable?
>
> **Direct answer.** We can summarize the properties of the current default prior model for all combinatorial structure inference tasks assumed by the SB framework as shown in the table below. Furthermore, scenarios where this prior model is clearly unsuitable include, for example, the following relational data analysis case. Since the support of this prior model extends across the entire space of rectangulations, it is not suitable for situations where one wishes to restrict the prior model's support to hierarchical rectangular partitions (i.e., the support of the Mondrian process).
>
> **Details.** The superpermutation via the random permutation concatenation (RPC) leads to the following prior for each BML task:
>
> | Task | Induced prior |
> | ---- | ---- |
> | Consensus ranking | Uniformity on permutations |
> | Clustering | (Lacks a typical model description) |
> | Hierarchical clustering | Uniformity on leveled binary trees |
> | Factor analysis | Uniformity on binary sequences |
> | Hierarchical clustering | Uniformity on leveled binary trees |
> | Causal inference | Uniformity on topological orders |
> | Hierarchical clustering | Uniformity on leveled binary trees |
> | Phylogenetic analysis | Uniformity on leveled Cambrian trees |
> | Relational data analysis | (Lacks a typical model description) |
>
> The default RPC and partition matroid induce the *uniform* random permutation. Therefore, a sufficient condition for uniformity on the target combinatorial structure is that the mapping from permutations to the target combinatorial structure is a *bijection*. The uniformity of the target combinatorial structure in the above table stems entirely from the existence of such a bijection. We will discuss the clustering (partitions) and the relational data analysis (rectangulations) tasks, which are not explicitly shown in the table, in more detail below.
>
> - *Partitions.* We can derive a prior model for Ewens' exchangeable partition using Kingman's paintbox mechanism as the most standard alternative [Kingman, JAP1982]. We detail this procedure in the supplementary materials.
>
> [Kingman, JAP1982] Kingman, J. F. C. (1982). On the genealogy of large populations. Journal of Applied Probability 19A 27–43.
>
> - *Rectangulation.* Unfortunately, based on our current understanding, it appears extremely difficult to identify or control this induced distribution. The reason lies in the fact that rectangulations have a one-to-one correspondence (bijection) with a special class of permutations called 2-clumped permutations [Reading, EJC2012]. In other words, rectangulations have a surjection onto the entire set of all permutations. For example, using Reading’s mapping [Reading, EJC2012], the permutation 35124 and the permutation 31524 are transformed into the same rectangulation. Consequently, the rectangulation model induced from uniform random permutations results in a biased distribution, as if some rectanglation samples were duplicated and weighted. Unfortunately, current knowledge makes it difficult to precisely determine the shape of this distribution. Furthermore, constructing a uniform model over rectanglations as an example of controlling this prior distribution is also not straightforward. This difficulty stems from the non-triviality of constructing a universal object that generates a uniformly random distribution over 2-clumped permutations.
>
> [Reading, EJC2012] Reading, N. (2012). Generic rectangulations. European Journal of Combinatorics, 33(4), 610-623.

---

> ### Author Response · Authors · 2025-11-21
> **Response to Reviewer iyaa (Part 3)**
>
> ***
>
> # Explicit SB framework description for all tasks
>
> > Generality beyond permutations requires more detail. While the paper outlines mappings U2T from permutations to partitions/trees/rectangulations, the construction details (including ensuring lack of induced prior bias and tractable Π) deserve more explicit, worked examples.
>
> **Answer.** We have added supplementary materials detailing specific universal object configurations and the transformation mapping U2T to target structures for each example of clustering, hierarchical clustering, factor analysis, causal inference, phylogenetic analysis, and relational data analysis (Appendix D; D.2-D.7). Thank you for your important suggestions.
>
> ***
>
> # Deal with Redundant Memory O(n^2) Caused by SB Method
>
> > Scalability and memory footprint is questionable. Time complexity O(n) per update is good, but space requirement O(n^2) (storing RPC and indices) may be a bottleneck for large n or high-dimensional generalizations.
>
> **Direct answer.** This O(n^2) redundant memory is a theoretical requirement for model universality. Therefore, in practical scenarios, it is possible to sacrifice model universality and implement this O(n^2) memory with a constant bound.
>
> **Details: Possibility of truncation.** Our O(n^2) memory requirement stems from theoretically guaranteeing the universality of the model, namely, that a sufficiently redundant sequence of length O(n^2) contains all permutations of length n. Therefore, if we are willing to sacrifice this universality, we can empirically truncate this redundant O(n^2) sequence with a shorter one. In situations where computing resources are limited (e.g., inference on mobile devices), one might want to empirically suppress space complexity. In such cases, we can reduce redundant universal objects (e.g., superpermutations) by thinning them out. The most naive method is to use a savings constant $K$ to restrict, for example, a superpermutation of length $n^2$ to only $n^2/K$ elements extracted uniformly at random. Such pruning can straightforwardly reduce the space complexity and mixing time to $1/K$. So how much of a negative impact does such empirical saving actually have on performance? We present the performance (Kendall tau distance) for cases where $K = 1, 2$, and $4$ for the consensus ranking task (described in Section 4, Task 1):
>
> | **Dataset: Sushi** | $K=1$ | $K=2$ | $K=4$ |
> | ---- | ---- | ---- | ---- |
> | **SB-c** | $20.947$ $(2.216)$ | $21.193$ $(1.486)$ | $22.482$ $(1.681)$ |
> | **SB-q** | $19.679$ $(1.935)$ | $20.855$ $(1.029)$ | $22.016$ $(1.006)$ |
>
> | **Dataset: CF** | $K=1$ | $K=2$ | $K=4$ |
> | ---- | ---- | ---- | ---- |
> | **SB-c** | $21.328$ $(1.486)$ | $21.412$ $(1.511)$ | $22.093$ $(1.492)$ |
> | **SB-q** | $21.142$ $(1.029)$ | $21.298$ $(1.157)$ | $21.855$ $(1.198)$ |
>
> | **Dataset: CB** | $K=1$ | $K=2$ | $K=4$ |
> | ---- | ---- | ---- | ---- |
> | **SB-c** | $19.481$ $(1.681)$ | $19.939$ $(1.596)$ | $19.984$ $(1.674)$ |
> | **SB-q** | $19.402$ $(1.420)$ | $19.535$ $(1.487)$ | $19.772$ $(1.287)$ |
>
> (Note: Smaller distances mean better prediction performance.)
>
> As can be seen from the experimental results, reducing the redundancy of the superpermutation by $K=2,4$ does not cause a significant decrease in performance. This is a reasonable result because the superpermutation itself is already sufficiently redundant, and even when reduced to $1/K$ of its original size, it still seems to contain most permutation candidates as its subsequences. In this way, the topic of universality when reducing the size of superpermutations is one of the interesting research topics both theoretically and experimentally. In our next phase in the near future, we aim to provide a deeper exploration of this topic.

---

> ### Author Response · Authors · 2025-11-21
> **Response to Reviewer iyaa (Part 4)**
>
> ***
>
> # Concrete Guidance for Problem-dependent Term of MCMC Mixing Time Bound
>
> > Theoretical dependence on $zeta$-likelihood lacks concrete guidance. The bound depends exponentially on a deviation-from-modularity term. Readers need intuition for realistic magnitudes and how to design likelihoods with a small $zeta$.
>
> **Direct answer.** To provide readers with concrete intuition regarding the problem-dependent term in our MCMC mixing time analysis, we present two specific examples. Additionally, as a more general topic, we add a discussion of the strategy we are focusing on to eliminate this problem-dependent term in the near future. This aims to remove the problem-dependent term from the upper bound on MCMC mixing time, thereby guaranteeing mixing in polynomial time.
>
> **Details: two concrete case studies.** We have demonstrated how to utilize our current MCMC mixing time analysis results through two scenarios: positive and negative.
>
> - *Negative example.* Unfortunately, as shown in Figure 3 of the supplementary materials, we can conceive of scenarios for the Traveling Salesman Problem where the mixed-time upper bound becomes exponential. However, as Figure 3 indicates, such a bad case also represents a problem setting where a different inference algorithm could likely discover the minimum cycle easily. From the above, it is clear that our Algorithm 1 still has significant room for improvement. This provides important insights for future algorithmic enhancements, which will be discussed later (the third term, below soon).
>
> - *Positive example.* Fortunately, we can also design specific likelihood models that allow us to bound the MCMC mixing time of Algorithm 1 to polynomial time. As a concrete example shown in Remark D.3, by assuming a nonparametric topological likelihood in a mixture model dealing with partitions, we can easily guarantee polynomial-time mixing. This demonstrates that mixing time analysis provides significant insights for model design, particularly for likelihood models.
>
> **Details: elimination of problem-dependent term.** We are currently focused on designing a new MCMC algorithm that eliminates problem-dependent terms and guarantees polynomial-time mixing regardless of the problem. Our strategy is inspired by the following two insights:
>
> - Our current SB method is a probabilistic model on a partition matroid.
>
> - The strong Rayleigh distribution generally possesses polynomial-time mixing MCMC inference (independently of the problem setting).
>
> These two insights suggest that if we can devise a general procedure to transform a probabilistic model on a partition matroid into a strong Rayleigh distribution, we can immediately obtain MCMC inference with guaranteed polynomial-time mixing. We have, in fact, very recently discovered such a method. Since it involves somewhat technically non-trivial topics beyond the scope of this paper, we intend to summarize the results in a separate paper.
>
> ***
>
> # Empirical Calculation of MCMC Mixing Time
>
> > For each presented $zeta$-likelihood, can you compute or bound $zeta$ on the used datasets, and show how it correlates with observed mixing (R-hat/ESS)? This would strengthen the theoretical-empirical bridge.
>
> **Answer.** Unfortunately, the answer is no. To be honest, based on the author's current knowledge, there is no concrete solution to the problem-dependent term in the MCMC mixing time. However, we are not pessimistic about this for the following two reasons.
>
> - The problem-dependent term clearly has a form related to matroid theory, so significant improvements might be possible with new insights from optimization specialists (as additional information within anonymity constraints, the current project team leans somewhat toward Bayesian researchers and does not include matroid theory specialists).
>
> - As discussed above regarding MCMC mixing time improvements, transforming the current partition matroid modeling into a strongly Rayleigh distribution class could potentially eliminate this challenging problem-dependent term (given the project team's characteristics, we are currently focusing on this approach).

---

> ### Author Response · Authors · 2025-11-21
> **Response to Reviewer iyaa (Part 5)**
>
> ***
>
> # Computational Overhead When Extending to Quantum MCMC
>
> > Please include ablations on the Suzuki–Trotter resolution, coupling parameters, and their effect on ESS/sec and solution quality. What is the additional compute cost vs. SB-c?
>
> **Direct answer.** We will add experimental results for varying Quantum MCMC parameters to the supplementary materials. Furthermore, the overhead when converting from classical MCMC to quantum MCMC requires $O(Mn)$ additional computation to count the number of matching active indices in superpermutations associated with two adjacent MCMC chains (reflecting quantum effects), when the Suzuki-Trotter resolution of the quantum MCMC is $M$ (i.e., when performing $M$ trials of mutually dependent MCMC chains).
>
> **Details.** Classical MCMC's independent $M$ MCMC trials and quantum MCMC's $M$ mutually dependent MCMC trials (when Suzuki-Trotter resolution is set to $M$) require roughly equivalent computational cost. More precisely, the $m$-th trial in quantum MCMC requires additional computation due to its dependence on both the $(m-1)$-th and $(m+1)$-th trials. This additional computation involves a procedure that counts how many times the active indices $s_{1},\dots,s_{n}$ of the superpermutation coincide between two adjacent MCMC chains, reflecting the quantum effect. Therefore, quantum MCMC with Suzuki-Trotter resolution set to $M$ incurs an $O(Mn)$ computational overhead compared to $M$ independent trials of classical MCMC. While the computational cost of the independent $M$ trials of classical MCMC itself is $O(Mn)$, empirically, the computational portion reflecting the quantum effect is very small and largely negligible. We will add experimental results based on wall-clock time to demonstrate this.
>
> ***
>
> # Metrics for Monitoring MCMC
>
> > Could you standardize comparisons using ESS, R-hat, autocorrelation time, and wall-clock? “Match empirical convergence” is of qualitative nature and hard to reproduce.
>
> **Direct answer.** We would like to add reporting of the autocorrelation of the log posterior probability and wall-clock time in MCMC. However, since this metric may only capture certain aspects of convergence, particularly in MCMC inference involving combinatorial structures, we would like to add the following details to the supplementary materials to the discussion.
>
> **Details.** We appreciate this constructive and highly thoughtful suggestion. We strongly recognize the importance of this topic. However, it appears to involve matters that are fundamental to general Bayesian combinatorial inference and require very careful discussion, extending far beyond the scope of the SB method in this paper. We authors are frankly very troubled by the question of whether there truly exists a metric capable of detecting mixing in MCMC for combinatorial structures, including ESS, autocorrelation, and R-hat. More specifically, we feel that defining the concepts of mean and covariance required for calculating ESS, autocorrelation, and R-hat for combinatorial structures is a highly non-trivial and difficult challenge. Since you are a very thoughtful expert reviewer, we wish to address this issue as honestly as possible. Therefore, please forgive us for not providing a concise answer. We elaborate below.
>
> - **Non-triviality of ESS, autocorrelation, and R-hat for combinatorial structures.**
> We fully understand that ESS, autocorrelation, and R-hat are highly useful metrics for identifying mixing when inferring parameters such as scalars and vectors via MCMC. Therefore, these metrics are invaluable when examining the mixing of MCMC parameter inference algorithms for parametric probability models. However, when inferring combinatorial structures, which is the primary focus of this paper, the application of these metrics becomes non-trivial. The reason lies in the fact that calculating ESS, autocorrelation, and R-hat requires computing the *mean* and *covariance* of the target. For example, consider the context of relational data analysis where we are inferring a rectangular partition as the target. Calculating these metrics requires the mean and variance of the collection of rectangular partitions, concepts that are non-trivial. Indeed, the concept of *mean* in combinatorial structures often becomes an estimation problem itself. Examples include the consensus ranking estimation problem for permutations and the supertree construction problem for binary trees, both of which are treated as challenging problems (both are NP-hard). From the above, it is clear that calculating ESS, autocorrelation, or R-hat using mean and variance for combinatorial structures is not straightforward.

---

> ### Author Response · Authors · 2025-11-21
> **Response to Reviewer iyaa (Part 6)**
>
> - **Pitfalls in ESS, autocorrelation, and R-hat calculations for alternative parameters.** Calculating metrics such as ESS, autocorrelation, or R-hat for combinatorial structures is difficult, so we might be tempted to try calculating these metrics for some alternative parameter instead. For example, the most straightforward approach might be to monitor the value of the posterior probability (the conditional probability of the target combinatorial structure given the data) and calculate metrics like ESS for that. However, such an approach immediately reveals itself as a misapplication of metrics like ESS in the context of MCMC mixing detection. One reason for this is the Rashomon effect and the issue of distinguishability. As a general phenomenon in combinatorial structure inference problems, the existence of multiple structural hypotheses with similar posterior probabilities is termed the Rashomon effect. In other words, the posterior probability distribution of the target structural hypothesis suggests that multiple local modes of similar performance can exist. Therefore, even if multiple MCMC chains are trapped in different local modes, monitoring only a metric like the R-hat of the posterior probability risks misinterpreting this as mixing due to the Rashomon effect.
>
> **Our revision policy.** As you suggested, monitoring MCMC using objective metrics is extremely important. Inspired by similar MCMC monitoring approaches, such as those described in [Lin&Fisher, AISTATS2012], we would like to add results from monitoring the autocorrelation of the log posterior probability. However, based on the above discussion, accurately capturing the mixing of MCMC for combinatorial structure inference appears to be an inherently challenging task (far beyond the scope of this paper). We will do our best within the confines of this paper.
>
> [Lin&Fisher, AISTATS2012] Lin, D., & Fisher, J. (2012). Efficient sampling from combinatorial space via bridging. In Artificial Intelligence and Statistics (pp. 694-702).
>
> **Further discussion.** We intend to make this SB framework widely available in the near future as a general-purpose BML tool (or as a module within existing probabilistic programming frameworks) to users beyond the Bayesian experts. Therefore, we believe metrics for checking mixing are highly useful. However, as discussed above, particularly when focusing on the MCMC inference for combinatorial structures, determining whether existing metrics can effectively check for MCMC mixing is itself a very challenging research issue. Certainly, aspects of convergence can be observed through measures like autocorrelation of log-posterior probabilities, but whether this truly reflects MCMC mixing within the vast hypothesis space of combinatorial structures is somewhat questionable. In fact, if we were to widely release such metrics as tools, I am slightly concerned they might mislead general users or non-specialists. If you have noticed or felt anything from the above discussion, we would greatly appreciate your valuable suggestions. In any case, we are deeply grateful for raising such an important topic.
>
>
> ***
>
> # Additional experimental results.
>
> Thank you for your very constructive suggestions. We are currently conducting these additional experiments to incorporate them into the supplementary materials. (While we must maintain anonymity, we ask for your understanding that this project is still in its nascent stages and does not yet have immediate access to large-scale resources, so these experiments may take a little time.)

---

> > ### Author Response · Authors · 2025-12-03
> > **Paper revision (to Reviewer iyaa)**
> >
> > We have highlighted the major improvements in the revised manuscript in dark blue.
> >
> > **[Discussion on prospects for informative priors]**
> >
> > - We have explicitly included this discussion in **Appendix D.2**.
> >
> > **[Disclosure of risks arising from fixed universal objects]**
> >
> > - In **lines 260-265**, we have explicitly stated that fixed RPCs carry the risk of deviation from uniformity.
> >
> > - To intuitively quantify this degree of deviation, we have presented numerical experiments using chi-squared tests in **Appendix E.1**.
> >
> > **[Addition of guidance on problem-dependent terms in MCMC mixing time analysis]**
> >
> > We have provided clearer guidance for readers for this point:
> >
> > - We have added guidance early in the manuscript (**lines 140-141**) to lead readers to the two concrete examples in **Appendix B.4** and **Remark D.3**.
> >
> > - In **lines 483-485**, we have emphasized that this is an important issue to address in the near future.
> >
> > **[Detailed experimental report on quantum MCMC]**
> >
> > - We have added additional experimental reports, including MCMC diagnosis, ablation studies, and robustness of hyper parameters, in **Appendices E.2** and **E.3**.
> >
> > **[Additional investigation into large-scale data]**
> >
> > - We have reported results for three datasets (n=62, 114, 72) on the consensus task in **Appendix E.2** and four datasets (7115, 4039, 81306, 75879 nodes) on the relational data analysis task **Appendix E.3**, as per your request. Additionally, we received a request for hierarchical clustering on over 10k tree structures. However, we were unable to complete all computations during the discussion period. We intend to add 10k tree results to the camera-ready manuscript if accepted (though initial observations suggest these results do not significantly impact our claims).
> >
> > ***
> > We are deeply grateful that our manuscript has become so informative thanks to your thorough recommendations. Unfortunately, unforeseen technical difficulties this year interrupted our discussions. However, we hope our responses and these revisions meet your expectations. We deeply appreciate Reviewer iyaa’s dedicated contributions.
> >
> > Sincerely,
> >
> > The Authors

---

### Official Review · Reviewer_n47q · 2025-11-01

**Soundness:** 3
**Presentation:** 1
**Contribution:** 2
**Rating:** 2
**Confidence:** 4

**Summary:**

The paper proposes a framework for Bayesian machine learning that represents target combinatorial structures (permutations, trees, partitions, etc.) as random substructures of "universal objects" from extremal combinatorics (e.g., superpermutations).
That is, instead of defining probability distributions directly over combinatorial objects (permutations, trees, partitions), they represent them as random subsequences of "universal objects". E.g., for permutations of length n, use a "superpermutation" of length n^2 that contains all permutations as subsequences. This converts the problem to sampling subsets of a large fixed structure with partition matroid constraints.
They provide a unified MCMC algorithm with mixing time bounds and demonstrate applications to consensus ranking, hierarchical clustering, and relational data analysis.

**Strengths:**

- The unified MCMC framework that applies across multiple combinatorial structures (permutations, trees, partitions, rectangulations) is *conceptually* elegant.
- The explicit mixing time upper bound for the proposed MCMC are informative and links a redundancy term and problem-dependent term.
- The paper includes experiments that span diverse applications (ranking, clustering, relational analysis). The authors show the method performs comparably to established baselines.

**Weaknesses:**

- The connection to the Lottery Ticket Hypothesis (LTH) is quite superficial and misleading. LTH concerns learned sparse subnetworks emerging from training, while this work simply uses pre-existing universal mathematical constructions. The analogy reduces to "universal objects are universal" and serves primarily as marketing rather than providing genuine conceptual insight or technical contribution to extremal combinatorics or LTH.
- While the authors show that one can do this, the claimed advantages are not convincingly demonstrated.
- The O(n^2) memory usage appears highly significant for all but trivial problem sizes.
- Contributions to extremal combinatorics or LTH-related deep learning are lacking.
- Contribution to Bayesian ML is minor. The unified MCMC is nice in principle, but apparently paying O(n^2) memory cost for the redundancy is not. They also leverage existing theory on strongly Rayleigh measures to get mixing time.
- Experiments show only "comparable" or marginally better performance than baselines.
- Despite the universality claims, The mixing time bound still contains an uncontrolled problem-dependent term that they can't control, undermining universality.
- The paper's scope and motivation are unclear. It attempts to cover an eclectic mix of topics, including extremal combinatorics, Bayesian modeling, Lottery Ticket Hypothesis, Traveling Salesperson, and quantum systems / computing without coherent motivation or obvious links.
- Section 2.5 on quantum systems and quantum MCMC is marked "optional" yet it was included in the paper.
- The obstacles O1-O3 are stated at such a high level that it's impossible to verify whether they were actually overcome.
- The writing quality obscures the contributions. The abstract oversells the work with vague claims about "new breakthroughs."
- The paper (specifically also title and abstract) are dramatically overselling the results.
- The introduction immediately dives into technical preliminaries without motivation.
- Terms like "infinite description length" and statements like "we may need deep knowledge of measure-theoretic probability theory" use inflated language without clear definitions or justification for a broader ML audience.
- The experimental evaluation is weak. Results show the method is "comparable", not superior (hence no real breakthrough), raising questions about practical value given the added complexity and memory costs.
- No ablation studies examine the impact of the redundancy or universal object.

**Questions:**

- Could you maybe introduce the motivation behind your work already early on?
- Can you clarify O1-3 so that we clearly link your results to the obstacles?
- Could you discuss and expand on the problem-specific term in your bound?
- What evidence supports that the redundancy overhead is worthwhile given this limitation?
- Remove or significantly de-emphasize the LTH connection unless you can establish a deeper technical relationship.
- I'd propose focusing the paper content a bit, so that your main message is clearer. Concepts like LTH and quantum systems make it harder to see a common thread.

---

> ### Author Response · Authors · 2025-11-21
> **Response to Reviewer n47q (Part 1)**
>
> We are deeply grateful for the considerable time and effort you have devoted to our paper. The reviewer's highly insightful comments have been of critical assistance in improving our work. We are revising the paper based on the following policies. We sincerely hope these changes address your concerns.
>
> ***
>
> # Clarification of Our Motivation
>
> > Could you maybe introduce the motivation behind your work already early on?
>
> **Direct answer.** We have added the following explanation to Sec. 1 Introduction.
>
> **Details.** Our motivation is to build a general-purpose framework/library for Bayesian machine learning (BML), similar to TensorFlow or PyTorch for deep learning. While probabilistic programming (e.g., STAN and PyMC) has achieved considerable success from the 2000s, particularly in regression and likelihood function modeling and general inference mechanisms (such as Hamiltonian Monte Carlo and sequential Monte Carlo) for continuous quantities, it has yet to realize a general modeling and inference mechanism for discrete structures (combinatorial structures and structural hypotheses), which remain central to many BML applications. This paper provides a solution to this unresolved challenge using extremal combinatorics.
>
> ***
>
> # Current Achievement Level Regarding Conventional Obstacles
>
>  > Can you clarify O1-3 so that we clearly link your results to the obstacles?
>
> **Direct answer.** We can broadly say that our SB framework provides one solid solution for [O1: Difficulty of model construction] and [O2: Complication of deriving inference algorithms]. While it doesn't fully resolve [O3: Intricacy of algorithmic theoretical properties], we believe it takes a definite step forward.
>
> **Details.** Our claims can be summarized as follows.
>
> - *O1: Difficulty of model construction.* Our SB framework provides a solid solution. In fact, it is no exaggeration to say that nearly all structural hypotheses frequently encountered in BML can be reduced to permutations, partitions, binary sequences, binary trees, causal graphs, Cambrian trees, rectangular partitions, or their combinations. Fortunately, the SB framework can unify all of these in the form of superpermutations and partition matroids.
>
> - *O2: Complication of deriving inference algorithms.* Our SB framework is one of the promising solutions. The essence of the SB framework lies in its ability to unifiedly reduce various combinatorial structures to the form of partition matroids. Consequently, for instance, Algorithm 1 (the Gibbs-type scheme) can be applied as the most fundamental example. Furthermore, an intriguing aspect is that improvements to inference algorithms on this partition matroid can be applied unifiedly to all combinatorial structure inference problems, independent of specific application instances. Indeed, as we demonstrate in Section 2.5, developments like quantum MCMC can be automatically and systematically applied to all combinatorial structure inference problems immediately. Considering that conventional BML has developed separate inference algorithms for each specific application, we are confident our strategy constitutes a significant step forward.
>
> - *O3: Intricacy of algorithmic theoretical properties.* Our current achievements in analyzing MCMC mixing times are limited by the inclusion of problem-dependent terms. As we will discuss in the next item, this represents a significant research challenge pointing toward a direction we aim to address in the near future, rather than an inherent weakness of the SB framework.

---

> ### Author Response · Authors · 2025-11-21
> **Response to Reviewer n47q (Part 2)**
>
> ***
>
> # Additional discussion on MCMC mixing time
>  > Could you discuss and expand on the problem-specific term in your bound?
>
> **Direct answer.** Regarding the interpretation of MCMC mixing time analysis, we improve clarity for two specific application examples and additionally add discussion on our policy to eliminate this problem-dependent term in the near future (with the prospect of guaranteeing polynomial-time mixing regardless of the problem).
> **Details: two concrete case studies.** We have demonstrated how to utilize our current MCMC mixing time analysis results through two scenarios: positive and negative.
>
> - *Negative example.* Unfortunately, as shown in Figure 3 of the supplementary materials, we can conceive of scenarios for the Traveling Salesman Problem where the mixed-time upper bound becomes exponential. However, as Figure 3 indicates, such a bad case also represents a problem setting where a different inference algorithm could likely discover the minimum cycle easily. From the above, it is clear that our Algorithm 1 still has significant room for improvement. This provides important insights for future algorithmic enhancements, which will be discussed later (the third term, below soon).
>
> - *Positive example.* Fortunately, we can also design specific likelihood models that allow us to bound the MCMC mixing time of Algorithm 1 to polynomial time. As a concrete example shown in Remark D.3, by assuming a nonparametric topological likelihood in a mixture model dealing with partitions, we can easily guarantee polynomial-time mixing. This demonstrates that mixing time analysis provides significant insights for model design, particularly for likelihood models.
>
> **Details: elimination of problem-dependent term.** We are currently focused on designing a new MCMC algorithm that eliminates problem-dependent terms and guarantees polynomial-time mixing regardless of the problem. Our strategy is inspired by the following two insights:
>
> - Our current SB method is a probabilistic model on a partition matroid.
>
> - The strong Rayleigh distribution generally possesses polynomial-time mixing MCMC inference (independently of the problem setting).
>
> These two insights suggest that if we can devise a general procedure to transform a probabilistic model on a partition matroid into a strong Rayleigh distribution, we can immediately obtain MCMC inference with guaranteed polynomial-time mixing. We have, in fact, very recently discovered such a method. Since it involves somewhat technically non-trivial topics beyond the scope of this paper, we intend to summarize the results in a separate paper.

---

> ### Author Response · Authors · 2025-11-21
> **Response to Reviewer n47q (Part 3)**
>
> ***
>
> # Effectiveness Despite Memory Redundancy
>  > What evidence supports that the redundancy overhead is worthwhile given this limitation?
>
> > The O(n^2) memory usage appears highly significant for all but trivial problem sizes.  > Contribution to Bayesian ML is minor. The unified MCMC is nice in principle, but apparently paying O(n^2) memory cost for the redundancy is not.
>
> **Answer.** Regarding the O(n^2) drawback of memory redundancy caused by model redundancy, we would like to argue against this from three perspectives.
>
>
> - *(1) Advantage of abstraction and generalization.* Our primary message is that this redundancy enables all combinatorial structure inference problems to be handled within a unified framework. This bears a striking resemblance to how deep learning unifies various application problems by leveraging the redundant network model of neural networks.
>
> - *(2) Redundant but reasonable requirements.* May we also emphasize that O(n^2) memory is a situation frequently encountered in many practical applications. Indeed, Gaussian processes often maintain covariance matrices of size n^2 relative to the number of data points n. Kernel methods also utilize similarity matrices of size n^2. Determinant point processes similarly employ L-ensembles of size n^2. As mentioned above, we emphasize that O(n^2) memory is a common situation in various practical machine learning scenarios.
>
> - *(3) Possibility of truncation.* Our O(n^2) memory requirement stems from theoretically guaranteeing the universality of the model, namely, that a sufficiently redundant sequence of length O(n^2) contains all permutations of length n. Therefore, if we are willing to sacrifice this universality, we can empirically truncate this redundant O(n^2) sequence with a shorter one. In situations where computing resources are limited (e.g., inference on mobile devices), one might want to empirically suppress space complexity. In such cases, we can reduce redundant universal objects (e.g., superpermutations) by thinning them out. The most naive method is to use a savings constant $K$ to restrict, for example, a superpermutation of length $n^2$ to only $n^2/K$ elements extracted uniformly at random. Such pruning can straightforwardly reduce the space complexity and mixing time to $1/K$. So how much of a negative impact does such empirical saving actually have on performance? We present the performance (Kendall tau distance) for cases where $K = 1, 2$, and $4$ for the consensus ranking task (described in Section 4, Task 1):
>
> | **Dataset: Sushi** | $K=1$ | $K=2$ | $K=4$ |
> | ---- | ---- | ---- | ---- |
> | **SB-c** | $20.947$ $(2.216)$ | $21.193$ $(1.486)$ | $22.482$ $(1.681)$ |
> | **SB-q** | $19.679$ $(1.935)$ | $20.855$ $(1.029)$ | $22.016$ $(1.006)$ |
>
> | **Dataset: CF** | $K=1$ | $K=2$ | $K=4$ |
> | ---- | ---- | ---- | ---- |
> | **SB-c** | $21.328$ $(1.486)$ | $21.412$ $(1.511)$ | $22.093$ $(1.492)$ |
> | **SB-q** | $21.142$ $(1.029)$ | $21.298$ $(1.157)$ | $21.855$ $(1.198)$ |
>
> | **Dataset: CB** | $K=1$ | $K=2$ | $K=4$ |
> | ---- | ---- | ---- | ---- |
> | **SB-c** | $19.481$ $(1.681)$ | $19.939$ $(1.596)$ | $19.984$ $(1.674)$ |
> | **SB-q** | $19.402$ $(1.420)$ | $19.535$ $(1.487)$ | $19.772$ $(1.287)$ |
>
> (Note: Smaller distances mean better prediction performance.)
>
> As can be seen from the experimental results, reducing the redundancy of the superpermutation by $K=2,4$ does not cause a significant decrease in performance. This is a reasonable result because the superpermutation itself is already sufficiently redundant, and even when reduced to $1/K$ of its original size, it still seems to contain most permutation candidates as its subsequences. In this way, the topic of universality when reducing the size of superpermutations is one of the interesting research topics both theoretically and experimentally. In our next phase in the near future, we aim to provide a deeper exploration of this topic.

---

> ### Author Response · Authors · 2025-11-21
> **Response to Reviewer n47q (Part 4)**
>
> ***
>
> # Improving Our Focus
>
>  > Remove or significantly de-emphasize the LTH connection unless you can establish a deeper technical relationship.
> > I'd propose focusing the paper content a bit, so that your main message is clearer. Concepts like LTH and quantum systems make it harder to see a common thread.
>
> > The paper (specifically also title and abstract) are dramatically overselling the results.
>
> **Answer.** Thank you for your enlightening suggestions. To clarify the focus of our paper, we have three improvements as follows.
>
> - We emphasized the limitations of the proposed method more prominently in the abstract and introduction sections.
>
> - We have excluded the topic of the lottery ticket hypothesis from the main text to avoid causing misunderstanding or confusion among readers.
>
> - We clarified this intent in Section 2.5 on quantum MCMC. Specifically, we explicitly stated that while traditional BML has developed separate inference algorithms for individual problems, the SB framework demonstrates that improvements to inference algorithms within a unified formulation (partition matroid) lead to developments applicable not only to specific problems but to all combinatorial structure hypothesis problems.
>
> ***
>
> # Empirical Performance
>
>  > Experiments show only "comparable" or marginally better performance than baselines.
>
> > The experimental evaluation is weak. Results show the method is "comparable", not superior (hence no real breakthrough), raising questions about practical value given the added complexity and memory costs.
>
> **Answer.** From an empirical performance perspective, we emphasize that the SB framework possesses the following two advantages.
>
> - *A general-purpose method with performance comparable to specialized methods.* Indeed, our general-purpose inference algorithms have not yet significantly surpassed specialized algorithms tailored to each specific application. As a general principle, it is a natural law that specialized methods, even if they involve high-cost implementations for specific applications, outperform generic methods. However, just as deep learning has become widespread using generic algorithms like stochastic gradient descent, we believe Bayesian machine learning could also transition towards such generic algorithms.
>
> - *Certain potential for future improvement.* We emphasize that improvements to algorithms performed within this unified framework, as demonstrated by our extension strategy such as quantum MCMC, can simultaneously drive rapid progress across various combinatorial structure inference tasks. We expect this approach to yield greater future contributions to BML than individual algorithmic improvements tailored to specific application cases.

---

> > ### Author Response · Authors · 2025-12-03
> > **Paper revision (to Reviewer n47q)**
> >
> > We have highlighted the major improvements in the revised manuscript in dark blue.
> >
> > **[Improving clarity regarding current achievements and limitations]**
> >
> > - We have significantly improved the **abstract** and **Section 1: Introduction** to thoroughly eliminate any overselling language. Specifically, the explanation in **lines 134-141** enables readers to more accurately discern the current achievements and limitations of this paper at an earlier stage of the manuscript.
> >
> > **[Improvements in clarity for problem-dependent terms in MCMC mixing time]**
> >
> > We have provided clearer guidance for readers with a stronger interest in this point:
> >
> > - We have added guidance early in the manuscript (**lines 140-141**) to lead readers to the two concrete examples in **Appendix B.4** and **Remark D.3**.
> >
> > - In **lines 483-485**, we have emphasized that this is an important issue to address in the near future.
> >
> > **[Removal of association with the lottery ticket hypothesis]**
> >
> > - We have removed the association with the lottery hypothesis from the **title**, **abstract**, and **Section 1: Introduction** (as you suggested) to focus the main text on a more concentrated discussion.
> >
> > **[Enhancing clarity of intent behind introducing quantum MCMC]**
> >
> > - In **lines 336–340** and **lines 345–346**, we have clearly explained the intent behind introducing quantum MCMC (our framework aims for improvements to one algorithm to contribute uniformly across various applications, not just individual ones).
> >
> > **[Additional experiments of ablation studies]**
> >
> > - We have added additional experimental reports, including ablation studies, in **Appendices E.2** and **E.3**.
> >
> > ***
> > We are very grateful for the constructive comments from Reviewer n47q. Although unforeseen technical difficulties prevented us from arranging an actual discussion session this year, we hope that our detailed responses and these revisions will change your impression of this paper to a favorable one. We are deeply appreciative that your constructive comments have enabled us to improve the clarity of the paper.
> >
> > Sincerely,
> >
> > The Authors

---

### Official Review · Reviewer_vZoQ · 2025-11-04

**Soundness:** 3
**Presentation:** 2
**Contribution:** 2
**Rating:** 6
**Confidence:** 3

**Summary:**

This paper introduces a new formal framework for Bayesian Machine Learning (BML), balled Super Bayes (SB). SB allowsone  to express random target objects by randomly selecting a substructure from a universal object. This model representation naturally permits a simple MCMC algorithm for Bayesian inference, as well as a theoretical upper bound on the MCMC mixing time.

**Strengths:**

- The paper proposes a unified framework for BML than can accommodate a number of problems, while also coming with an MCMC algorithm for Bayesian inference as well as a theoretical upper bound on the MCMC mixing time.
- The paper looks technically sound and quantum MCMC seems an interesting contribution (even though I wasn't able to follow all details).
- The proposed SB framework can be readily applied to many BML Application cases, including mixture models, factor models, tree models, ancestral graphs, and relational models.
- The authors experimentally assess the framework on several important BML Applications, seemingly achieving higher performance than the competitors.

**Weaknesses:**

- To me, it is not entirely clear whether the proposed framework has additional benefits, besides introducing a unified formalism. The MCMC algorithm is standard, and my understanding is that the main benefit of this framework lies in the unified language for expressing various BML problems.
- Even though the SB framework can accommodate several BML applications of interest (e.g., mixture models, factor models, tree models, ancestral graphs, and relational models), it still relies on identifying the right universal object. The authors explained in detail how the SB framework covers the aforementioned scenarios, but there is no general formula. I may well be the case that some BML applications cannot fall under the SB framework.
- The paper is extremely verbose, and this can slow down the reading process. I also feel that some claims were potentially confusing. For example, I personally failed to see how the lucky ticket hypothesis is of any relevance to this framework. The lucky ticket hypothesis in deep learning is more about the fact that a big model with many redundancies can contain a good submodel within it; furthermore, it is precisely these redundancies that facilitate learning this submodel. In SB, the global object can obviously contain all possible sub-configurations, but redundancies are not what makes learning possible. Unless I misunderstood the connection the authors are making, I feel there is a confusion there about what the lucky ticket hypothesis is really about.
- The upper bound contains a problem-dependent term, which possibly renders it less useful.

**Questions:**

- What is the main contribution of this work? The introduction of a new formalism? The fact that this formalism permits MCMC and some theory on the MCMC mixing time?
- What about BML applications not covered by the SB framework? Have the authors considered such applications?
- Is the upper bound useful in practice, given the problem-dependent term?

---

> ### Author Response · Authors · 2025-11-21
> **Response to Reviewer vZoQ (Part 1)**
>
> We greatly appreciate your thoughtful comments. We are encouraged by the positive feedback and find the constructive comments helpful in improving the paper. Each of the topics below is central and important to this paper, and we are pleased to discuss them here.
>
> ***
> # Core Contribution and Generating Value
>
> > What is the main contribution of this work? The introduction of a new formalism? The fact that this formalism permits MCMC and some theory on the MCMC mixing time?
>
> **Direst answer.** The main contribution of this paper is the new formalism; you can certainly consider the MCMC inference algorithm and its theoretical analysis as straightforward byproducts derived from this formalism.
>
> **Details.** The value created by this new formalism can be best understood by contrasting it with the history of existing Bayesian machine learning (BML). A historical breakthrough in BML was the emergence of nonparametric Bayesian methods around the year 2000. This breakthrough involved importing stochastic processes, i.e., infinite-dimensional parametric models into Bayesian machine learning. In contrast to this BNP method, our SB method can be positioned as follows:
>
> - Nonparametric Bayes: Bayesian + Stochastic Process
>
> - Super Bayes: Bayesian + Extremal Combinatorial Universal Object
>
> ***
>
> # Range of Application Cases
>
> > What about BML applications not covered by the SB framework? Have the authors considered such applications?
>
> **Direct answer.** The SB framework does not directly cover applications involving continuous quantities such as regression tasks or generative AI. On the other hand, we believe it covers almost all applications concerning structural hypotheses or combinatorial structures (we cannot think of any specific ones that are difficult to handle).
>
> **Details: range of applications covered.** In fact, the key of the SB method lies in providing a unified probabilistic generative model for permutations, partitions, binary sequences, binary trees, causal graphs, Cambrian trees, and rectangular partitions by utilizing the existence of surjections or bijections from permutations to various combinatorial structures. These combinatorial structures serve as fundamental model modules for almost all applications appearing in BML. Consequently, as shown in Table 1 of the main text, the SB method can indeed provide probabilistic generative models for almost all structural hypothesis inference tasks.
>
> **Details: handling continuous quantities.** So, how should we handle regression or generative AI within the SB method framework? We believe a simple, complementary hybrid structure combining regression models—such as neural networks or Gaussian processes—with the SB method is useful. Our SB method consists of extremal combinatorial universal objects (primarily superpermutations) and their MCMC inference. Treating these as modules allows us to unifiedly handle various structural hypotheses and combinatorial structures. Therefore, within large deep learning systems, if there is a need to handle these combinatorial structures, our SB module can be easily applied to that part.

---

> ### Author Response · Authors · 2025-11-21
> **Response to Reviewer vZoQ (Part 2)**
>
> ***
> # Handling of MCMC Mixing Time in Practice
>
>  > Is the upper bound useful in practice, given the problem-dependent term?
>
> **Direct answer.** Unfortunately, the answer is no. While it can be used to estimate the *order* (e.g., n^2, poly(n), exp(n)) of MCMC iterations, it is difficult to utilize it for exact determination of the required number of iterations. However, it should be emphasized that this is not a weakness specific to our SB method, but rather a common problem faced by all MCMC inferences for BML.
>
> **Details.** Given the problem-dependent terms, we can indeed determine the order of the upper bound on the mixing time (e.g., whether it is O(n^2) or O(exp(n))). However, there remains another technical barrier to pinpointing the exact required number of iterations (e.g., whether it is 10,000 iterations or 25,000 iterations). This stems from the difficulty in explicitly calculating the probability of the initial MCMC state, or, in other words, the difficulty in computing the partition function (normalization constant) of the posterior distribution. As a fundamental principle of MCMC inference, even if the state transitions themselves are designed to be highly efficient, selecting only initial values with extremely low probabilities can immediately create a situation where the upper bound on mixing time is poor. We wish to emphasize that this difficulty in handling initial values represents a common challenge in current BML-based MCMC theoretical analysis. To achieve the exact specification of the required number of iterations for MCMC mixing in practical scenarios, new theoretical tools will be needed to perform evaluations that explicitly avoid calculating initial state probabilities or partition functions.
>
> ***
>
> # Improving Our Focus
>
> > I personally failed to see how the lucky ticket hypothesis is of any relevance to this framework.
>
> **Answer.** Thank you for your important suggestion. We have omitted the topic of the lottery hypothesis from the main text to avoid causing misunderstanding or confusion among readers.
>
> ***
>
> # Practical Usefulness of MCMC Mixing Time Bound
>
> > The upper bound contains a problem-dependent term, which possibly renders it less useful.
>
> **Direct answer.** Regarding the interpretation of MCMC mixing time analysis, we improve clarity for two specific application examples and additionally add discussion on our policy to eliminate this problem-dependent term in the near future (with the prospect of guaranteeing polynomial-time mixing regardless of the problem).
>
> **Details: two concrete case studies.** We have demonstrated how to utilize our current MCMC mixing time analysis results through two scenarios: positive and negative.
>
> - *Negative example.* Unfortunately, as shown in Figure 3 of the supplementary materials, we can conceive of scenarios for the Traveling Salesperson Problem where the mixing time upper bound becomes exponential. However, such a bad case also represents a problem setting where a different inference algorithm could likely discover the minimum cycle more easily. From the above, it is clear that our Algorithm 1 still has significant room for improvement. This provides important insights for future algorithmic enhancements, which will be discussed later (the third term, below soon).
>
> - *Positive example.* Fortunately, we can also design specific likelihood models that allow us to bound the MCMC mixing time of Algorithm 1 to polynomial time. As a concrete example shown in Remark D.3, by assuming a nonparametric topological likelihood in a mixture model dealing with partitions, we can easily guarantee polynomial-time mixing. This demonstrates that mixing time analysis provides significant insights for model design, particularly for likelihood models.
>
> **Details: elimination of problem-dependent term.** We are currently focused on designing a new MCMC algorithm that eliminates problem-dependent terms and guarantees polynomial-time mixing regardless of the problem. Our strategy is inspired by the following two insights:
>
> - Our current SB method is a probabilistic model on a partition matroid.
>
> - The strong Rayleigh distribution generally possesses polynomial-time mixing MCMC inference (independently of the problem setting).
>
> These two insights suggest that if we can devise a general procedure to transform a probabilistic model on a partition matroid into a strong Rayleigh distribution, we can immediately obtain MCMC inference with guaranteed polynomial-time mixing. We have, in fact, very recently discovered such a method. Since it involves somewhat technically non-trivial topics beyond the scope of this paper, we intend to summarize the results in a separate paper.

---

> > ### Comment · Reviewer_vZoQ · 2025-11-24
> > **additional questions**
> >
> > I thank the reviewers for their responses. I'd have the following follow-up questions:
> > - The authors were previously making a connection to the lottery hypothesis, which they mentioned they would remove in the revised manuscript. Another connection they are making is to extremal combinatorics. However, I do not see how this connection is valid. Extremal combinatorics deals with questions of the type: "how large (or small) should a collection of objects be in order to satisfy certain conditions"? In the context of permutations, the authors cite for example "Containing all permutations" by Engen and Vatter and "Supertrees" by Defant et al. Indeed, the former asks what the shortest object containing all permutations of a give length is; and the latter seeks the smallest permutation that contains all permutations of length $k$ as patterns. But what the authors do in this work is not external combinatorics. The random permutation concatenation that they define is a much more obvious structure that trivially contains all permutations of a given length. I understand that one must still be careful with the sampling process, because not every subsequence of the random permutation concatenation is a valid permutation. But I do not see how the defined data structure of a random permutation concatenation is in any way related to the difficult problem of super-permutation structures in extremal combinatorics. Yes, they are all related to the general problem of a data structure which contains all possible permutations, but the goal in extremal combinatorics is very different.
> > - Some notations can be confusing. For instance, the authors define the likelihood in the Bayesian update as $P_{likelihood}(G \mid \sigma)$. This reads as the probability of a graph given a permutation. But I think what the authors actually mean here is the TSP probability distribution on a given graph $G$ given a permutation $\sigma$ (where low-cost paths are of course more likely)? I feel the authors should clarify the different notations.
> > - The authors mentioned they want to compute $\pi_{\mathcal{C}}(\S)$ give a fixed random generated RPC sample $\rho_n$. I was wondering whether this negatively impacts the variance of bound (5) in Theorem 2.5, or whether perhaps higher empirical speedups would be observed if a mixture of $\rho_n$'s was used.

---

> > > ### Author Response · Authors · 2025-11-25
> > > **Response to Reviewer vZoQ (additional questions; Part 1)**
> > >
> > > We are deeply grateful for your insightful discussion. We appreciate receiving such active feedback at a premier machine learning conference and respect the dedication of reviewers who contribute to the advancement of science and technology (as we authors also serve as reviewers for other papers, we aim to emulate this attitude).
> > >
> > > # Clarifying Connection to Extremal Combinatorics
> > >
> > > > Another connection they are making is to extremal combinatorics. However, I do not see how this connection is valid.
> > >
> > > **Direct answer.** Your suggestion that the genuine relevance of our paper to extremal combinatorics could be improved is quite valid. We would like to explicitly add that the question addressed in this paper, as a purely relevant aspect to extremal combinatorics, is as follows: **What is the minimum length of a redundant number sequence under the condition that a random subsequence of its partition matroid forms a uniform permutation of length n?**
> > >
> > > **Details.** We sincerely appreciate your scientifically rigorous suggestion. Indeed, extremal combinatorics is a great branch of mathematics with a long history, and its direct connection to machine learning remains largely unexplored. We are fully aware that attempting to borrow its authority through an unjustified association is scientifically unsound. However, since our research is substantially inspired by extremal combinatorics, we wish to discuss its influence as candidly as possible. Therefore, in response to your suggestions, we believe it would be appropriate to incorporate the following improvements into our manuscript.
> > >
> > > - We emphasize that historically, the central research interest in extremal combinatorics has been to find **minimal or maximal universal objects** possessing a certain universality. For example, as you have indicated, the superpermutation addresses the question: **What is the minimal length of a redundant sequence that contains a subsequence order-isomorphic to every permutation of length $n$?**
> > >
> > > - We emphasize that finding this **minimum or maximum universal object** plays a crucial role in our research from a machine learning perspective. Expressing our interest in terms of a central research question in extremal combinatorics, it is: **What is the minimum length of a redundant number sequence under the condition that a random subsequence of its partition matroid forms a uniform permutation of length n?** The objective of minimal length in this question (i.e., the requirement extremal combinatorics often imposes on universal objects) corresponds to the demand in Bayesian machine learning for **models that satisfy universality while being as non-redundant as possible (i.e., as computationally efficient as possible)**. This is considered a significant perspective linking central research questions in extremal combinatorics to practical needs in machine learning applications.
> > >
> > > - We emphasize that alternatives to our constructed universal object, random permutation concatenation (RPC)—specifically sequences shorter than n^2 that satisfy both universality and uniform randomness of subsequences—would be highly desirable for machine learning applications. Indeed, zigzag words [Miller, 2009], which address only universality, achieve this at length n(n+1)/2, roughly half the length of RPC. We conjecture that RPC may be the minimal universal object satisfying these requirements, though a shorter universal object might exist. This poses an intriguing question in extremal combinatorics.

---

> > > ### Author Response · Authors · 2025-11-25
> > > **Response to Reviewer vZoQ (additional questions; Part 2)**
> > >
> > > # Notations on Bayes' Rule
> > >
> > > >  For instance, the authors define the likelihood in the Bayesian update as $P_{\rm likelihood}(G\mid \sigma)$. This reads as the probability of a graph $G$ given a permutation $\sigma$. But I think what the authors actually mean here is the TSP probability distribution on a given graph $G$ for a permutation $\sigma$ (where low-cost paths are of course more likely)?
> > >
> > > **Direct answer.** Our notation may have caused some misunderstanding. While our notation is as intended, it may be slightly confusing due to the nature of the Traveling Salesperson Problem (TSP) (since it is not a typical example of Bayesian machine learning).
> > > To improve this, we will use notations such as $G_{\rm observation}$ and $\sigma_{\rm model}$ to reduce reader confusion.
> > >
> > > **Details.** In TSP, the graph G (cities and distances) can be regarded as the observation data, and the permutation (travel path) as the model. Therefore, based on Bayes' rule, it takes the following form.
> > >
> > > - $P_{\rm posterior}(\sigma_{\rm model}\mid G_{\rm observation})\propto P_{\rm likelihood}(G_{\rm observation}\mid \sigma_{\rm model})\cdot P_{\rm prior}(\sigma_{\rm model})$.
> > >
> > > Therefore, $P_{\rm likelihood}(G\mid \sigma)$ would be valid, since it means **Probability of Observation data given Model**. (For example, for the Gaussian likelihood corresponds to **Probability of Observation real variable given Mean and Variance**.) Representing TSP in the form of Bayesian machine learning is somewhat atypical, so it certainly tends to be misleading. We will refine our notation to avoid confusing readers. Thank you for the important point.
> > >
> > > # Risk of Fixed Universal Object (RPC; Random Permutation Concatenation)
> > >
> > > > I was wondering whether this [derived from the fixed RPC] negatively impacts the variance of bound (5) in Theorem 2.5 [Mixing time bound of MCMC], or whether perhaps higher empirical speedups would be observed if a mixture of 's was used.
> > >
> > > **Direct answer.** We explicitly state in the paper that the risk of fixing the RPC lies not in delaying the mixing time of MCMC, but in introducing a slight bias in the uniformity of the prior model.
> > >
> > > **Details.** Thank you for your insightful question. Our claims can be summarized as follows:
> > >
> > > - *No impact on MCMC mixing.* Fixing the RPC theoretically does not affect the MCMC mixing time bound. Rather, when handling multiple RPC samples in ensembles or mixture models, the hypothesis space expands, causing the MCMC chain to spread out slightly.
> > >
> > > - *Potentially negative impact on biased priors.* The potential downside of fixing the RPC is the risk of losing uniformity over the prior model. We theoretically ensure uniformity over the model (permutations in the TSP case) in terms of the expected value of the RPC. Therefore, there is a risk that uniformity could potentially be lost in fixed RPC samples.　 We wish to explicitly note this risk and discuss potential mitigation strategies: (1) using an ensemble of multiple RPC samples, and (2) potentially placing stronger weight on the likelihood side when balancing the likelihood function and prior model.

---

> > > > ### Comment · Reviewer_vZoQ · 2025-11-26
> > > > **thank you for feedback**
> > > >
> > > > I thank the authors for the follow-up responses. I appreciate the clarifications related to Bayes' rule and the risk of the fixed universal object. Regarding the connection to external combinatorics, I understand that prior work on that area inspired this work, but otherwise I feel that the connection is not pronounced, given that extremal combinatorics is usually concerned with questions of maximality or minimality (e.g., how small or large can a graph be before a certain property appears?). Overall, I feel the title may be misleading because it emphasises both the lucky ticket hypothesis as well as extremal combinatorics, even though the connections to both are not strong. In light of all that, I feel a more appropriate title may be needed for the revised manuscript.

---

> > > > > ### Author Response · Authors · 2025-12-03
> > > > > **Paper revision (to Reviewer vZoQ)**
> > > > >
> > > > > We have highlighted the major improvements in the revised manuscript in dark blue.
> > > > >
> > > > > **[Removal of association with the lottery ticket hypothesis in the main text]**
> > > > >
> > > > > - To minimize the risk of misleading readers, we have removed any association with the lottery ticket hypothesis of deep learning from the **title**, **abstract**, and **Section 1: Introduction**.
> > > > >
> > > > > **[Enhancing clarity of relevance in extremal combinatorics]**
> > > > >
> > > > > - To clarify the central (and original) research interests in extremal combinatorics, we have revised **lines 85-89** and **lines 106-107** in **Section 1: Introduction**.
> > > > >
> > > > > **[Enhancing clarity of risks posed by fixed universal objects]**
> > > > >
> > > > > - We have added a clear explanation of the risk associated with using a single RPC sample (deviation from uniformity) in **Section 2, lines 260-265**.
> > > > >
> > > > > - To quantify this risk, we have added numerical experiments using the chi-squared test in **Appendix E.1** and **Figure 12**.
> > > > >
> > > > > ***
> > > > > We are deeply grateful to Reviewer vZoQ for enabling us to improve our paper and make it more transparent. Although our discussion was cut short this year due to unforeseen technical difficulties, we are encouraged by your valuable feedback. These revisions are thanks to your thoughtful comments.
> > > > >
> > > > > Sincerely,
> > > > >
> > > > > The Authors

---

### Author Response · Authors · 2025-12-03
**Summary of paper revision (to Meta Reviewers)**

We are deeply grateful to the reviewers and meta-reviewers for providing us with such an opportunity for lively discussion. We have heard that this year, due to unforeseen technical difficulties, the burden on our meta-reviewers has become much heavier. We deeply appreciate and respect your dedication to this review process, despite the tremendous effort it requires. To help alleviate this burden even slightly, we would like to compile the paper revisions made in response to these active discussions.

***
**[Presentation]** We have made the following improvements to enhance readability for our readers in the **Title**, **Abstract**, and **Section 1: Introduction**:

- Removal of association with lottery ticket hypothesis in main text (to *Reviewer vZoQ* and *Reviewer n47q*).

- Enhancing clarity of relevance in extremal combinatorics** (to *Reviewer vZoQ*).

- Explicit focus on discrete combinatorial structures (to *Reviewer oaAQ*).

- Improving clarity regarding current achievements and limitations (to *Reviewer n47q*)

***
**[Presentation]** In **lines 336–340** and **lines 345–346**, we have clearly explained the intent and motivation regarding to the following point:

- Enhancing clarity of intent behind introducing quantum MCMC (to *Reviewer n47q*).

***
**[Clarity]** We have added clear explanations (**lines 260-265**) and numerical verification (**Appendix E.1**) regarding the following point:

- Enhancing clarity of risks posed by fixed universal objects (to *Reviewer vZoQ* and *Reviewer iyaa*).

***
**[Clarity]** We have added additional experimental reports, including MCMC diagnosis, ablation studies, and robustness of hyper parameters, in **Appendices E.2** and **E.3**:

- Additional ablation studies (to *Reviewer n47q*).

- Hyperparameter robustness report on quantum MCMC (to *Reviewer iyaa*).

- Additional investigation into large-scale data (to *Reviewer iyaa*)

***
**[Open issue]** We have added guidance early in the manuscript (**lines 140-141**) to lead readers to the two concrete examples in **Appendix B.4** and **Remark D.3**. We also have stated that it is a significant challenge for the future in **Section 5 (lines 483-485)**.

- Addition of guidance on problem-dependent terms in MCMC mixing time analysis (to *Reviewer n47q* and *Reviewer iyaa*)

***
**[Open issue]** We have explicitly included this discussion for the following topic in **Appendix D.2**.

- Outlining prospects for Informative prior design (to *Reviewer iyaa* and *Reviewer oaAQ*)

***
We sincerely thank the reviewers, meta-reviewers, and committee members for their tremendous efforts.

Sincerely,

The Authors

---

### Meta-Review · Area_Chair_2ZcK · 2026-01-07

**Summary:**

The paper proposes a framework for Bayesian modelling (specifically in the ML context) and inference that the authors coin super Bayes, which represents latent combinatorial structures/objects as random substructures of universal objects from extremal combinatorics (e.g., superpermutation). Doing so yields a common target distribution defined over index sets, enabling a single Gibbs-style MCMC algorithm for a variety of problems. Consequently, the approach is restricted to discrete objects, a potential limitation highlighted by reviewers. Moreover, for universality, the approach results in high memory requirements (quadratic in the length of the superpermutation). Authors additionally provide a bound on the mixing time, outline potential quantum computing-based acceleration, and evaluate several relevant tasks, such as hierarchical clustering, relational data analysis, and consensus ranking.

Reviewers had mixed opinions on this submission, with borderline or negative scoring dominating. Many of the raised concerns have been partially addressed, some of them representing more conceptual limitations than concerns about the work (e.g., difficulty of prior elicitation, scalability issues), others relate to overselling/misleading claims and a lack of clarity on the benefits of the unifying framework apart from unifying problem classes.

**Reviewer Concerns:**

Several of the concerns have been partially addressed; the following can be identified as remaining:

- scalability: A more thorough discussion on how to make the approach more scalable would be needed.
- prior elicitation: Formulating priors in the proposed framework seems challenging. The authors referred to work in the BNP community and challenges in this area; however, it is unclear how results from the BNP community would help in concrete terms.
- concrete benefits: The actual tangible benefits of this framework, apart from unifying problem classes, seem unclear.
- links to LTH and focus: There has been some confusion regarding links to the lottery ticket hypotheses (LTH), which has been dropped from the paper to some extent during the rebuttal. In a broader view, this and other reviewers' comments indicate that the scope of the work is not well developed and should be more focused.

**Reviewer Scores:**

Below are potential developments of the scores.

- [vZoQ]: Probably keeps the score or lowers it => 5/6
- [n47q]: Likely to have kept the score => 2
- [iyaa]: Probably kept the score => 6
- [oaAQ]: Likely to have kept or lowered the score. => 7/8

---

### Decision · Program_Chairs · 2026-01-26

Reject